Dear Editor,
thanks for these hints. In most cases it was possible to incorporate them.
We changed the text regarding your suggestions on "on the other hand".
The units are always per kg. Only when we apply eq. (2), where we multiply a volume, we use
volumetric concentrations. This also holds for table 1: These are the "wad_sta" values of eq. (2).
In the context of this manuscript TA is used as concentration. Nonetheless, it was possible to omit
the term "TA concentration" in most cases. Only when the description appears technical (like for the
description of the effect of river input on TA) we had to stay with "TA concentrations".
Line 507 (old version):
Large parts of the produced organic matter are exported out of the validation area. This implies that
uptake of nitrate would increase TA in this area. We changed this sentence: "Assuming large parts
of organic matter exported out of the validation area this production compensates the
missing TA generation by benthic denitrification. This amount of nitrate would not fully be
available for primary production if parts of it would be consumed by denitrification."
Line 528 (old version):
You are right, we changed this sentence: "Shallow oxidation of biogenic methane formed in
deep or shallow tidal flat sediments (not modelled) (Höpner & Michaelis, 1994; Neira &
Rackemann, 1996; Böttcher et al., 2007) has the potential both to lower or enhance the
buffer capacity, thus counteracting or promoting the respective effect of carbonate
dissolution."
==============================================================================
Dear Johannes Paetsch,
We are pleased to inform you that the Associate Editor report for the
following BG manuscript is now available:
Title: The impact of intertidal areas on the carbonate system of the
southern North Sea
Author(s): Fabian Schwichtenberg et al.
MS No.: bg-2020-24
MS Type: Research article
Iteration: Correction
The Associate Editor has decided that some corrections are necessary before
the manuscript can be published. Please log in using your Copernicus Office
user ID 133937 to find the Associate Editor report at:
https://editor.copernicus.org/BG/ms_records/bg-2020-24
We kindly ask you to upload the files required for the production process
no later than 23 Jul 2020 at:
https://editor.copernicus.org/BG/production_file_upload/bg-2020-24

**Associate Editor Decision: Publish subject to technical corrections** (15 Jul 2020) by Jack Middelburg

Comments to the Author:

Dear Dr. Schwichtenberg and co-authors:

Thank you for submitting your second revision to Biogeosciences. I have read it, and although some alternative processes might be involved, I am happy to inform you that your paper is now accepted for publication.

While reading I identified the need for a few minor corrections:

All through, please only use "on the other hand" when it is following a 'on the one hand'.

Please check your units again: sometimes you use micromole per liter sometimes per kg (e.g. see Table 1 vs. Table 2-4). Try to avoid use of TA concentrations if writing TA or TA level will do. See comment of one of the referees about TA equivalent vs. concentration (but this is discipline dependent).

Line 507: Logic. Nitrate uptake by phytoplankton only produces alkalinity if the organic matter is preserved. It was unclear from your text whether this is the case.

Line 528: specify aerobic oxidation here. Anaerobic oxidation may generate TA and thus enhance buffering.

With best regards

94    Jack Middelburg, Associate Editor

# The impact of intertidal areas on the carbonate system of the southern North Sea

Fabian Schwichtenberg[1,6], Johannes Pätsch[1,5], Michael Ernst Böttcher[2,3,4], Helmuth Thomas[5], Vera Winde[2], Kay-Christian Emeis[5]

[1] Theoretical Oceanography, Universität Hamburg, Bundesstr. 53, D-20146 Hamburg, Germany

[2] Geochemistry & Isotope Biogeochemistry Group, Department of Marine Geology, Leibniz Institute of Baltic Sea Research (IOW), Seestr. 15, D-18119 Warnemünde, Germany

[3] Marine Geochemistry, University of Greifswald, Friedrich-Ludwig-Jahn Str. 17a, D-17489 Greifswald, Germany

[4] Department of Maritime Systems, Interdisciplinary Faculty, University of Rostock, Albert-Einstein-Str. 21, D-18059 Rostock, Germany

[5] Institute of Coastal Research, Helmholtz Zentrum Geesthacht (HZG), Max-Planck-Str. 1, D-21502 Geesthacht, Germany

[6] Present Address: German Federal Maritime and Hydrographic Agency, Bernhard-Nocht-Str. 78, D-20359 Hamburg, Germany

Correspondence to Johannes Pätsch (johannes.paetsch@uni-hamburg.de)

**Abstract**

The coastal ocean is strongly affected by ocean acidification because of its shallow water depths, low volume, and the closeness to terrestrial dynamics. Earlier observations of dissolved inorganic carbon (DIC) and total alkalinity (TA) in the southern part of the North Sea, a Northwest-European shelf sea, revealed lower acidification effects than expected. It has been assumed that anaerobic degradation and subsequent TA release in the adjacent back-barrier tidal areas ('Wadden Sea') in summer time is responsible for this phenomenon. In this study the exchange rates of TA and DIC between the Wadden Sea tidal basins and the North Sea and the consequences for the carbonate system in the German Bight are estimated using a 3-D ecosystem model. The aim of this study is to differentiate the various sources contributing to observed high summer TA in the southern North Sea. Measured TA

and DIC in the Wadden Sea are considered as model boundary conditions. This procedure acknowledges the dynamic behaviour of the Wadden Sea as an area of effective production and decomposition of organic material. According to the modelling results, 39 Gmol TA yr$^{-1}$ were exported from the Wadden Sea into the North Sea, which is less than a previous estimate, but within a comparable range. The interannual variabilities of TA and DIC, mainly driven by hydrodynamic conditions, were examined for the years 2001 – 2009. Dynamics in the carbonate system is found to be related to specific weather conditions. The results suggest that the Wadden Sea is an important driver for the carbonate system in the southern North Sea. On average 41 % of TA inventory changes in the German Bight were caused by riverine input, 37 % by net transport from adjacent North Sea sectors, 16 % by Wadden Sea export, and 6 % are caused by internal net production of TA. The dominant role of river input for the TA inventory disappears when focussing on TA concentration changes due to the corresponding freshwater fluxes diluting the marine TA concentrations. The ratio of exported TA versus DIC reflects the dominant underlying biogeochemical processes in the Wadden Sea. Whereas, aerobic degradation of organic matter plays a key role in the North Frisian Wadden Sea during all seasons of the year, anaerobic degradation of organic matter dominated in the East Frisian Wadden Sea. Despite of the scarcity of high-resolution field data it is shown that anaerobic degradation in the Wadden Sea is one of the main contributors of elevated summer TA values in the southern North Sea.

## 1. Introduction

Shelf seas are highly productive areas constituting the interface between the inhabited coastal areas and the global ocean. Although they represent only 7.6 % of the world ocean's area, current estimates assume that they contribute approximately 21 % to total global ocean $CO_2$ sequestration (Borges, 2011). At the global scale the uncertainties of these estimates are significant due to the lack of spatially and temporally resolved field data. Some studies investigated regional carbon cycles in detail (e.g., Kempe & Pegler, 1991; Brasse et al., 1999; Reimer et al., 1999; Thomas et al., 2004; 2009; Artioli et al., 2012; Lorkowski et al., 2012; Burt et al., 2016; Shadwick et al., 2011; Laruelle et al., 2014; Carvalho et al., 2017) and pointed out sources of uncertainties specifically for coastal settings.

However, natural pH dynamics in coastal- and shelf- regions, for example, have been shown
to be up to an order of magnitude higher than in the open ocean (Provoost et al, 2010).
Also, the nearshore effects of $CO_2$ uptake and acidification are difficult to determine,
because of the shallow water depth and a possible superposition by benthic-pelagic
coupling, and strong variations in fluxes of TA are associated with inflow of nutrients from
rivers, pelagic nutrient driven production and respiration (Provoost et al., 2010), submarine
groundwater discharge (SGD; Winde et al., 2014), and from benthic-pelagic pore water
exchange (e.g., Billerbeck et al., 2006; Riedel et al., 2010; Moore et al., 2011; Winde et al.,
2014; Santos et al., 2012; 2015; Brenner et al., 2016; Burt et al., 2014; 2016; Seibert et al.,
2019). Finally, shifts within the carbonate system are driven by impacts from watershed
processes and modulated by changes in ecosystem structure and metabolism (Duarte et al.,

71 2013).

Berner et al. (1970) and Ben-Yakoov (1973) were among the first who investigated elevated
TA and pH variations caused by microbial dissimilatory sulphate reduction in the anoxic pore
water of sediments. At the Californian coast, the observed enhanced TA export from
sediments was related to the burial of reduced sulphur compounds (pyrite) (Dollar et al.,
1991; Smith & Hollibaugh, 1993; Chambers et al., 1994). Other studies conducted in the
Satilla and Altamaha estuaries and the adjacent continental shelf found non-conservative
mixing lines of TA versus salinity, which was attributed to anaerobic TA production in
nearshore sediments (Wang & Cai, 2004; Cai et al., 2010). Iron dynamics and pyrite
formation in the Baltic Sea were found to impact benthic TA generation from the sediments
(Gustafsson et al., 2019; Łukawska-Matuszewska and Graca, 2017).
The focus of the present study is the southern part of the North Sea, located on the
Northwest-European Shelf. This shallow part of the North Sea is connected with the tidal
basins of the Wadden Sea via channels between barrier islands enabling an exchange of
water, and dissolved and suspended material (Rullkötter, 2009; Lettmann et al., 2009;
Kohlmeier and Ebenhöh, 2009). The Wadden Sea extends from Den Helder (The
Netherlands) in the west to Esbjerg (Denmark) in the north and covers an area of about
9500 km$^2$ (Ehlers, 1994). The entire system is characterised by semidiurnal tides with a tidal
range between 1.5 m in the westernmost part and 4 m in the estuaries of the rivers Weser
and Elbe (Streif, 1990). During low tide about 50 % of the area are falling dry (van Beusekom
et al., 2019). Large rivers discharge nutrients into the Wadden Sea, which in turn shows a
high degree of eutrophication, aggravated by mineralisation of organic material imported
into the Wadden Sea from the open North Sea (van Beusekom et al., 2012).
In comparison to the central and northern part of the North Sea, TA levels in the southern
part are significantly elevated during summer (Salt et al., 2013; Thomas et al., 2009; Brenner
et al., 2016; Burt et al., 2016). The observed high TA levels have been attributed to an impact
from the adjacent tidal areas (Hoppema, 1990; Kempe & Pegler, 1991; Brasse et al., 1999;
Reimer et al., 1999; Thomas et al., 2009; Winde et al., 2014), but this impact has not been
rigorously quantified. Using several assumptions, Thomas et al. (2009) calculated an annual
TA export from the Wadden Sea / Southern Bight of 73 Gmol TA yr$^{-1}$ to close the TA budget
for the southern North Sea.
The aim of this study is to reproduce the elevated summer levels of TA in the southern North
Sea with a 3D biogeochemical model that has TA as prognostic variable. With this tool at
hand, we balance the budget TA in the relevant area on an annual basis. Quantifying the
different budget terms, like river input, Wadden Sea export, internal pelagic and benthic
production, degradation and respiration allows us to determine the most important
contributors to TA variations. In this way we refine the budget terms by Thomas et al. (2009)
and replace the original closing term by data. The new results are discussed on the
background of the budget approach proposed by Thomas et al. (2009).

## 2. Methods
### 2.1. Model specifications

#### 2.1.1. Model domain and validation area

The ECOHAM model domain for this study (Fig. 1) was first applied by Pätsch et al. (2010).
For model validations (magenta: validation area, Fig. 1), an area was chosen that includes
the German Bight as well as parts along the Danish and the Dutch coast. The western
boundary of the validation area is situated at 4.5° E. The southern and northern boundaries
are at 53.5° and 55.5° N, respectively. The validation area is divided by the magenta dashed
line at 7° E into the western and eastern part. For the calculation of box averages of DIC and
TA a bias towards the deeper areas with more volume and more data should be avoided.
Therefore, each water column covered with data within the validation area delivered one
mean value, which is calculated by vertical averaging. These mean water column averages
were horizontally interpolated onto the model grid. After this procedure average box values
were calculated. In case of box-averaging model output, the same procedure was applied,
but without horizontal interpolation.

### 2.1.2. The hydrodynamic module

The physical parameters temperature, salinity, horizontal and vertical advection as well as
turbulent mixing were calculated by the submodule HAMSOM (Backhaus, 1985), which was
integrated in the ECOHAM model. It is a baroclinic primitive equation model using the
hydrostatic and Boussinesq approximation. It is applied to several regional sea areas
worldwide (Mayer et al., 2018; Su & Pohlmann, 2009). Details are described by Backhaus &
Hainbucher (1987) and Pohlmann (1996). The hydrodynamic model ran prior to the
biogeochemical part. Daily result fields were stored for driving the biogeochemical model in
offline mode. Surface elevation, temperature and salinity resulting from the Northwest-
European Shelf model application (Lorkowski et al., 2012) were used as boundary conditions
at the southern and northern boundaries. The temperature of the shelf run by Lorkowski et
al. (2012) showed a constant offset compared with observations (their Fig. 3), because
incoming solar radiation was calculated too high. For the present simulations the shelf run
has been repeated with adequate solar radiation forcing.
River-induced horizontal transport due to the hydraulic gradient is incorporated (Große et
al., 2017; Kerimoglu et al., 2018). This component of the hydrodynamic horizontal transport
corresponds to the amount of freshwater discharge.
Within this study we use the term flushing time. It is the average time when a basin is filled
with laterally advected water. The flushing time depends on the specific basin: large basins
have usually higher flushing times than smaller basins. High flushing times correspond with
low water renewal times.

### 2.1.3. The biogeochemical module

The relevant biogeochemical processes and their parameterisations have been detailed in
Lorkowski et al. (2012). In former model setups TA was restored to prescribed values derived
from observations (Thomas et al., 2009) with a relaxation time of two weeks (Kühn et al.,
2010; Lorkowski et al., 2012). The changes in TA treatment for the study at hand is described

below. Results from the Northwest-European Shelf model application (Lorkowski et al., 2012) were used as boundary conditions for the recent biogeochemical simulations at the southern and northern boundaries (Fig. 1).

The main model extension was the introduction of a prognostic treatment of TA in order to study the impact of biogeochemical and physical driven changes of TA onto the carbonate system and especially on acidification (Pätsch et al., 2018). The physical part contains advective and mixing processes as well as dilution by riverine freshwater input. The pelagic biogeochemical part is driven by planktonic production and respiration, formation and dissolution of calcite, pelagic and benthic degradation and remineralisation, and also by atmospheric deposition of reduced and oxidised nitrogen. All these processes impact TA. In this model version benthic denitrification has no impact on pelagic TA. Other benthic anaerobic processes are not considered. Only the carbonate ions from benthic calcite dilution increase pelagic TA. Aerobic remineralisation releases ammonium and phosphate, which enter the pelagic system across the benthic-pelagic interface and alter the pelagic TA. The theoretical background to this has been outlined by Wolf-Gladrow et al. (2007).

The years 2001 to 2009 were simulated with 3 spin up years in 2000. Two different scenarios (A and B) were conducted. Scenario A is the reference scenario without implementation of any Wadden Sea processes. For scenario B we used the same model configuration as for scenario A and additionally implemented Wadden Sea export rates of TA and DIC as described in section 2.3.1. The respective Wadden Sea export rates (Fig. 2) are calculated by the temporal integration of the product of wad_sta and wad_exc over one month (see section 2.3.1, equation 2).

## 2.2. External sources and boundary conditions

### 2.2.1. Freshwater discharge

Daily data of freshwater fluxes from 16 rivers were used (Fig. 1). For the German Bight and the other continental rivers daily observations of runoff provided by Pätsch & Lenhart (2008) were incorporated. The discharges of the rivers Elbe, Weser and Ems were increased by 21 %, 19 % and 30 % in order to take additional drainage into account that originated from the area downstream of the respective points of observation (Radach and Pätsch, 2007). The respective tracer loads were increased accordingly. The data of Neal (2002) were

implemented for the British rivers for all years with daily values for freshwater. The annual
amounts of freshwater of the different rivers are shown in the appendix (Table A1). Riverine
freshwater discharge was also considered for the calculation of the concentrations of all
biogeochemical tracers in the model.
### 2.2.2. River input
**Data sources**
River load data for the main continental rivers were taken from the report by Pätsch &
Lenhart (2008) that was kept up to date continuously so that data for the years 2007 – 2009
were also available (https://wiki.cen.uni-hamburg.de/ifm/ECOHAM/DATA_RIVER). They
calculated daily loads of nutrients and organic matter based on data provided by the
different river authorities. Additionally, loads of the River Eider were calculated according to
Johannsen et al. (2008).
Up to now, all ECOHAM applications used constant riverine DIC concentrations. TA was not
used. For the study at hand we introduced time varying riverine TA and DIC values. New data
of freshwater discharge were introduced, as well as TA and DIC loads for the British rivers
(Neal, 2002). Monthly mean concentrations of nitrate, TA and DIC were added for the Dutch
rivers (www.waterbase.nl) and for the German river Elbe (Amann et al., 2015). The Dutch
river data were observed in the years 2007 – 2009. The river Elbe data were taken in the
years 2009 – 2011. These concentration data were prescribed for all simulation years as
mean annual cycle.
The data sources and positions of the river mouths of all 16 rivers are shown in Table A2 and
in Fig. 1. The respective riverine concentrations of TA and DIC are given in Table A3.
Schwichtenberg (2013) describes the river data in detail.
A few small flood gates ("Siel") and rivers transport fresh water from the recharge areas into
the intertidal areas (Streif, 1990). The recharge areas for these inlets differ considerably
from each other, leading to different relative contributions for the fresh water input.
Whereas the catchments of Schweiburger Siel (22.2 km$^2$) and the Hooksieler Binnentief are
only of minor importance, the Vareler Siel, the Eckenwarder Siel, and the Maade Siel are of
medium importance, and the highest contribution may originate from the Wangersiel, the
Dangaster Siel, and the Jade-Wapeler Siel (Lipinski, 1999).

**Effective river input**

In order to analyse the net effect on concentrations in the sea due to river input, the effective river input ($Riv_{eff}$ [Gmol yr$^{-1}$]) is introduced:


$$Riv_{eff} = \frac{\Delta C|_{riv}}{\rho \cdot yr} \cdot V \cdot C \qquad\qquad (1)$$


with $\Delta C|_{riv}$ [µmol kg$^{-1}$]: the concentration change in the river mouth cell due to river load *riv* and the freshwater flux from the river. $V$ [l] is the volume of the river mouth cell, $\rho$ [kg l$^{-1}$] density of water, yr is one year, C [$10^{-15}$ l$^{-1}$] is a constant.

Bulk alkalinity discharged by rivers is quite large but most of the rivers entering the North Sea (here the German Bight) have lower TA concentrations than the sea water. In case of identical concentrations, the effective river load $Riv_{eff}$ is zero. The TA related molecules enter the sea, and in most cases, they are leaving it via transport. In case of tracing or budgeting both the real TA river discharge and the transport must be recognized. In order to understand TA concentration changes in the sea $Riv_{eff}$ is appropriate.

226

### *2.2.3. Meteorological forcing*

The meteorological forcing was provided by NCEP Reanalysis (Kalnay et al., 1996) and interpolated on the model grid field. It consisted of six-hourly fields of air temperature, relative humidity, cloud coverage, wind speed, atmospheric pressure, and wind stress for every year. 2-hourly and daily mean short wave radiation were calculated from astronomic insolation and cloudiness with an improved formula (Lorkowski et al., 2012).

### **2.3. The Wadden Sea**

### *2.3.1. Implementation of Wadden Sea dynamics*

For the present study the exchange of TA and DIC between North Sea and Wadden Sea was implemented into the model by defining sinks and sources of TA and DIC for some of the south-eastern cells of the North Sea grid (Fig. 1). The cells with adjacent Wadden Sea were

separated into three exchange areas: The East Frisian, the North Frisian Wadden Sea and the
Jade Bay, marked by "E", "N" and "J" (Fig. 1, right side).
Two parameters were determined in order to quantify the TA and DIC exchange between
the Wadden Sea and the North Sea.

242       1.  Concentration changes of pelagic TA and DIC in the Wadden Sea during one tide, and

243       2.  Water mass exchange between the back-barrier islands and the open sea during one

244          tide

Measured concentrations of TA and DIC (Winde, 2013; Winde et al., 2014) as well as
modelled water mass exchange rates of the export areas by Grashorn (2015) served as bases
for the calculated exchange. Details on flux calculations and measurements are described
below. The daily Wadden Sea exchange of TA and DIC was calculated as:

$$wad\_flu = \frac{wad\_sta * wad\_exc}{vol} \qquad (2)$$

Differences in measured concentrations in the Wadden Sea during rising and falling water
levels, as described in section 2.3.2, were temporally interpolated and summarized as
$wad\_sta$ [mmol m$^{-3}$]. Modelled daily Wadden Sea exchange rates of water masses (tidal
prisms during falling water level) were defined as $wad\_exc$ [m$^3$ d$^{-1}$], and the volume of the
corresponding North Sea grid cell was $vol$ [m$^3$]. $wad\_flu$ [mmol m$^{-3}$ d$^{-1}$] were the daily
concentration changes of TA and DIC in the respective North Sea grid cells.
In fact, some amounts of the tidal prisms return without mixing with North Sea water, and
calculations of Wadden Sea – North Sea exchange should therefore consider flushing times
in the respective back-barrier areas. Since differences in measured concentrations between
rising and falling water levels were used, this effect is already assumed to be represented in
the data. This approach enabled the use of tidal prisms without consideration of any flushing
times.

261       **2.3.2. Wadden Sea - measurements**

The flux calculations for the Wadden Sea – North Sea exchange were carried out in tidal
basins of the East and North Frisian Wadden Sea (Spiekeroog Island, Sylt-Rømø) as well as in
the Jade Bay. For the present study seawater samples representing tidal cycles during
different seasons (Winde, 2013). The mean concentrations of TA and DIC during rising and
falling water levels and the respective differences (ΔTA and ΔDIC) are given in Table 1.
Measurements in August 2002 were taken from Moore et al. (2011). The Δ-values were used
as *wad_sta* and were linearly interpolated between the times of observations for the
simulations. In this procedure, the linear progress of the Δ-values does not represent the
natural behaviour perfectly, especially if only few data are available. As a consequence,
possible short events of high TA and DIC export rates that occurred in periods outside the
observation periods may have been missed.
Due to the low number of concentration measurements a statistical analysis of uncertainties
of ΔTA and ΔDIC was not possible. They were measured with a lag of 2 hours after low tide
and high tide. This was done in order to obtain representative concentrations of rising and
falling water levels. As a consequence, only 2 - 3 measurements for each location and season
were considered for calculations of ΔTA and ΔDIC.
### 2.3.3. Wadden Sea – modelling the exchange rates
Grashorn (2015) performed the hydrodynamic computations of exchanged water masses
(*wad_exc*) with the model FVCOM (Chen et al., 2003) by adding up the cumulative seaward
transport during falling water level (tidal prisms) between the back-barrier islands that were
located near the respective ECOHAM cells with adjacent Wadden Sea area. These values are
given in Table 2 for each ECOHAM cell in the respective export areas. The definition of the
first cell N1 and the last cell E4 is in accordance to the clockwise order in Fig. 1 (right side).
The mean daily runoff of all N-, J- and E-positions was 8.1 km³ d$^{-1}$, 0.8 km³ d$^{-1}$ and 2.3 km³ d$^{-1}$
respectively.
### *2.3.4. Additional Sampling of DIC and TA*
DIC and TA for selected freshwater inlets sampled in October 2010 and May 2011 are
presented in Table 3. Sampling and analyses took place as described by Winde et al. (2014)
and are here reported for completeness and input for discussion only. The autumn data are
deposited under doi:10.1594/PANGEA.841976. The samples for TA measurements were
filled without headspace into pre-cleaned 12 ccm Exetainer®, filled with 0.1 ml saturated
HgCl$_2$ solution. The samples for DIC analysis were completely filled into 250 ccm ground-
glass-stoppered bottles, and then poisoned with 100 µl of a saturated HgCl$_2$ solution. The
DIC concentrations were determined at IOW by coulometric titration according to Johnson
et al. (1993), using reference material provided by A. Dickson (University of California, San
Diego; Dickson et al., 2003) for the calibration (batch 102). TA was measured by
potentiometric titration using HCl using a Schott titri plus equipped with an IOline electrode
A157. Standard deviations for DIC and TA measurements were better than +/-2 and +/-
10 µmol kg$^{-1}$, respectively.

### 2.4. Statistical analysis

A statistical overview of the simulation results in comparison to the observations (Salt et al.,
2013) is given in Table 4 and 5. In the validation area (magenta box in Fig. 1) observations of
10 different stations were available, each with four to six measurements at different depths
(51 measured points). Measured TA and DIC of each point were compared with modelled TA
and DIC in the respective grid cells, respectively. The standard deviations (Stdv), the root
means square errors (RMSE), and correlation coefficients (r) were calculated for each
simulation. In addition to the year 2008, which we focus on in this study, observations were
performed at the same positions in summer 2005 and 2001. These data are also statistically
compared with the model results.

## 3. Results

### 3.1. Model validation - TA in summer 2008

The results of scenarios A and B were compared with observations of TA in August 2008 (Salt
et al., 2013) for surface water. The observations revealed high TA levels in the German Bight
(east of 7° E and south of 55° N) and around the Danish coast (around 56° N) as shown in
Fig. 3a. The observed concentrations in these areas ranged between 2350 and
2387 µmol TA kg$^{-1}$. These findings were in accordance with observed TA in August /
September 2001 (Thomas et al., 2009). TA in other parts of the observation domain ranged
between 2270 µmol TA kg$^{-1}$ near the British coast (53° N – 56° N) and 2330 µmol TA kg$^{-1}$ near
the Dutch coast and the Channel. In the validation box the overall average and the standard
deviation of all observed TA concentrations (Stdv) was 2334 and 33 µmol TA kg$^{-1}$,
respectively.
In scenario A the simulated surface TA showed a more homogeneous pattern than
observations with maximum values of 2396 µmol TA kg$^{-1}$ at the western part of the Dutch
coast and even higher (2450 µmol TA kg$^{-1}$) in the river mouth of the Wash estuary at the
British coast. Minimum values of 2235 and 2274 µmol TA kg$^{-1}$ were simulated at the mouths
of the rivers Elbe and Firth of Forth. The modelled TA ranged from 2332 to 2351 µmol TA kg$^{-1}$
in the German Bight and in the Jade Bay. Strongest underestimations in relation to
observations are located in a band close to the coast stretching from the East Frisian Islands
to 57° N at the Danish coast (Fig. 4a). The deviation of simulation results of scenario A from
observations in the validation box was represented by a RMSE of 28 µmol TA kg$^{-1}$. The
standard deviation was 7 µmol TA kg$^{-1}$ and the correlation amounted to r = 0.77 (Table 4). In
the years 2005 and 2001 similar statistical values are found, but the correlation coefficient
was smaller.
The scenario B was based on a Wadden Sea export of TA and DIC as described above. The
major difference in TA of this scenario compared to A occurred east of 6.5° E. Surface TA
there peaked in the Jade Bay (2769 µmol TA kg$^{-1}$) and were elevated off the North Frisian
and Danish coasts from 54.2° to 56° N (> 2400 µmol TA kg$^{-1}$). Strongest underestimations in
relation to observations are noted off the Danish coast between 56° and 57° N (Fig. 4b). In
the German Bight the model overestimated the observations slightly, while at the East
Frisian Islands the model underestimates TA. When approaching the Dutch Frisian Islands
the simulation overestimates TA compared to observations and strongest overestimations
can be seen near the river mouth of River Rhine. Compared to scenario A the simulation of
scenario B was closer to the observations in terms of RMSE (18 µmol TA kg$^{-1}$) and the
standard deviation (Stdv = 22 µmol TA kg$^{-1}$). Also, the correlation (r = 0.86) improved
(Table 4). In the years 2001 and 2005 the observed mean values are slightly overestimated
by the model. The statistical values for 2001 are better than for 2005, where scenario A
better compares with the observations.

### 351    *3.2. Model validation - DIC concentrations in summer 2008*

Analogously to TA the simulation results were compared with surface observations of DIC in
summer 2008 (Salt et al., 2013). They also revealed high values in the German Bight (east of
7° E and south of 55° N) and around the Danish coast (near 56° N) which is shown in Fig. 5.
The observed DIC concentrations in these areas ranged between 2110 and 2173 µmol DIC kg$^{-}$
$^{1}$. Observed DIC concentrations in other parts of the model domain ranged between 2030
and 2070 µmol DIC kg$^{-1}$ in the north western part and 2080 - 2117 µmol DIC kg$^{-1}$ at the Dutch
coast. In the validation box the overall average and the standard deviation of all observed
DIC concentrations were 2108 and 25.09 µmol DIC kg$^{-1}$, respectively.
The DIC concentrations in scenario A ranged between 1935 and 1977 µmol DIC kg$^{-1}$ at the
North Frisian and Danish coast (54.5° N - 55.5° N) and 1965 µmol DIC kg$^{-1}$ in the Jade Bay.
Maxima of up to 2164 µmol DIC kg$^{-1}$ were modelled at the western part of the Dutch coast
north of the mouth of River Rhine (Fig. 5). The DIC concentrations in the German Bight
showed a heterogeneous pattern in the model, and sometimes values decreased from west
to east, which contrasts the observations (Fig. 5a). This may be the reason for the negative
correlation coefficient r = -0.64 between model and observations (Table 5). The significant
deviation from observation of results from scenario A is also indicated by the RMSE of
43 µmol DIC kg$^{-1}$, and a standard deviation of 14 µmol DIC kg$^{-1}$. In 2001 and 2005 the
simulation results of this scenario A are better, which is expressed in positive correlation
coefficients and small RMSE values.
In scenario B the surface DIC concentrations at the Wadden Sea coasts increased: The North
Frisian coast shows concentrations of up to 2200 µmol DIC kg$^{-1}$ while the German Bight has
values of 2100 − 2160 µmol DIC kg$^{-1}$, and Jade Bay concentrations were higher than
2250 µmol DIC kg$^{-1}$. The other areas are comparable to scenario A. In scenario B the RMSE in
the validation box decreased to 26 µmol DIC kg$^{-1}$ in comparison to scenario A. The standard
deviation decreased to 9.1 µmol DIC kg$^{-1}$, and the correlation improved to r = 0.55 (Table 5).
The average values are close to the observed ones for all years, even though in 2005 a large
RMSE was found.
The comparison between observations and simulation results of scenario A (Fig. 4c) clearly
show model underestimations in the south-eastern area and are strongest in the inner
German Bight towards the North Frisian coast (> 120 µmol DIC kg$^{-1}$). Scenario B also models
values lower than observations in the south-eastern area (Fig. 4d), but the agreement
between observation and model results is reasonable. Only off the Danish coast near 6.5° E,
56° N the model underestimates DIC by 93 µmol DIC kg$^{-1}$.

### 3.3. Hydrodynamic conditions and flushing times

The calculations of Wadden Sea TA export in Thomas et al. (2009) were based on several assumptions concerning riverine input of bulk TA and nitrate, atmospheric deposition of NOx, water column inventories of nitrate and the exchange between the Southern Bight and the adjacent North Sea (Lenhart et al., 1995). The latter was computed by considering that the water in the Southern Bight is flushed with water of the adjacent open North Sea at time scales of six weeks. For the study at hand, flushing times in the validation area in summer and winter are presented for the years 2001 to 2009 in Fig. 6. Additionally, monthly mean flow patterns of the model area are presented for June, July and August for the years 2003 and 2008, respectively (Fig. 7). They were chosen to highlight the pattern in summer 2003 with one of the highest flushing times (lowest water renewal times), and that in 2008 corresponding to one of the lowest flushing times (highest water renewal times).

The flushing times were determined for the three areas 1 – validation area, 2 – western part of the validation area, 3 – eastern part of the validation area. They were calculated by dividing the total volume of the respective areas 1 – 3 by the total inflow into the areas $m^3$ $(m^3 s^{-1})^{-1}$. Flushing times (rounded to integer values) were consistently higher in summer than in winter, meaning that highest inflow occurred in winter. Summer flushing times in the whole validation area ranged from 54 days in 2008 to 81 days in 2003 and 2006, whereas the winter values in the same area ranged from 32 days in 2008 to 51 days in 2003 and 2009. The flushing times in the western and eastern part of the validation area were smaller due to the smaller box sizes. Due to the position, flushing times in the western part were consistently shorter than in the eastern part. These differences ranged from 5 days in winter 2002 to 14 days in summer 2006 and 2008. The interannual variabilities of all areas were higher in summer than in winter.

The North Sea is mainly characterised by an anti-clockwise circulation pattern (Otto et al., 1990; Pätsch et al., 2017). This can be observed for the summer months in 2008 (Fig. 7). More disturbed circulation patterns in the south-eastern part of the model domain occurred in June 2003: In the German Bight and in the adjacent western area two gyres with reversed rotating direction are dominant. In August 2003 the complete eastern part shows a clockwise rotation which is due to the effect of easterly winds as opposed to prevalent westerlies. In this context such a situation is called meteorological blocking situation.

**3.4. Seasonal and interannual variability of TA and DIC**
The period from 2001 to 2009 was simulated for the scenarios A and B. For both scenarios
monthly mean surface TA was calculated in the validation area and are shown in Fig. 8a and
8b. The highest TA in scenario A was 2329 µmol TA kg$^{-1}$ and occurred in July 2003. The
lowest TA in each year was about 2313 to 2318 µmol TA kg$^{-1}$ and occurred in February and
March. Scenario B showed generally higher values: Summer concentrations were in the
range of 2348 to 2362 µmol TA kg$^{-1}$ and the values peaked in 2003. The lowest values
occurred in the years 2004 – 2008. Also, winter values were higher in scenario B than in
scenario A: They range from 2322 to 2335 µmol TA kg$^{-1}$.

Corresponding to TA, monthly mean surface DIC in the validation area is shown in Fig. 8c and
8d. In scenario A the concentrations increased from October to February and decreased
from March to August (Fig. 8c). In scenario B the time interval with increasing concentrations
was extended into March. Maximum values of 2152 to 2172 µmol DIC kg$^{-1}$ in scenario A
occur in February and March of each model year, and minimum values of 2060 to
2080 µmol DIC kg$^{-1}$ in August. Scenario B shows generally higher values: Highest values in
February and March are 2161 to 2191 µmol DIC kg$^{-1}$. Lowest values in August range from
2095 to 2112 µmol DIC kg$^{-1}$. The amplitude of the annual cycle is smaller in scenario B,
because the Wadden Sea export shows highest values in summer (Fig. 2).
The pattern of monthly TA and DIC of the reference scenario A differ drastically in that TA
does not show a strong seasonal variability, whereas DIC does vary significantly. In case of
DIC this is due to the biological drawdown during summer. Contrariwise, the additional input
(scenario B) from the Wadden Sea in summer creates a strong seasonality for TA and instead
flattens the variations in DIC.
**4. Discussion**

Thomas et al. (2009) estimated the contribution of shallow intertidal and subtidal areas to
the alkalinity budget of the SE North Sea. That estimate (by closure of mass fluxes) was
about 73 Gmol TA yr$^{-1}$ originating from the Wadden Sea fringing the southern and eastern
coast. These calculations were based on observations from the CANOBA dataset in 2001 and
2002. The observed high TA levels in the south-eastern North Sea were also encountered in
August 2008 (Salt et al., 2013) and these measurements were used for the main model
validation in this study. Our simulations result in 39 Gmol TA yr$^{-1}$ as export from the Wadden
Sea into the North Sea. Former modelling studies of the carbonate system of the North Sea
(Artioli et al., 2012; Lorkowski et al., 2012) did not consider the Wadden Sea as a source of
TA and DIC, and good to reasonable agreement to observations from the CANOBA dataset
was only achieved in the open North Sea in 2001 / 2002 (Thomas et al., 2009). Subsequent
simulations that included TA export from aerobic and anaerobic processes in the sediment
improved the agreement between data and models (Pätsch et al., 2018). When focusing on
the German Bight, however, the observed high TA levels in summer measurements east of
7° E could not be simulated satisfactorily.
The present study confirms the Wadden Sea as an important TA source for the German Bight
and quantifies the annual Wadden Sea TA export rate to 39 Gmol TA yr$^{-1}$. Additionally, the
contributions by most important rivers have been more precisely quantified and narrow
down uncertainties in the budgets of TA and DIC in the German Bight. All steps that were
required to calculate the budget including uncertainties are discussed in the following.

*4.1. Uncertainties of Wadden Sea – German Bight exchange rates of TA and DIC*
The Wadden Sea is an area of effective benthic decomposition of organic material (Böttcher
et al., 2004; Billerbeck et al., 2006; Al-Rai et al., 2009; van Beusekom et al., 2012) originating
both from land and from the North Sea (Thomas et al., 2009). In general, anaerobic
decomposition of the organic matter generates TA and increases the $CO_2$ buffer capacity of
seawater. On longer time scales TA can only be generated by processes that involve
permanent loss of anaerobic remineralisation products (Hu and Cai, 2011). A second
precondition is the nutrient availability to produce organic matter, which in turn serves as
necessary component of anaerobic decomposition (Gustafsson et al., 2019). The Wadden
Sea export rates of TA and DIC modelled in the present study are based on concentration
measurements during tidal cycles in the years 2002 and 2009 to 2011 (Table 1), and on
calculated tidal prisms of two day-periods that are considered to be representative of annual
mean values. This approach introduces uncertainties with respect to the true amplitudes of
concentrations differences in the tidal cycle and in seasonality due to the fact that
differences in concentrations during falling and rising water levels were linearly interpolated.
These interpolated values are based on four to five measurements in the three export areas
and were conducted in different years. Consequently, the approach does not reproduce the
exact TA and DIC levels in the years 2001 to 2009, because only meteorological forcing, river
loads and nitrogen deposition were specified for these particular years. The simulation of
scenario B thus only approximates Wadden Sea export rates. More measurements
distributed with higher resolution over the annual cycle would clearly improve our
estimates. Nevertheless, the implementation of Wadden Sea export rates here results in
improved reproduction of observed high TA levels in the German Bight in summer in
comparison to the reference run A (Fig. 3).
We calculated the sensitivity of our modelled annual TA export rates on uncertainties of the
$\Delta$-values of Table 1. As the different areas North- and East Frisian Wadden Sea and Jade Bay
has different exchange rates of water, for each region the uncertainty of 1 $\mu$mol kg$^{-1}$ in $\Delta$TA
at all times has been calculated. The East Frisian Wadden Sea export would differ by
0.84 Gmol TA yr$^{-1}$, the Jade Bay export by 0.09 Gmol TA yr$^{-1}$ and the North Frisian export by
3 Gmol TA yr$^{-1}$.
Primary processes that contribute to the TA generation in the Wadden Sea are
denitrification, sulphate reduction, or processes that are coupled to sulphate reduction and
other processes (Thomas et al., 2009). In our model, the implemented benthic denitrification
does not generate TA (Seitzinger & Giblin, 1996), because modelled benthic denitrification
does not consume nitrate (Pätsch & Kühn, 2008). Benthic denitrification is coupled to
nitrification in the upper layer of the sediment (Raaphorst et al., 1990), giving reason for
neglecting TA generation by this process in the model. The modelled production of N$_2$ by
benthic denitrification falls in the range of 20 – 25 Gmol N yr$^{-1}$ in the validation area, which
would result in a TA production of about 19 – 23 Gmol TA yr$^{-1}$ (Brenner et al., 2016). In the
model nitrate uptake by phytoplankton produces about 40 Gmol TA yr$^{-1}$. Assuming large
parts of organic matter exported out of the validation area this production compensates the
missing TA generation by benthic denitrification. This amount of nitrate would not fully be
available for primary production if parts of it would be consumed by denitrification. Different

from this, the TA budget of Thomas et al. (2009) included estimates for the entire benthic denitrification as a TA generating process.

Sulphate reduction (not modelled here) also contributes to alkalinity generation. On longer time scales the net effect is vanishing as the major part of the reduced components are immediately re-oxidised in contact with oxygen. Iron- and sulphate- reduction generates TA but only their reaction product iron sulphide (essentially pyrite) conserves the reduced components from re-oxidation. As the formation of pyrite consumes TA, the TA contribution of iron reduction in the North Sea is assumed to be small and to balance that of pyrite formation (Brenner et al., 2016).

Atmospheric nitrogen deposition is taken into account in the simulations. Oxidised N-species ($NO_x$) dominate reduced species ($NH_y$) slightly in the validation area during 6 out of 9 simulation years. This implies that the deposition of dissolved inorganic nitrogen decreases TA in 6 of 9 years. The average decrease within 6 years is about 0.4 Gmol TA yr$^{-1}$, whereas the average increase within 3 years is only 0.1 Gmol TA yr$^{-1}$. Thomas et al. (2009) also assumed a dominance of oxidised species and consequently defined a negative contribution to the TA budget.

Dissolution of biogenic carbonates may be an efficient additional enhancement of the $CO_2$ buffer capacity (that is: source of TA), since most of the tidal flat surface sediments contain carbonate shell debris (Hild, 1997). Shallow oxidation of biogenic methane formed in deep or shallow tidal flat sediments (not modelled) (Höpner & Michaelis, 1994; Neira & Rackemann, 1996; Böttcher et al., 2007) has the potential both to lower or enhance the buffer capacity, thus counteracting or promoting the respective effect of carbonate dissolution. The impact of methane oxidation on the developing TA / DIC ratio in surface sediments, however, is complex and controlled by a number of superimposing biogeochemical processes (e.g., Akam et al., 2020).

The net effect of evaporation and precipitation in the Wadden Sea also has to be considered in budgeting TA. Although these processes are balanced in the North Sea (Schott, 1966), enhanced evaporation can occur in the Wadden Sea due to increased heating during low tide around noon. Onken & Riethmüller (2010) estimated an annual negative freshwater budget in the Hörnum Basin based on long-term hydrographic time series from observations

in a tidal channel. From this data a mean salinity difference between flood and ebb currents
of approximately -0.02 is calculated. This would result in an increase of TA by 1 μmol TA kg$^{-1}$,
which is within the range of the uncertainty of measurements. Furthermore, the enhanced
evaporation estimated from subtle salinity changes interferes with potential input of
submarine groundwater into the tidal basins, that been identified by Moore et al. (2011),
Winde et al. (2014), and Santos et al. (2015). The magnitude of this input is difficult to
estimate at present, for example from salinity differences between flood and ebb tides,
because the composition of SGD passing the sediment-water interfacial mixing zone has to
be known. Although first characteristics have been reported (Moore et al., 2011; Winde et
al., 2014; Santos et al., 2015), the quantitative effect of additional DIC, TA, and nutrient input
via both fresh and recirculated SGD into the Wadden Sea remains unclear.
An input of potential significance are small inlets that provide fresh water as well as DIC and
TA (Table 3). The current data base for seasonal dynamics of this source, however, is limited
and, therefore, this source cannot yet be considered quantitatively in budgeting approaches.

**4.2 TA / DIC ratios over the course of the year**

Ratios of TA and DIC generated in the tidal basins (Table 1) give some indication of the
dominant biogeochemical mineralisation and re-oxidation processes occurring in the
sediments of individual Wadden Sea sectors, although these processes have not been
explicitly modelled here (Chen & Wang, 1999; Zeebe & Wolf-Gladrow, 2001; Thomas et al.
2009; Sippo et al., 2016; Wurgaft et al., 2019; Akam et al., 2020). Candidate processes are
numerous and the export ratios certainly express various combinations, but the most
quantitatively relevant likely are aerobic degradation of organic material (resulting in a
reduction of TA due to nitrification of ammonia to nitrate with a TA / DIC ratio of -0.16),
denitrification (TA / DIC ratio of 0.8, see Rassmann et al., 2020), and anaerobic processes
related to sulphate reduction of organoclastic material (TA / DIC ratio of 1, see Sippo et al.,
2016). Other processes are aerobic (adding only DIC) and anaerobic (TA / DIC ratio of 2)
oxidation of upward diffusing methane, oxidation of sedimentary sulphides upon
resuspension into an aerated water column (no effect on TA / DIC) followed by oxidation of
iron (consuming TA), and nitrification of ammonium (consuming TA, TA / DIC ratio is -2, see
Pätsch et al., 2018 and Zhai et al. 2017).
The TA / DIC export ratios of DIC and TA for the individual tidal basins in three Wadden Sea
sectors (East Frisian, Jade Bay and North Frisian) as calculated from observed $\Delta$TA and $\Delta$DIC
over tidal cycles in different seasons are depicted in Fig. 9. They may give an indication of
regionally and seasonally varying processes occurring in the sediments of the three study
regions. The ratios vary between 0.2 and 0.5 in the North Frisian Wadden Sea with slightly
more TA than DIC generated in spring, summer and autumn, and winter having a negative
ratio of -0.5. The winter ratio coincides with very small measured differences of DIC in
imported and exported waters ($\Delta$DIC = -2 $\mu$mol kg$^{-1}$) and the negative TA / DIC ratio may thus
be spurious. The range of ratios in the other seasons is consistent with sulphate reduction
and denitrification as the dominant processes in the North Frisian tidal basins.
The TA / DIC ratios in the Jade Bay samples were consistently higher than those in the North
Frisian tidal basin and vary between 1 and 2 in spring and summer, suggesting a significant
contribution by organoclastic sulphate reduction and anaerobic oxidation of methane
(Al-Raei et al., 2009). The negative ratio of -0.4 in autumn is difficult to explain with
remineralisation or re-oxidation processes, but as with the fall ratio in Frisian tidal basin, it
coincides with a small change in $\Delta$DIC (-3 $\mu$mol kg$^{-1}$) at positive $\Delta$TA (8 $\mu$mol kg$^{-1}$). Taken at
face value, the resulting negative ratio of -0.4 implicates a re-oxidation of pyrite, normally at
timescales of early diagenesis thermodynamically stable (Hu and Cai, 2011), possibly
promoted by increasing wind forces and associated aeration and sulphide oxidation of
anoxic sediment layers (Kowalski et al., 2013). The DIC export rate from Jade Bay had its
minimum in autumn, consistent with a limited supply and mineralisation of organic matter,
possibly modified by seasonally changing impacts from small tidal inlets (Table 3).
The TA / DIC ratio of the East Frisian Wadden Sea is in the approximate range of those in
Jade Bay, but has one unusually high ratio in November caused by a significant increase in TA
of 14 $\mu$mol kg$^{-1}$ at a low increase of 5 $\mu$mol kg$^{-1}$ in DIC. Barring an analytical artefact, the
maximum ratio of 3 may reflect a short-term effect of iron reduction.
Based on these results, processes in the North Frisian Wadden Sea export area differ from
the East Frisian Wadden Sea and the Jade Bay areas. The DIC export rates suggest that
significant amounts of organic matter were degraded in North Frisian tidal basins, possibly
controlled by higher daily exchanged water masses in the North Frisian (8.1 km³ d⁻¹) than in
the East Frisian Wadden Sea (2.3 km³ d⁻¹) and in the Jade Bay (0.8 km³ d⁻¹) (compare
Table 2). However, TA export rates of the North Frisian and the East Frisian Wadden Sea
were in the same range.
Regional differences in organic matter mineralisation in the Wadden Sea have been
discussed by van Beusekom et al. (2012) and Kowalski et al. (2013) in the context of
connectivity with the open North Sea and influences of eutrophication and sedimentology.
They suggested that the organic matter turnover in the entire Wadden Sea is governed by
organic matter import from the North Sea, but that regionally different eutrophication
effects as well as sediment compositions modulate this general pattern. The reason for
regional differences may be related to the shape and size of the individual tidal basins. van
Beusekom et al. (2012) found that wider tidal basins with a large distance between barrier
islands and mainland, as is the case in the North Frisian Wadden Sea, generally have a lower
eutrophication status than narrower basins predominating in the East Frisian Wadden Sea.
Together with the high-water exchange rate the accumulation of organic matter is reduced
in the North Frisian Wadden Sea and the oxygen demand per volume is lower than in the
more narrow eutrophicated basins. Therefore, aerobic degradation of organic matter
dominated in the North Frisian Wadden Sea, where the distance between barrier islands and
mainland is large. This leads to less TA production (in relation to DIC production) than in the
East Frisian Wadden Sea, where anaerobic degradation of organic matter dominated in more
restricted tidal basins.

### 4.3. TA budgets and variability of TA inventory in the German Bight


Modelled TA and DIC in the German Bight have a high interannual and seasonal variability
(Fig. 8). The interannual variability of the model results are mainly driven by the physical
prescribed environment. Overall, the TA variability is more sensitive to Wadden Sea export
rates than DIC variability, because the latter is dominated by biological processes. However,
the inclusion of Wadden Sea DIC export rates improved correspondence with observed DIC
concentrations in the near-coastal North Sea.
It is a logical step to attribute the TA variability to variabilities of the different sources. In
order to calculate a realistic budget, scenario B was considered. Annual and seasonal
budgets of TA sources and sinks in this scenario are shown in Table 6. Note that $Riv_{eff}$ is not
taken into account for the budget calculations. This is explained in the Method Section 2.2.2
"River Input".
Comparing the absolute values of all sources and sinks of the mean year results in a relative
ranking of the processes. 41 % of all TA inventory changes in the validation area were due to
river loads, 37 % were due to net transport, 16 % were due to Wadden Sea export rates, 6 %
were due to internal processes. River input ranged from 78 to 152 Gmol TA $yr^{-1}$ and had the
highest absolute variability of all TA sources in the validation area. This is mostly due to the
high variability of annual freshwater discharge, which is indicated by low (negative) values of
$Riv_{eff}$. The latter values show that the riverine TA loads together with the freshwater flux
induce a small dilution of TA in the validation area for each year. Certainly, this ranking
depends mainly on the characteristics of the Elbe estuary. Due to high TA in rivers Rhine and
Meuse (Netherlands) they had an effective river input of +24 Gmol TA $yr^{-1}$ in 2008, which
constitutes a much greater impact on TA changes than the Elbe river. In a sensitivity test, we
switched off the TA loads of rivers Rhine and Meuse for the year 2008 and found that the
net flow of -71 Gmol TA $yr^{-1}$ decreased to -80 Gmol TA $yr^{-1}$, which indicates that water
entering the validation box from the western boundary is less TA-rich in the test case than in
the reference run.
At seasonal time scales (Table 6 lower part) the net transport dominated the variations from
October to March, while internal processes play a more important role from April to June
(28 %). The impact of effective river input was less than 5 % in every quarter. The Wadden
Sea TA export rates had an impact of 36 % on TA mass changes in the validation area from
July to September. Note that these percentages are related to the sum of the absolute
values of the budgeting terms.
Summing up the sources and sinks, Wadden Sea exchange rates, internal processes and
effective river loads resulted in highest sums in 2002 and 2003 (51 and 52 Gmol TA $yr^{-1}$) and
lowest in 2009 (44 Gmol TA $yr^{-1}$). For the consideration of TA variation we excluded net
transport and actual river loads, because these fluxes are diluted and do not necessarily
change the TA concentrations. In agreement with this, the highest TA was simulated in
summer 2003 (Fig. 8). The high interannual variability of summer concentrations was driven
essentially by hydrodynamic differences between the years. Flushing times and their
interannual variability were higher in summer than in winter (Fig. 6) of every year. High
flushing times or less strong circulation do have an accumulating effect on exported TA in
the validation area. To understand the reasons of the different flushing times monthly
stream patterns were analysed (Fig. 7). Distinct anticlockwise stream patterns defined the
hydrodynamic conditions in every winter. Summer stream patterns were in most years
weaker, especially in the German Bight (compare Fig. 7, June 2003). In August 2003 the
eastern part of the German Bight shows a clockwise rotation, which transports TA-enriched
water from July back to the Wadden-Sea area for further enrichment. This could explain the
highest concentrations in summer 2003.
Thomas et al. (2009) estimated that 73 Gmol TA yr$^{-1}$ were produced in the Wadden Sea.
Their calculations were based on measurements in 2001 and 2002. The presented model
was validated with data measured in August 2008 (Salt et al., 2013) at the same positions.
High TA in the German Bight was observed in summer 2001 and in summer 2008. Due to the
scarcity of data, the West Frisian Wadden Sea was not considered in the simulations, but, as
the western area is much larger than the eastern area, the amount of exported TA from that
area can be assumed to be in the same range as from the East Frisian Wadden Sea (10 to
14 Gmol TA yr$^{-1}$). With additional export from the West Frisian Wadden Sea, the maximum
overall Wadden Sea export may be as high as 53 Gmol TA yr$^{-1}$. Thus, the TA export from the
Wadden Sea calculated in this study is 20 to 34 Gmol TA yr$^{-1}$ lower than that assumed in the
study of Thomas et al. (2009). This is mainly due to the flushing time that was assumed by
Thomas et al. (2009). They considered the water masses to be flushed within six weeks
(Lenhart et al., 1995). Flushing times calculated in the present study were significantly longer
and more variable in summer. Since the Wadden Sea export calculated by Thomas et al.
(2009) was defined as a closing term for the TA budget, underestimated summerly flushing
times led to an overestimation of the exchange with the adjacent North Sea.
Table 4 shows that our scenario B underestimates observed TA by about 5.1 µmol kg$^{-1}$ in
2008. Scenario A has lower TA than scenario B in the validation area. The difference is about
11 µmol kg$^{-1}$. This means that the Wadden Sea export of 39 Gmol TA yr$^{-1}$ results in a
concentration difference of 11 µmol kg$^{-1}$. Assuming linearity, the deviation between
scenario B and the observations (5.1 µmol kg$^{-1}$) would be compensated by an additional
Wadden Sea export of about 18 Gmol TA yr$^{-1}$. If we assume that the deviation between
observation and scenario B is entirely due to uncertainties or errors in the Wadden Sea
export estimate, then the uncertainty of this export is 18 Gmol TA yr$^{-1}$.
Another problematic aspect in the TA export estimate by Thomas et al. (2009) is the fact that
their TA budget merges the sources of anaerobic TA generation from sediment and from the
Wadden Sea into a single source "anaerobic processes in the Wadden Sea". Burt et al. (2014)
found a sediment TA generation of 12 mmol TA m$^{-2}$ d$^{-1}$ at one station in the German Bight
based on Ra-measurements. This fits into the range of microbial gross sulphate reduction
rates reported by Al-Raei et al. (2009) in the back-barrier tidal areas of Spiekeroog island,
and by Brenner et al. (2016) at the Dutch coast. Within the latter paper, the different
sources of TA from the sediment were quantified. The largest term was benthic calcite
dissolution, which would be cancelled out in terms of TA generation assuming a steady-state
compensation by biogenic calcite production. Extrapolating the southern North Sea TA
generation (without calcite dissolution) from the data for one station of Brenner et al. (2016)
results in an annual TA production of 12.2 Gmol in the German Bight (Area = 28.415 km$^2$).
This is likely an upper limit of sediment TA generation, as the measurements were done in
summer when seasonal fluxes are maximal. This calculation reduces the annual Wadden Sea
TA generation estimated by Thomas et al. (2009) from 73 to 61 Gmol, which is still higher
than our present estimate. In spite of the unidentified additional TA-fluxes, both the
estimate by Thomas et al. (2009) and our present model-based quantification confirm the
importance of the Wadden-Sea export fluxes of TA on the North Sea carbonate system at
present and in the future.
*4.4 The impact of exported TA and DIC on the North Sea and influences on export*

713          *magnitude*

Observed high TA and DIC in the SE North Sea are mainly caused by TA and DIC export from
the Wadden Sea (Fig. 3-5). TA could be better reproduced than DIC in the model
experiments, which was mainly due to the higher sensitivity of DIC to modelled biology.
Nevertheless, from a present point of view the Wadden Sea is the main driver of TA in the
German Bight. Future forecast studies of the evolution of the carbonate system in the
German Bight will have to specifically focus on the Wadden Sea and on processes occurring
there. In this context the Wadden Sea evolution during future sea level rise is the most
important factor. The balance between sediment supply from the North Sea and sea level
rise is a general precondition for the persistence of the Wadden Sea (Flemming and Davis,
1994; van Koningsveld et al., 2008). An accelerating sea level rise could lead to a deficient
sediment supply from the North Sea and shift the balance at first in the largest tidal basins
and at last in the smallest basins. (CPSL, 2001; van Goor et al., 2003). The share of intertidal
flats as potential sedimentation areas is larger in smaller tidal basins (van Beusekom et al.,
2012), whereas larger basins have a larger share of subtidal areas. Thus, assuming an
accelerating sea level rise, large tidal basins will turn into lagoons, while tidal flats may still
exist in smaller tidal basins. This effect could decrease the overall Wadden Sea export rates
of TA, because sediments would no longer be exposed to the atmosphere and the products
of sulphate reduction would re-oxidise in the water column. Moreover, benthic-pelagic
exchange in the former intertidal flats would be more diffusive and less advective then today
due to a lowering of the hydraulic gradients during ebb tides, when parts of the sediment
become unsaturated with water. This would decrease TA export into the North Sea. Caused
by changes in hydrography and sea level the sedimentological composition may also change.
If sediments become more sandy, aerobic degradation of organic matter is likely to become
more important (de Beer et al., 2005). In fine grained silt diffusive transport plays a key role,
while in the upper layer of coarse (sandy) sediments advection is the dominant process.
Regionally, the North Frisian Wadden Sea will be more affected by rising sea level because
there the tidal basins are larger than the tidal basins in the East Frisian Wadden Sea and
even larger than the inner Jade Bay.
The Wadden Sea export of TA and DIC is driven by the turnover of organic material.
Decreasing anthropogenic eutrophication can lead to decreasing phytoplankton biomass and
production (Cadée & Hegeman, 2002; van Beusekom et al., 2009). Thus, the natural
variability of the North Sea primary production becomes more important in determining the
organic matter turnover in the Wadden Sea (McQuatters-Gollop et al., 2007; McQuatters-
Gollop & Vermaat, 2011). pH values in Dutch coastal waters decreased from 1990 to 2006
drastically. Changes in nutrient variability were identified as possible drivers (Provoost et al.,
2010), which is consistent with model simulations by Borges and Gypens (2010). Moreover,
despite the assumption of decreasing overall TA export rates from the Wadden Sea the
impact of the North Frisian Wadden Sea on the carbonate system of the German Bight could
potentially adjust to a change of tidal prisms and thus a modulation in imported organic
matter. If less organic matter is remineralised in the North Frisian Wadden Sea, less TA and
DIC will be exported into the North Sea.
In the context of climate change, processes that have impact on the freshwater budget of
tidal mud flats will gain in importance. Future climate change will have an impact in coastal
hydrology due to changes in ground water formation rates (Faneca Sànchez et al., 2012;
Sulzbacher et al., 2012), that may change both surface and subterranean run-off into the
North Sea. An increasing discharge of small rivers and groundwater into the Wadden Sea is
likely to increase DIC, TA, and possibly nutrient loads and may enhance the production of
organic matter. Evaporation could also increase due to increased warming and become a
more important process than today (Onken & Riethmüller, 2010), as will methane cycling
change due to nutrient changes, sea level and temperature rise (e.g., Höpner and Michaelis,
1994; Akam et al., 2020).
Concluding, in the course of climate change the North Frisian Wadden Sea will be affected
first by sea level rise, which will result in decreased TA and DIC export rates due to less
turnover of organic matter there. This could lead to a decreased buffering capacity in the
German Bight for atmospheric $CO_2$. Overall, less organic matter will be remineralised in the
Wadden Sea.

*5    Conclusion and Outlook*

We present a budget calculation of TA sources in the German Bight and relate 16 % of the
annual TA inventory changes to TA exports from the Wadden Sea. The impact of riverine
bulk TA seems to be less important due to the comparatively low TA levels in the Elbe
estuary, a finding that has to be proven by future research.
The evolution of the carbonate system in the German Bight under future changes depends
on the development of the Wadden Sea. The amount of TA and DIC that is exported from
the Wadden Sea depends on the amount of organic matter and / or nutrient that are
imported from the North Sea and finally remineralised in the Wadden Sea. Decreasing
riverine nutrient loads led to decreasing phytoplankton biomass and production (Cadée &
Hegeman, 2002; van Beusekom et al., 2009), a trend that is expected to continue in the
future (European Water Framework Directive). However, altered natural dynamics of
nutrient cycling and productivity can override the decreasing riverine nutrient loads (van
Beusekom et al., 2012), but these will not generate TA in the magnitude of denitrification of
river-borne nitrate.
Sea level rise in the North Frisian Wadden Sea will potentially be more affected by a loss of
intertidal areas than the East Frisian Wadden Sea (van Beusekom et al., 2012). This effect will
likely reduce the turnover of organic material in this region of the Wadden Sea, which may
decrease TA production and transfer into the southern North Sea.
Thomas et al. (2009) estimated that the Wadden Sea facilitates approximately 7 – 10% of the
annual $CO_2$ uptake of the North Sea. This is motivation for model studies on the future role
of the Wadden Sea in the $CO_2$ balance of the North Sea under regional climate change.
Future research will also have to address the composition and amount of submarine ground
water discharge, as well as the magnitude and seasonal dynamics in discharge and
composition of small water inlets at the coast, which are in this study only implicitly included
and in other studies mostly ignored due to a lacking data base.
**Data availability**
The river data are available at https://wiki.cen.uni-hamburg.de/ifm/ECOHAM/DATA_RIVER
and www.waterbase.nl. Meteorological data are stored at https://psl.noaa.gov/. The North
Sea TA and DIC data are stored at https://doi.org/10.1594/PANGAEA.438791 (2001),
https://doi.org/10.1594/PANGAEA.441686 (2005). The data of the North Sea cruise 2008
have not been published, yet, but can be requested via the CODIS data portal
(http://www.nioz.nl/portals-en; registration required). Additional Wadden Sea TA and DIC
data are deposited under doi:10.1594/PANGEA.841976.

**Author contributions**
The scientific concept for this study was originally developed by JP and MEB. FS wrote the
basic manuscript as part of his PhD thesis. VW provided field analytical data, as part of her
PhD thesis. JP developed the original text further with contributions from all co-authors.

**Competing interests**

The authors declare that they have no conflict of interest.

**Acknowledgements**

The authors appreciate the two constructive reviews, which greatly helped to improve the
manuscript, and the editorial handling by Jack Middelburg. I. Lorkowski, W. Kühn, and F.
Große are acknowledged for stimulating discussions, S. Grashorn for providing tidal prisms
and P. Escher for laboratory support. This work was financially supported by BMBF during
the Joint Research Project BIOACID (TP 5.1, 03F0608L and TP 3.4.1, 03F0608F), with further
support from Leibniz Institute for Baltic Sea Research. We also acknowledge the support by
the Cluster of Excellence 'CliSAP' (EXC177), University of Hamburg, funded by the German
Science Foundation (DFG) and the support by the German Academic Exchange service
(DAAD, MOPGA-GRI, #57429828) with funds of the German Federal Ministry of Education
and Research (BMBF). We used NCEP Reanalysis data provided by the NOAA/OAR/ESRL
PSL, Boulder, Colorado, USA, from their Web site at https://psl.noaa.gov/.



**Tables**
**Table 1: Mean TA and DIC concentrations [µmol l$^{-1}$] during rising and falling water levels**
**and the respective differences (Δ-values) that were used as wad_sta in (1). Areas are the**
**North Frisian (N), the East Frisian (E) Wadden Sea and the Jade Bay (J).**

| Area | Date | TA (rising) | TA (falling) | ΔTA | DIC (rising) | DIC (falling) | ΔDIC |
|------|------|-------------|--------------|-----|--------------|---------------|------|
| N | 29.04.2009 | 2343 | 2355 | 12 | 2082[*] | 2106 | 24 |
|   | 17.06.2009 | 2328 | 2332 | 4 | 2170 | 2190 | 20 |
|   | 26.08.2009 | 2238 | 2252 | 14 | 2077 | 2105 | 28 |
|   | 05.11.2009 | 2335 | 2333 | -2 | 2205 | 2209 | 4 |
| J | 20.01.2010 | 2429 | 2443 | 14 | 2380 | 2392 | 12 |
|   | 21.04.2010 | 2415 | 2448 | 33 | 2099 | 2132 | 33 |
|   | 26.07.2010 | 2424 | 2485 | 61 | 2159 | 2187 | 28 |
|   | 09.11.2010 | 2402 | 2399 | -3 | 2302 | 2310 | 8 |
| E | 03.03.2010 | 2379 | 2393 | 14 | 2313 | 2328 | 15 |
|   | 07.04.2010 | 2346 | 2342 | -4 | 2068 | 2082 | 14 |
|   | 17./18.05.2011 | 2445 | 2451 | 6 | 2209 | 2221 | 12 |
|   | 20.08.2002 | 2377 | 2414 | 37 | 2010 | 2030 | 20 |
|   | 01.11.2010 | 2423 | 2439 | 16 | 2293 | 2298 | 5 |

[*]: This value was estimated.

**Table 2: Daily Wadden Sea runoff to the North Sea at different export areas.**

| Position | wad_exc [10$^6$ m³ d$^{-1}$] |
|----------|------------------------------|
| N1 | 273 |
| N2 | 1225 |
| N3 | 1416 |
| N4 | 1128 |
| N5 | 4038 |
| N6 | 18 |
| J1 - J3 | 251 |
| E1 | 380 |
| E2 | 634 |
| E3 | 437 |
| E4 | 857 |




**Table 3: Examples for the carbonate system composition of small fresh water inlets draining into the Jade Bay and the backbarrier tidal area of Spiekeroog Island, given in (µmol kg$^{-1}$). Autumn results (A) (October 31$^{st}$, 2010) are taken from Winde et al. (2014); spring sampling (S) took place on May 20$^{th}$, 2011.**

| Site | Position | DIC(A) | TA(A) | DIC(S) | TA(S) |
|------|----------|--------|-------|--------|-------|
| Neuharlingersiel | 53°41.944 N 7°42.170 E | 2319 | 1773 | 1915 | 1878 |
| Harlesiel | 53°42.376 N 7°48.538 E | 3651 | 3183 | 1939 | 1983 |
| Wanger-/Horumersiel | 53°41.015 N 8°1.170 E | 5405 | 4880 | 6270 | 6602 |
| Hooksiel | 53°38.421 N 8°4.805 E | 2875 | 3105 | 3035 | 3302 |
| Maade | 53°33.534 N 8°7.082 E | 5047 | 4448 | 5960 | 6228 |
| Mariensiel | 53°30.895 N 8°2.873 E | 6455 | 5904 | 3665 | 3536 |
| Dangaster Siel | 53°26.737N 8°6.577 E | 1868 | 1246 | 1647 | 1498 |
| Wappelersiel | 53°23.414 N 8°12.437 E | 1373 | 630 | 1358 | 1152 |
| Schweiburger Siel | 53°24.725 N 8°16.968 E | 4397 | 3579 | 4656 | 4493 |
| Eckenwarder Siel | 53°31.249 N 8°16.527 E | 6542 | 6050 | 2119 | 4005 |





**Table 4: Averages (µmol kg$^{-1}$), standard deviations (µmol kg$^{-1}$), RMSE (µmol kg$^{-1}$), and**
**correlation coefficients r for the observed TA concentrations and the corresponding**
**scenarios A and B within the validation area.**

| TA | Average | Stdv | RMSE | r |
|---|---|---|---|---|
| Obs 2008 | 2333.52 | 32.51 | | |
| Obs 2005 | 2332.09 | 21.69 | | |
| Obs 2001 | 2333.83 | 33.19 | | |
| Sim A 2008 | 2327.64 | 6.84 | 27.97 | 0.77 |
| Sim A 2005 | 2322.16 | 5.21 | 22.05 | 0.45 |
| Sim A 2001 | 2329.79 | 5.32 | 31.89 | 0.24 |
| Sim B 2008 | 2338.60 | 22.09 | 18.34 | 0.86 |
| Sim B 2005 | 2339.48 | 26.81 | 31.81 | 0.18 |
| Sim B 2001 | 2342.96 | 17.28 | 30.07 | 0.47 |













**Table 5: Averages (µmol kg$^{-1}$), standard deviations (µmol kg$^{-1}$), RMSE (µmol kg$^{-1}$), and correlation coefficients r for the observed DIC concentrations and the corresponding scenarios A and B within the validation area.**


| DIC | Average | Stdv | RMSE | r |
|---|---|---|---|---|
| Obs 2008 | 2107.05 | 24.23 | | |
| Obs 2005 | 2098.20 | 33.42 | | |
| Obs 2001 | 2105.49 | 25.21 | | |
| Sim A 2008 | 2080.93 | 14.24 | 43.48 | -0.64 |
| Sim A 2005 | 2083.53 | 21.94 | 26.97 | 0.73 |
| Sim A 2001 | 2077.53 | 17.61 | 38.89 | 0.22 |
| Sim B 2008 | 2091.15 | 9.25 | 25.87 | 0.55 |
| Sim B 2005 | 2101.26 | 10.97 | 33.96 | 0.10 |
| Sim B 2001 | 2092.69 | 11.71 | 25.33 | 0.48 |











**Table 6: Annual TA budgets in the validation area of the years 2001 to 2009, annual**
**averages and seasonal budgets of January to March, April to June, July to September and**
**October to December [Gmol]. Net Flow is the annual net TA transport across the**
**boundaries of the validation area. Negative values indicate a net export from the**
**validation area to the adjacent North Sea. Δcontent indicates the difference of the TA**
**contents between the last and the first time steps of the simulated year or quarter.**

| | Wadden Sea export | internal processes | river loads | Riv$_{eff}$ | net flow | Δcontent |
|---|---|---|---|---|---|---|
| | Gmol/yr | Gmol/yr | Gmol/yr | Gmol/yr | Gmol/yr | Gmol |
| 2001 | 39 | 13 | 87 | -5 | 38 | 177 |
| 2002 | 39 | 19 | 152 | -7 | -223 | -13 |
| 2003 | 39 | 16 | 91 | -3 | -98 | 48 |
| 2004 | 39 | 13 | 78 | -5 | -8 | 122 |
| 2005 | 39 | 12 | 89 | -5 | -98 | 42 |
| 2006 | 39 | 12 | 88 | -4 | -56 | 83 |
| 2007 | 39 | 12 | 110 | -5 | -132 | 29 |
| 2008 | 39 | 14 | 93 | -5 | -71 | 75 |
| 2009 | 39 | 10 | 83 | -5 | -151 | -19 |
| Average | Gmol/yr | Gmol/yr | Gmol/yr | Gmol/yr | Gmol/yr | Gmol |
| | 39 | 14 | 101 | -5 | -89 | 65 |
| t = 3 mon | Gmol/t | Gmol/t | Gmol/t | Gmol/t | Gmol/t | Gmol |
| Jan - Mar | 7 | -1 | 38 | -1 | -49 | -5 |
| Apr - Jun | 10 | 15 | 23 | -2 | 6 | 54 |
| Jul - Sep | 17 | -2 | 15 | -2 | 13 | 43 |
| Oct - Dec | 4 | 1 | 25 | 0 | -56 | -26 |

## 6. Figure Captions

Figure 1: Upper panel: Map of the south-eastern North Sea and the bordering land. Lower panel: Model domains of ECOHAM (red) and FVCOM (blue), positions of rivers 1 – 16 (left, see Table 2) and the Wadden Sea export areas grid cells (right). The magenta edges identify the validation area, western and eastern part separated by the magenta dashed line.

Figure 2: Monthly Wadden Sea export of DIC and TA [Gmol mon$^{-1}$] at the North Frisian coast (N), East Frisian coast (E) and the Jade Bay in scenario B. The export rates were calculated for DIC and TA based on measured concentrations and simulated water fluxes.

Figure 3: Surface TA [µmol TA kg$^{-1}$] in August 2008 observed (a) and simulated with scenario A (b) and B (c). The black lines indicate the validation box.

Figure 4: Differences between TA surface summer observations and results from scenario A (a) and B (b) and the differences between DIC surface observations and results from scenario A (c) and B (d), all in µmol kg$^{-1}$. The black lines indicate the validation box.

Figure 5: Surface DIC concentrations [µmol DIC kg$^{-1}$] in August 2008 observed (a) and simulated with scenario A (b) and B (c). The black lines indicate the validation box.

Figure 6: Flushing times in the validation area in summer (June to August) and winter (January to March). The whole validation area is represented in blue, green is the western part of the validation area (4.5° E to 7° E) and red is the eastern part (east of 7° E).

Figure 7: Monthly mean simulated streamlines for summer months 2003 and 2008.

Figure 8: Simulated monthly mean TA (scenario A (a), scenario B (b)) [µmol TA kg$^{-1}$] and DIC (scenario A (c), scenario B (d)) [µmol DIC kg$^{-1}$] in the validation area for the years 2001-2009.

Figure 9: Temporally interpolated TA/DIC ratio of the export rates in the North Frisian, East Frisian, and Jade Bay. These ratios are calculated using the Δ-values of Table 1.

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

**8. Appendix**

**Table A1: Annual riverine freshwater discharge [km³ yr$^{-1}$]. The numbering refers to Fig. 1.**

| | 2001 | 2002 | 2003 | 2004 | 2005 | 2006 | 2007 | 2008 | 2009 |
|---|---|---|---|---|---|---|---|---|---|
| 1) Elbe | 23.05 | 43.38 | 23.95 | 19.56 | 25.56 | 26.98 | 26.61 | 24.62 | 24.28 |
| 2) Ems | 3.47 | 4.48 | 3.15 | 3.52 | 2.99 | 2.54 | 4.32 | 3.32 | 2.58 |
| 3) Noordzeekanaal | 3.21 | 2.98 | 2.49 | 3.05 | 3.03 | 2.96 | 1.55 | 3.05 | 2.46 |
| 4) Ijsselmeer (east) | 9.55 | 9.94 | 6.27 | 7.97 | 7.35 | 7.30 | 9.10 | 8.23 | 6.59 |
| 5) Ijsselmeer (west) | 9.55 | 9.94 | 6.27 | 7.97 | 7.35 | 7.30 | 9.10 | 8.23 | 6.59 |
| 6) Nieuwe Waterweg | 50.37 | 51.33 | 34.72 | 42.91 | 41.61 | 44.21 | 49.59 | 49.76 | 44.69 |
| 7) Haringvliet | 33.10 | 35.18 | 17.92 | 10.77 | 12.36 | 16.02 | 24.00 | 15.70 | 11.06 |
| 8) Scheldt | 7.28 | 2.74 | 4.31 | 3.64 | 3.59 | 3.74 | 4.63 | 4.57 | 3.63 |
| 9) Weser | 11.43 | 18.97 | 11.80 | 10.52 | 10.37 | 9.72 | 16.21 | 12.59 | 9.58 |
| 10) Firth of Forth | 2.72 | 3.76 | 2.06 | 3.01 | 3.00 | 2.84 | 2.85 | 3.59 | 3.66 |
| 11) Tyne | 1.81 | 2.25 | 1.18 | 2.04 | 1.92 | 1.78 | 2.09 | 2.70 | 2.05 |
| 12) Tees | 1.33 | 1.78 | 0.94 | 1.59 | 1.27 | 1.45 | 1.49 | 1.99 | 1.55 |
| 13) Humber | 10.76 | 12.10 | 7.16 | 10.51 | 7.68 | 11.11 | 12.03 | 13.87 | 9.60 |
| 14) Wash | 5.46 | 4.39 | 3.08 | 3.91 | 1.96 | 2.72 | 5.24 | 4.77 | 3.21 |
| 15) Thames | 4.47 | 3.23 | 2.41 | 2.13 | 0.96 | 1.57 | 3.52 | 3.20 | 2.38 |
| 16) Eider | 0.67 | 0.97 | 0.47 | 0.70 | 0.68 | 0.67 | 0.63 | 0.58 | 0.57 |
| Sum | 178.2 | 207.4 | 128.1 | 133.7 | 131.6 | 142.9 | 172.9 | 160.7 | 134.4 |










**Table A2: River numbers in Fig. 1, their positions and source of data**

| Number in Fig. 1 | Name | River mouth position | | Data source |
|---|---|---|---|---|
| 1 | Elbe | 53°53'20"N | 08°55'00"E | Pätsch & Lenhart (2008); TA-, DIC- and nitrate-concentrations by Amann (2015) |
| 2 | Ems | 53°29'20"N | 06°55'00"E | Pätsch & Lenhart (2008) |
| 3 | Noordzeekanaal | 52°17'20"N | 04°15'00"E | Pätsch & Lenhart (2008); TA-, DIC- and nitrate- |

| | | | | | | | | | | |
|---|---|---|---|---|---|---|---|---|---|---|
| | | | | | | | | concentrations from waterbase.nl | | |
| 4 | Ijsselmeer (east) | 53°17'20"N | 05°15'00"E | As above | | | | | | |
| 5 | Ijsselmeer (west) | 53°05'20"N | 04°55'00"E | As above | | | | | | |
| 6 | Nieuwe Waterweg | 52°05'20"N | 03°55'00"E | As above | | | | | | |
| 7 | Haringvliet | 51°53'20"N | 03°55'00"E | As above | | | | | | |
| 8 | Scheldt | 51°29'20"N | 03°15'00"E | As above | | | | | | |
| 9 | Weser | 53°53'20"N | 08°15'00"E | Pätsch & Lenhart (2008) | | | | | | |
| 10 | Firth of Forth | 56°05'20"N 02°45'00"W | | HASEC (2012) | | | | | | |
| 11 | Tyne | 55°05'20"N 01°25'00"W | | HASEC (2012) | | | | | | |
| 12 | Tees | 54°41'20"N 01°05'00"W | | HASEC (2012) | | | | | | |
| 13 | Humber | 53°41'20"N 00°25'00"W | | HASEC (2012) | | | | | | |
| 14 | Wash | 52°53'20"N | 00°15'00"E | HASEC (2012): sum of 4 rivers: Nene, Ouse, Welland and Witham | | | | | | |
| 15 | Thames | 51°29'20"N | 00°55'00"E | HASEC (2012) | | | | | | |
| 16 | Eider | 54°05'20"N | 08°55'00"E | Johannsen et al, 2008 | | | | | | |


**Table A3: Monthly values of TA, DIC and NO$_3$ concentrations [µmol kg$^{-1}$] of rivers, the annual mean and the standard deviation**

| River parameter | Jan | Feb | Mar | Apr | May | Jun | Jul | Aug | Sep | Oct | Nov | Dec | Mean | SD |
|---|---|---|---|---|---|---|---|---|---|---|---|---|---|---|
| Elbe TA | 2380 | 2272 | 2293 | 2083 | 2017 | 1967 | 1916 | 1768 | 1988 | 2156 | 2342 | 2488 | 2139 | 218 |
| Noordzeekanaal TA | 3762 | 3550 | 3524 | 3441 | 4748 | 3278 | 3419 | 3183 | 3027 | 3299 | 3210 | 3413 | 3488 | 441 |
| Nieuwe Waterweg TA | 2778 | 2708 | 2765 | 3006 | 2883 | 2658 | 2876 | 2695 | 2834 | 2761 | 2834 | 2927 | 2810 | 102 |
| Haringvliet TA | 2588 | 2635 | 2532 | 3666 | 2826 | 2829 | 2659 | 2660 | 2496 | 2816 | 2758 | 2585 | 2754 | 309 |
| Scheldt TA | 3781 | 3863 | 3708 | 3725 | 3758 | 3626 | 3722 | 3514 | 3367 | 3666 | 3825 | 3801 | 3696 | 140 |
| Ijsselmeer TA | 2829 | 3005 | 2472 | 2259 | 2611 | 1864 | 1672 | 1419 | 1445 | 2172 | 2286 | 2551 | 2215 | 521 |
| Elbe DIC | 2415 | 2319 | 2362 | 2179 | 2093 | 2025 | 1956 | 1853 | 2018 | 2200 | 2428 | 2512 | 2197 | 211 |
| Noordzeekanaal DIC | 3748 | 3579 | 3470 | 3334 | 3901 | 3252 | 3331 | 3136 | 2977 | 3214 | 3183 | 3405 | 3378 | 264 |
| Nieuwe Waterweg DIC | 2861 | 2794 | 2823 | 2991 | 2879 | 2657 | 2886 | 2706 | 2828 | 2773 | 2907 | 3036 | 2845 | 108 |
| Haringvliet DIC | 2673 | 2735 | 2600 | 3661 | 2850 | 2846 | 2687 | 2681 | 2512 | 2859 | 2803 | 2670 | 2798 | 292 |
| Scheldt DIC | 3798 | 3909 | 3829 | 3737 | 3704 | 3592 | 3705 | 3490 | 3316 | 3648 | 3733 | 3868 | 3694 | 167 |
| Ijsselmeer DIC | 2824 | 3008 | 2458 | 2234 | 2576 | 1826 | 1636 | 1369 | 1399 | 2134 | 2285 | 2565 | 2193 | 538 |
| Elbe NO$_3$ | 247 | 330 | 277 | 225 | 193 | 161 | 129 | 103 | 112 | 157 | 267 | 164 | 197 | 72 |
| Noordzeekanaal NO$_3$ | 150 | 168 | 190 | 118 | 79 | 71 | 64 | 73 | 78 | 92 | 107 | 137 | 111 | 42 |
| Nieuwe Waterweg NO$_3$ | 232 | 243 | 231 | 195 | 150 | 140 | 132 | 135 | 113 | 145 | 201 | 220 | 178 | 47 |
| Haringvliet NO$_3$ | 233 | 252 | 218 | 200 | 143 | 144 | 133 | 117 | 128 | 127 | 143 | 228 | 172 | 50 |
| Scheldt NO$_3$ | 320 | 341 | 347 | 345 | 243 | 221 | 219 | 215 | 189 | 202 | 190 | 274 | 259 | 63 |
| Ijsselmeer NO$_3$ | 136 | 159 | 190 | 192 | 135 | 46 | 20 | 14 | 7 | 18 | 20 | 79 | 85 | 73 |


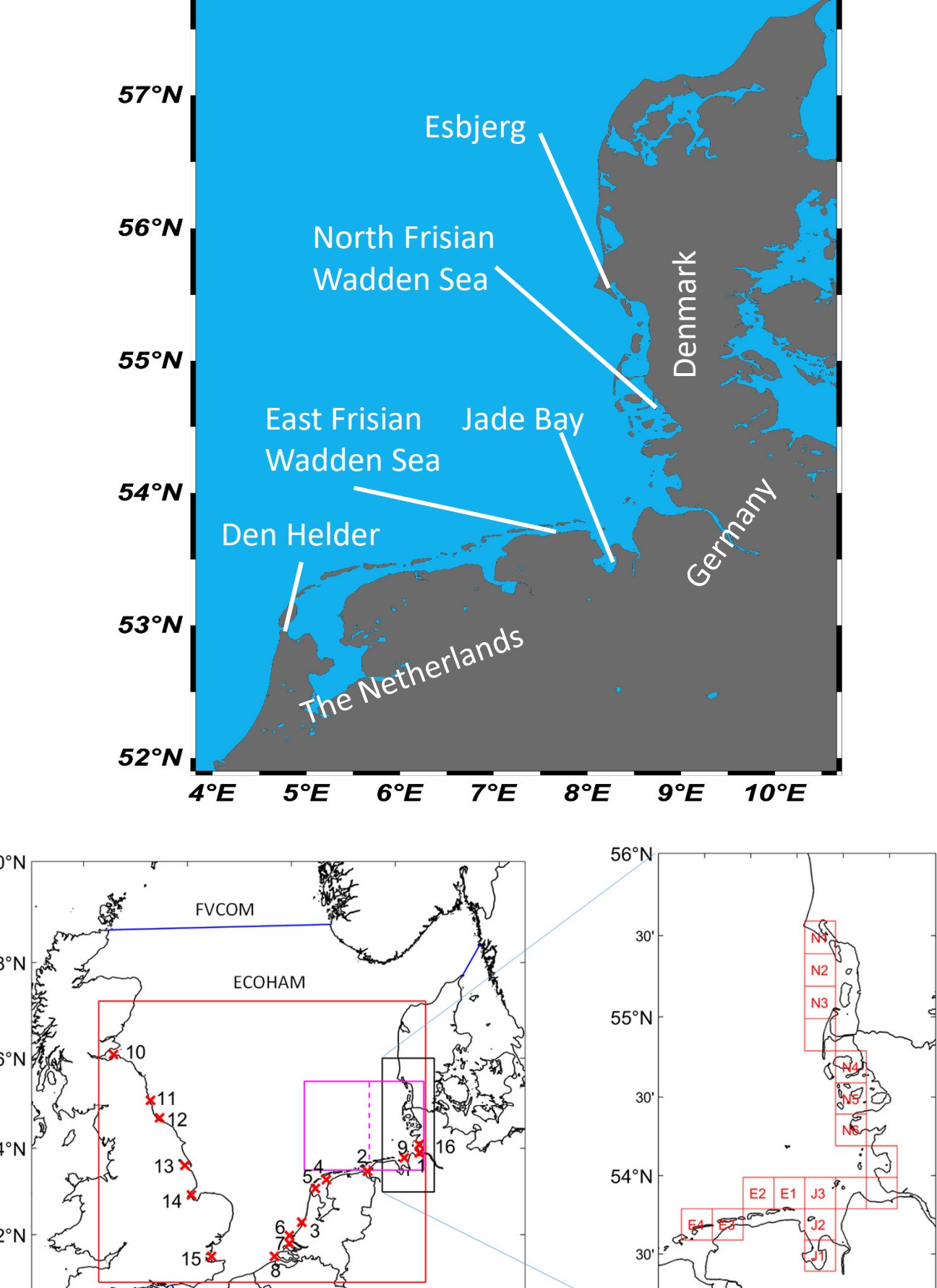

Fig. 1

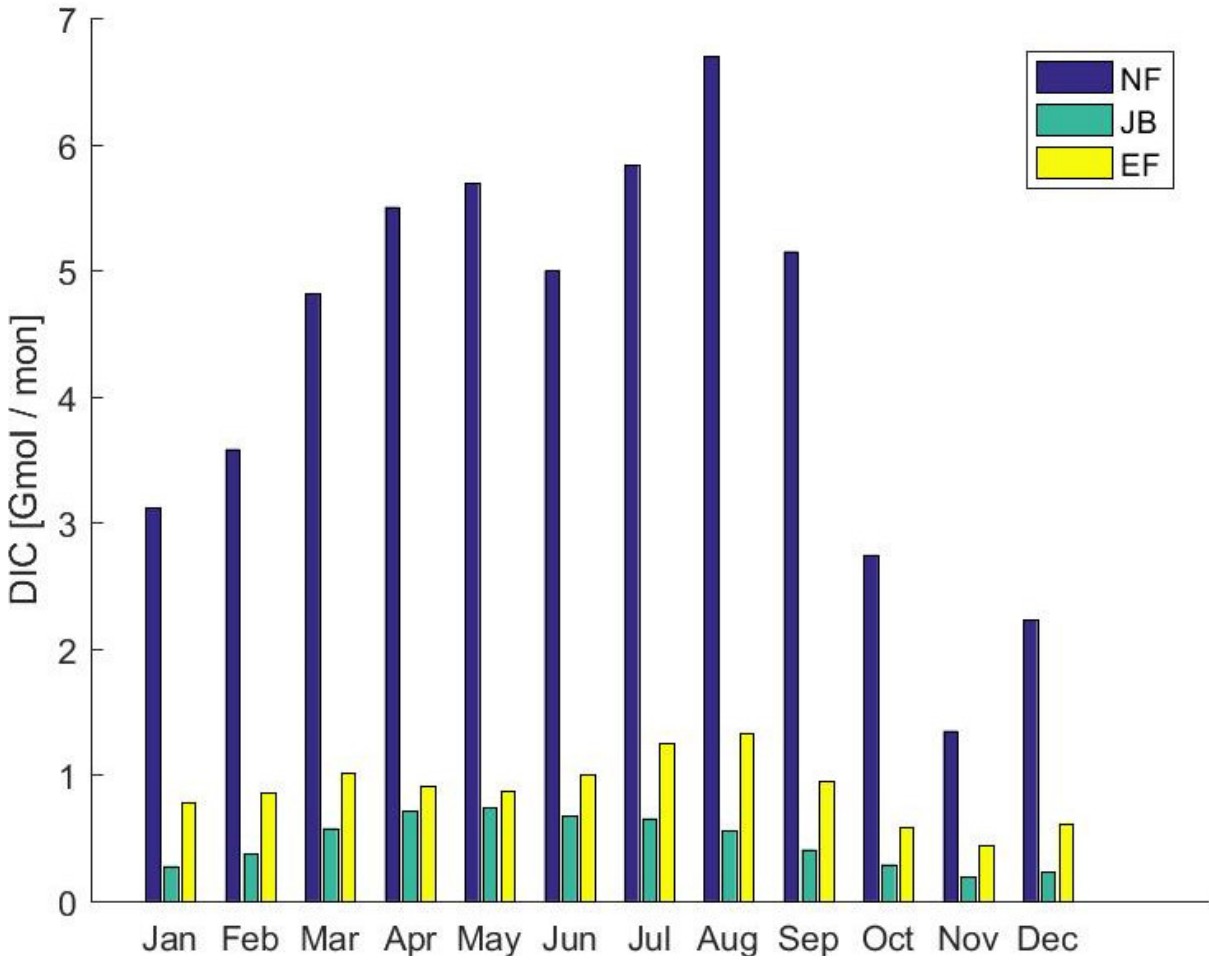

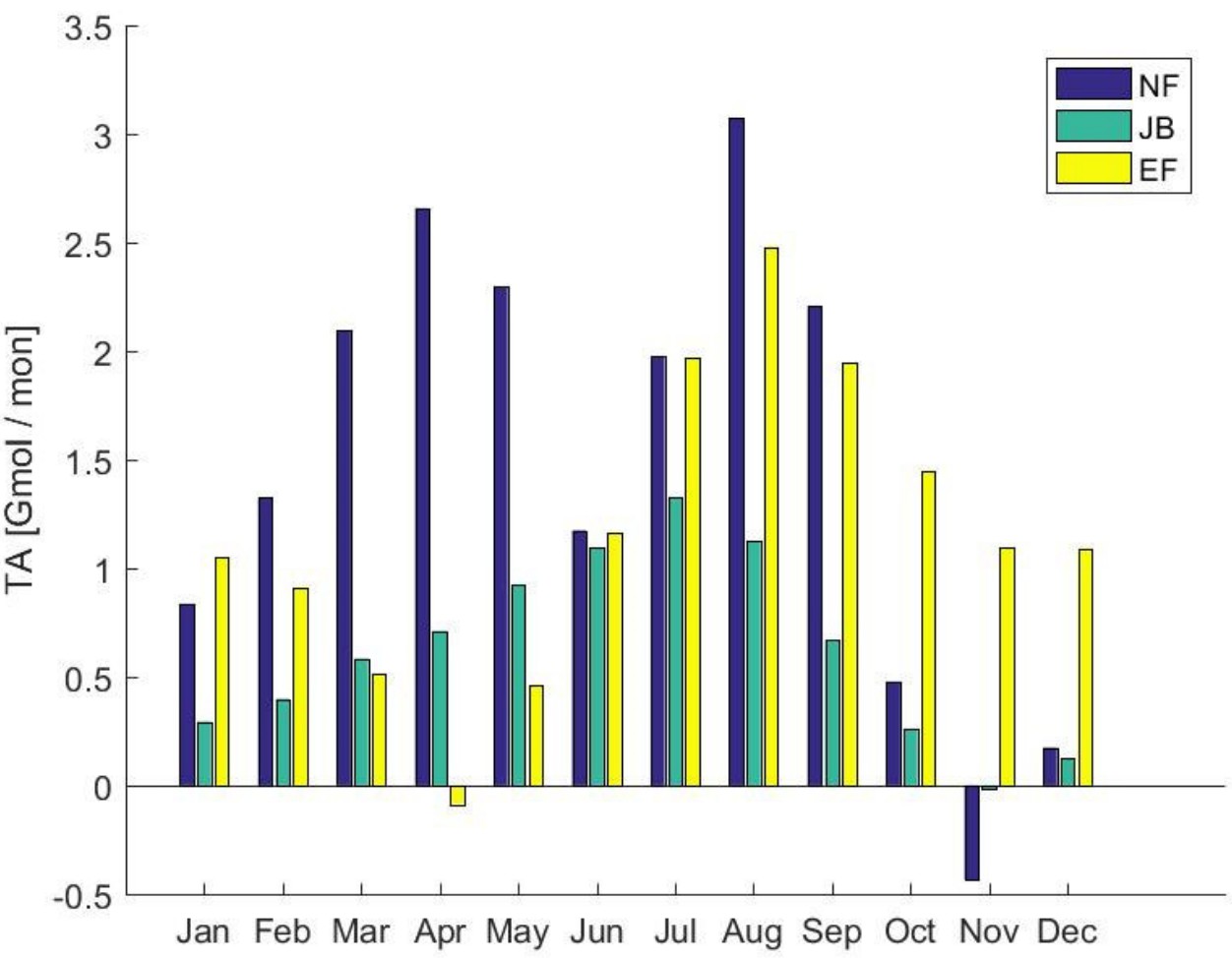

Fig. 2

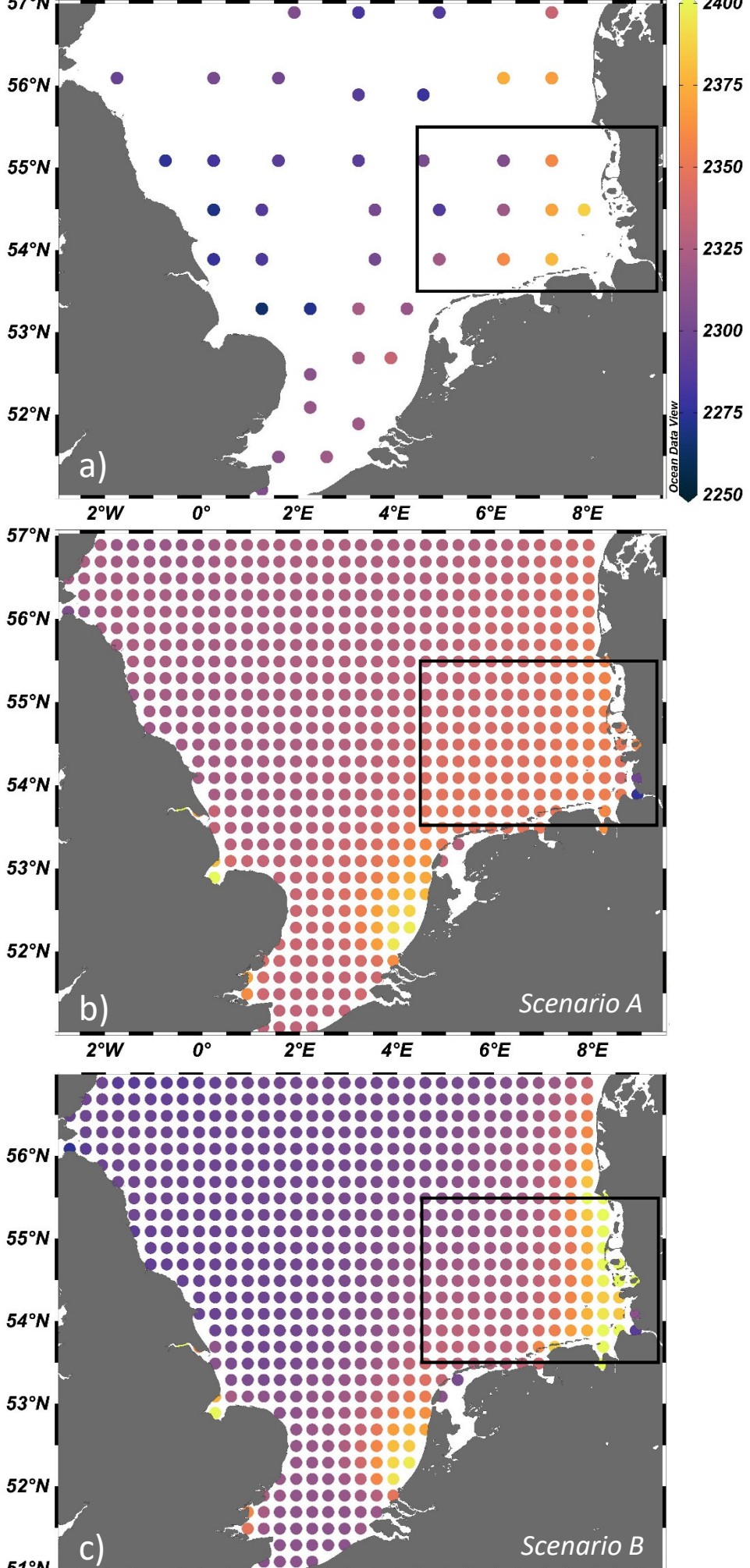

Fig. 3

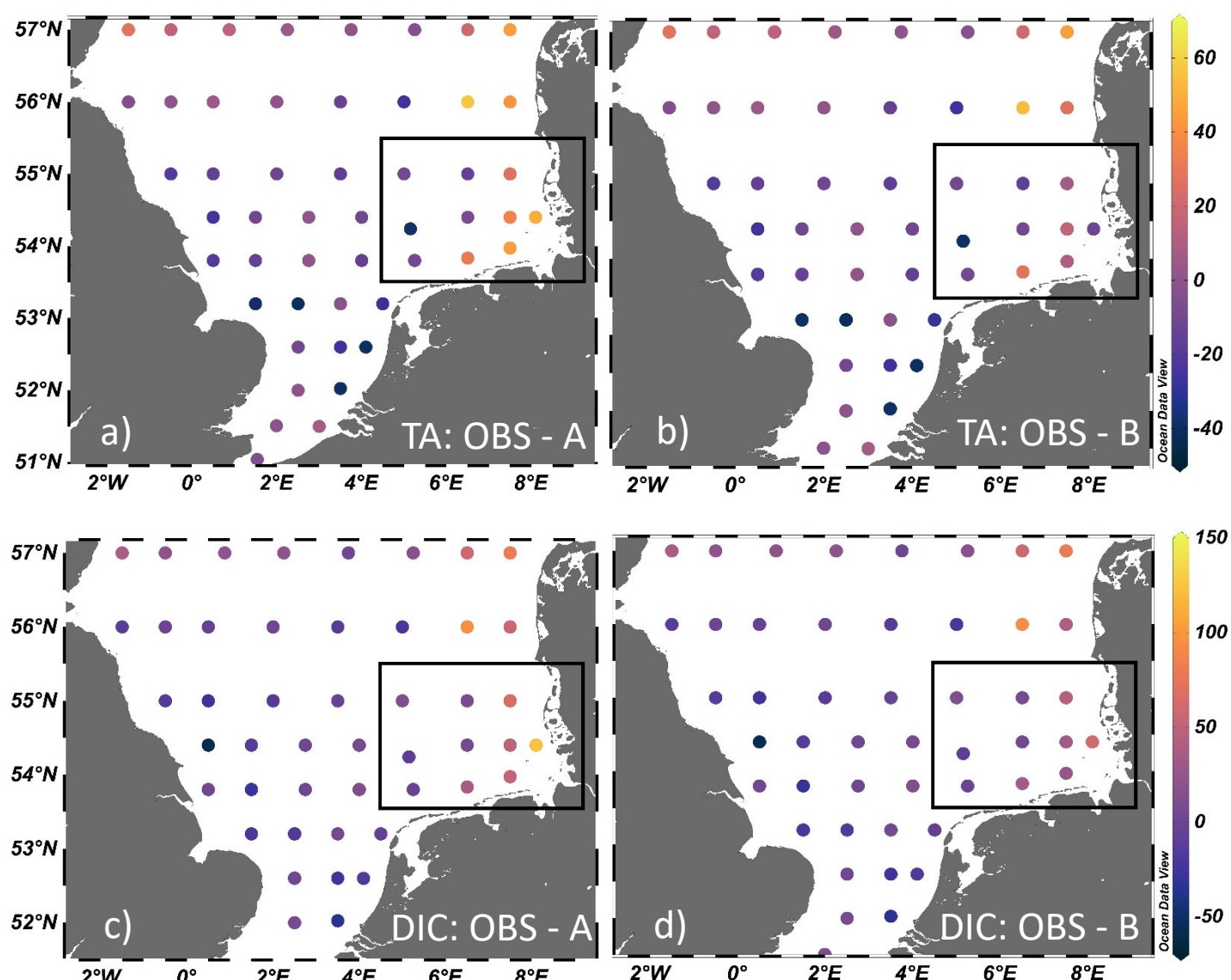

Fig. 4

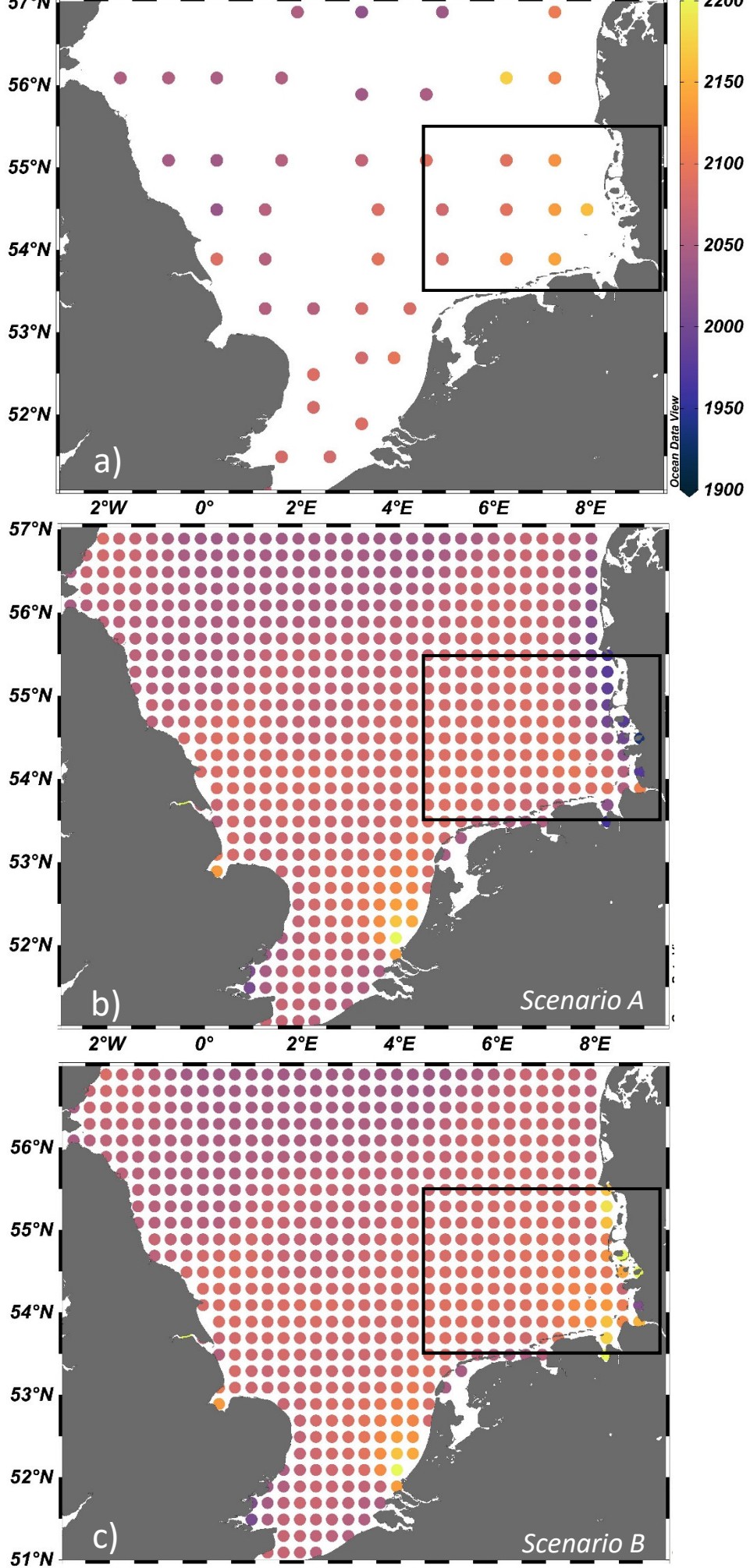

Fig. 5

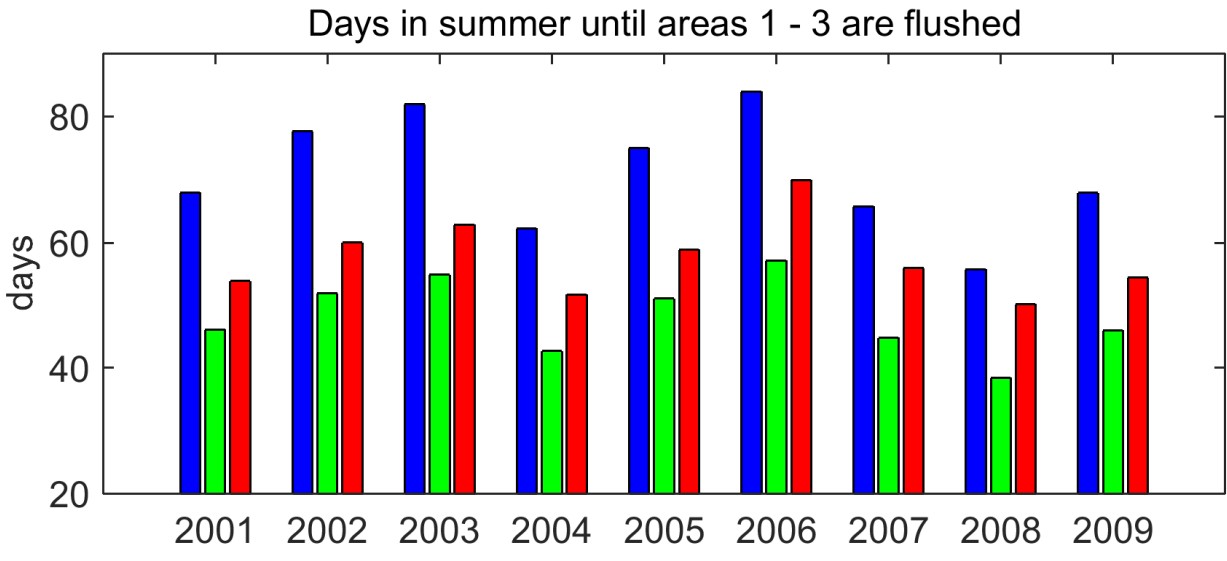

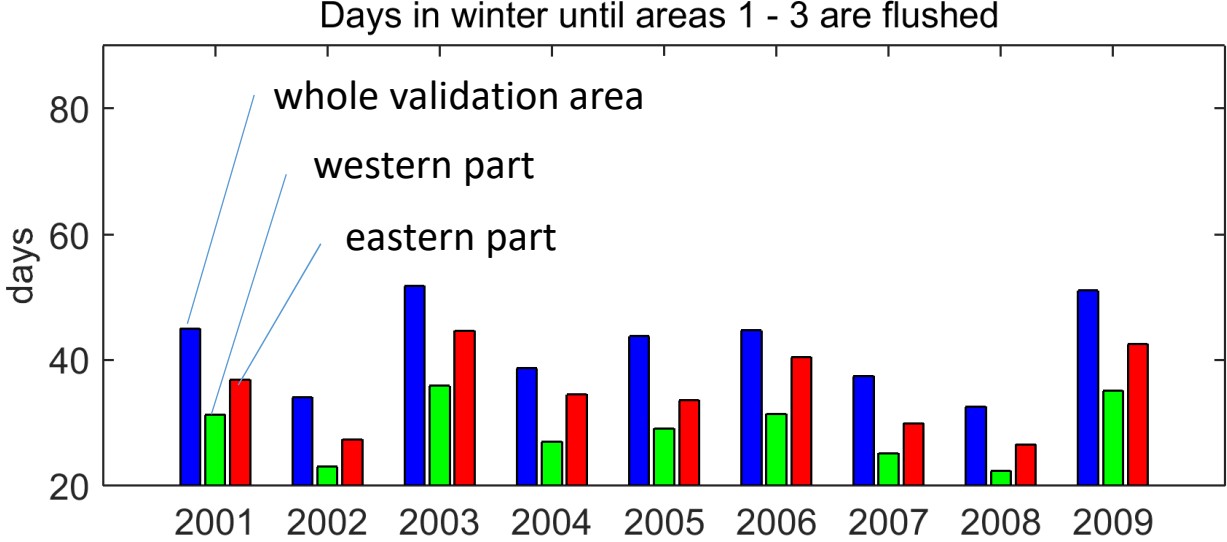

Fig. 6

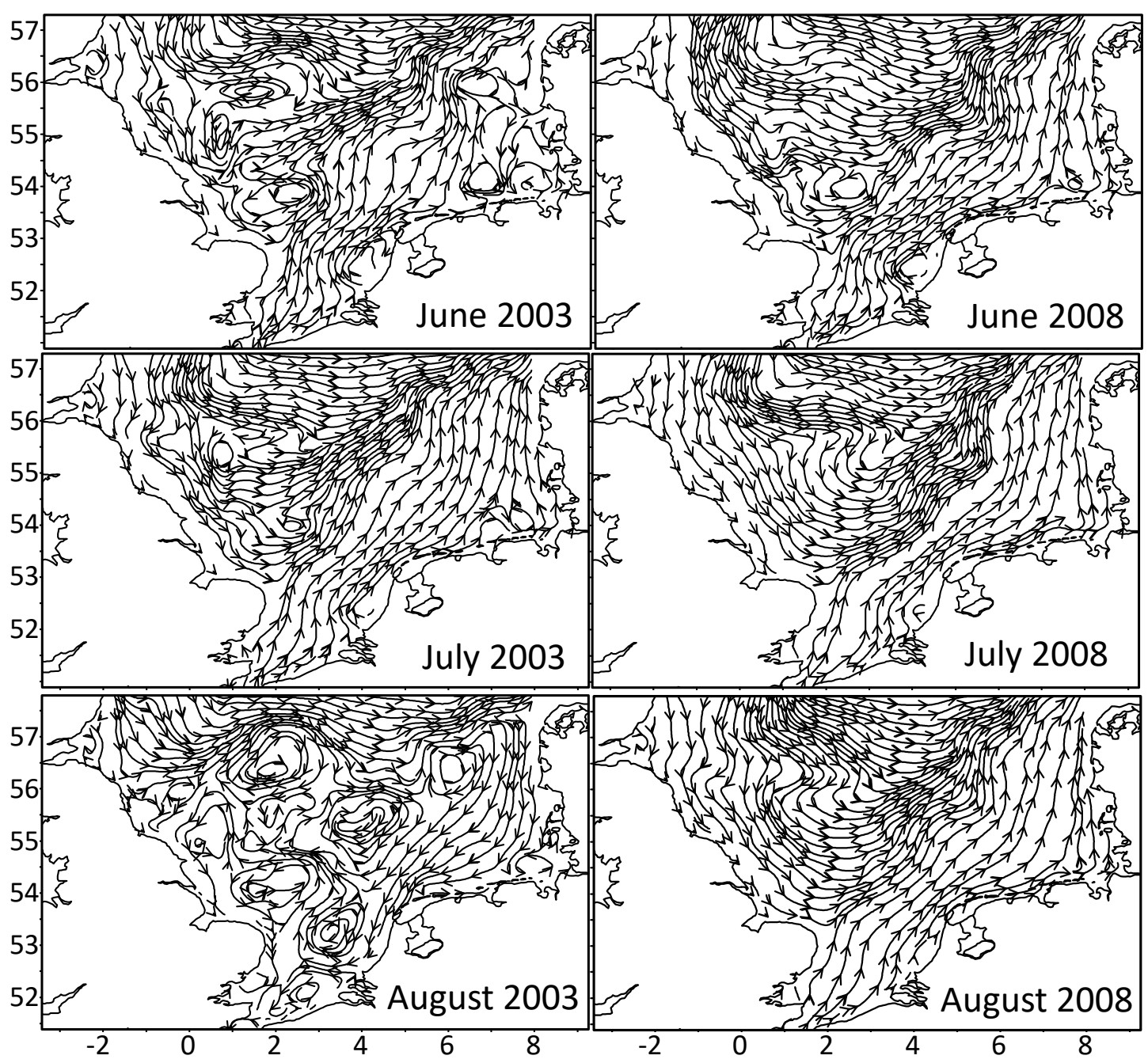

Fig. 7

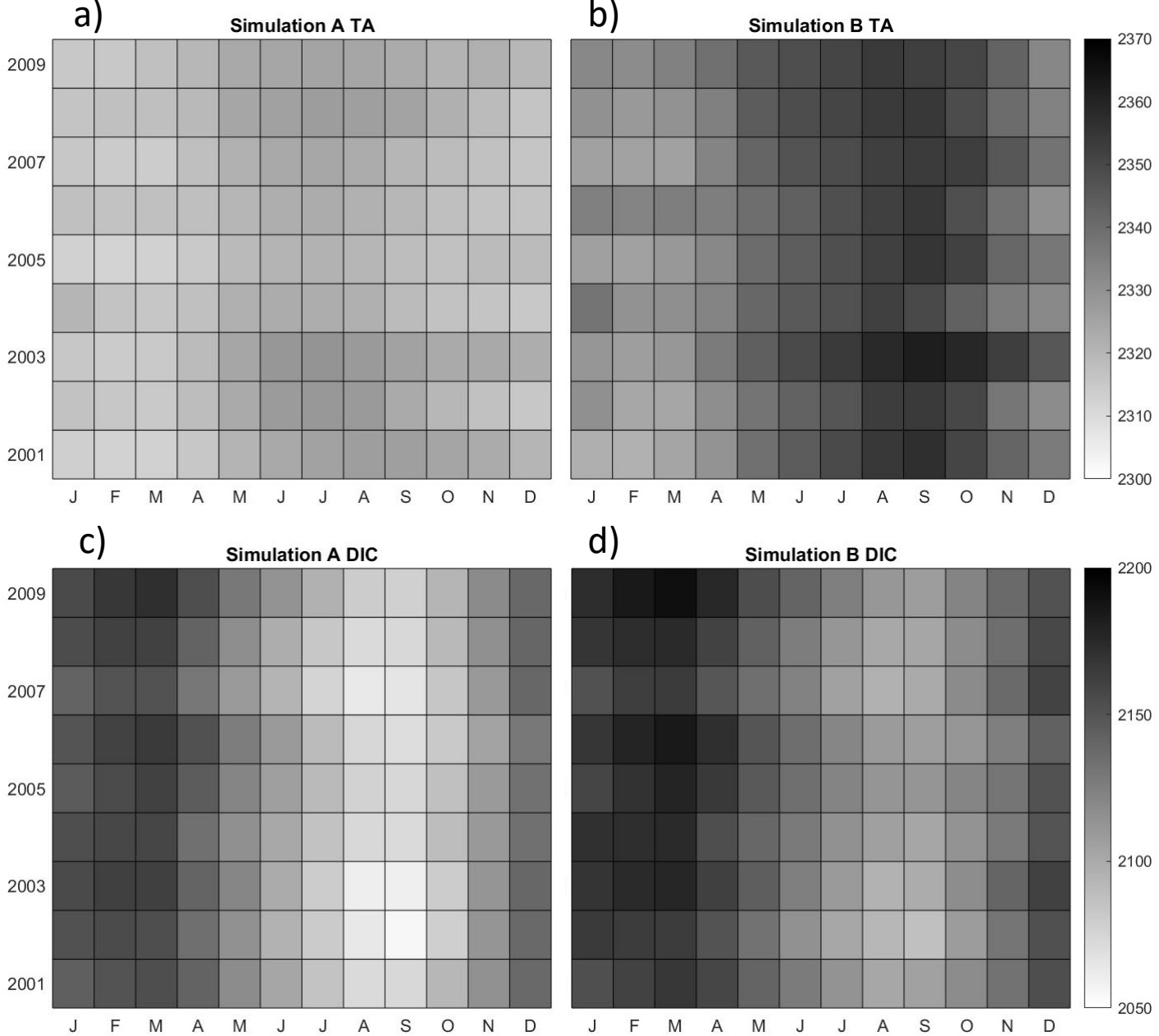

Fig. 8

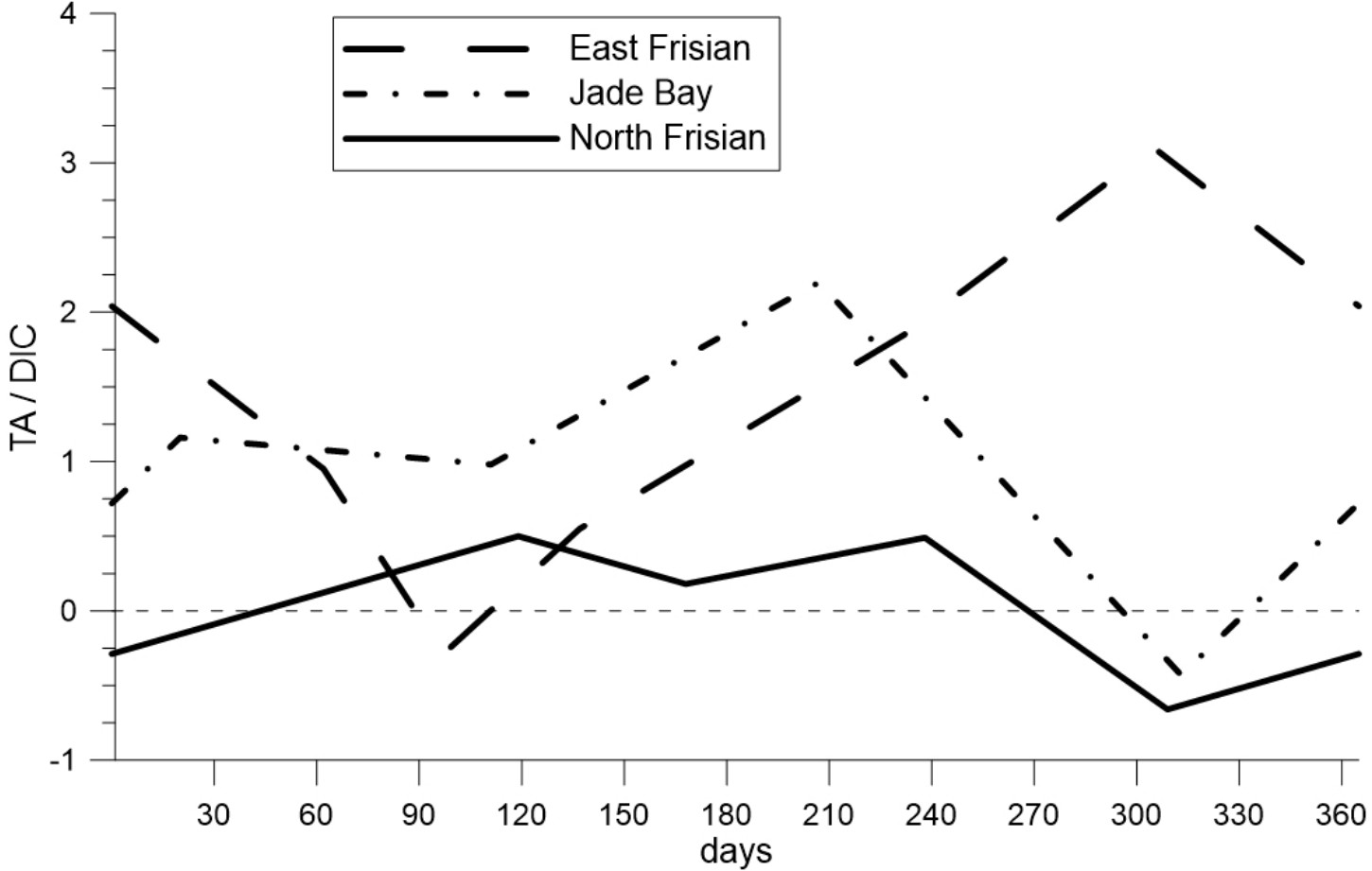

Fig. 9

# The impact of intertidal areas on the carbonate system of the southern North Sea

Fabian Schwichtenberg[1,6], Johannes Pätsch[1,5], Michael Ernst Böttcher[2,3,4], Helmuth Thomas[5], Vera Winde[2], Kay-Christian Emeis[5]

[1] Theoretical Oceanography, Universität Hamburg, Bundesstr. 53, D-20146 Hamburg, Germany

[2] Geochemistry & Isotope Biogeochemistry Group, Department of Marine Geology, Leibniz Institute of Baltic Sea Research (IOW), Seestr. 15, D-18119 Warnemünde, Germany

[3] Marine Geochemistry, University of Greifswald, Friedrich-Ludwig-Jahn Str. 17a, D-17489 Greifswald, Germany

[4] Department of Maritime Systems, Interdisciplinary Faculty, University of Rostock, Albert-Einstein-Str. 21, D-18059 Rostock, Germany

[5] Institute of Coastal Research, Helmholtz Zentrum Geesthacht (HZG), Max-Planck-Str. 1, D-21502 Geesthacht, Germany

[6] Present Address: German Federal Maritime and Hydrographic Agency, Bernhard-Nocht-Str. 78, D-20359 Hamburg, Germany

Correspondence to Johannes Pätsch (johannes.paetsch@uni-hamburg.de)

**Abstract**

The coastal ocean is strongly affected by ocean acidification because of its shallow water depths, low volume, and the closeness to terrestrial dynamics. Earlier observations of dissolved inorganic carbon (DIC) and total alkalinity (TA) in the southern part of the North Sea, a Northwest-European shelf sea, revealed lower acidification effects than expected. It has been assumed that anaerobic degradation and subsequent TA release in the adjacent back-barrier tidal areas ('Wadden Sea') in summer time is responsible for this phenomenon. In this study the exchange rates of TA and DIC between the Wadden Sea tidal basins and the North Sea and the consequences for the carbonate system in the German Bight are estimated using a 3-D ecosystem model. The aim of this study is to differentiate the various sources contributing to observed high summer TA ~~concentrations~~ in the southern North Sea.

Measured TA and DIC concentrations in the Wadden Sea are considered as model boundary
conditions. This procedure acknowledges the dynamic behaviour of the Wadden Sea as an
area of effective production and decomposition of organic material. According to the
modelling results, 39 Gmol TA yr$^{-1}$ were exported from the Wadden Sea into the North Sea,
which is less than a previous estimate, but within a comparable range. The interannual
variabilities of TA and DIC concentrations, mainly driven by hydrodynamic conditions, were
examined for the years 2001 – 2009. Dynamics in the carbonate system is found to be related
to specific weather conditions. The results suggest that the Wadden Sea is an important driver
for the carbonate system in the southern North Sea. On average 41 % of TA inventory changes
in the German Bight were caused by riverine input, 37 % by net transport from adjacent North
Sea sectors, 16 % by Wadden Sea export, and 6 % are caused by  internal net production of
TA. The dominant role of river input for the TA inventory disappears when focussing on TA
concentration changes due to the corresponding freshwater fluxes diluting the marine TA
concentrations. The ratio of exported TA versus DIC reflects the dominant underlying
biogeochemical processes in the Wadden Sea. Whereas, aerobic degradation of organic
matter plays a key role in the North Frisian Wadden Sea during all seasons of the year,
anaerobic degradation of organic matter dominated in the East Frisian Wadden Sea. Despite
of the scarcity of high-resolution field data it is shown that anaerobic degradation in the
Wadden Sea is one of the main contributors of elevated summer TA values in the southern
North Sea.
**1.  Introduction**
Shelf seas are highly productive areas constituting the interface between the inhabited coastal
areas and the global ocean. Although they represent only 7.6 % of the world ocean's area,
current estimates assume that they contribute approximately 21 % to total global ocean $CO_2$
sequestration (Borges, 2011). At the global scale the uncertainties of these estimates are
significant due to the lack of spatially and temporally resolved field data. Some studies
investigated regional carbon cycles in detail (e.g., Kempe & Pegler, 1991; Brasse et al., 1999;
Reimer et al., 1999; Thomas et al., 2004; 2009; Artioli et al., 2012; Lorkowski et al., 2012; Burt
et al., 2016; Shadwick et al., 2011; Laruelle et al., 2014; Carvalho et al., 2017) and pointed out
sources of uncertainties specifically for coastal settings.
However, natural pH dynamics in coastal- and shelf- regions, for example, have been shown
to be up to an order of magnitude higher than in the open ocean (Provoost et al, 2010).
Also, the nearshore effects of $CO_2$ uptake and acidification are difficult to determine, because
of the shallow water depth and a possible superposition by benthic-pelagic coupling, and
strong variations in fluxes of TA are associated with inflow of nutrients from rivers, pelagic
nutrient driven production and respiration (Provoost et al., 2010), submarine groundwater
discharge (SGD; Winde et al., 2014), and from benthic-pelagic pore water exchange (e.g.,
Billerbeck et al., 2006; Riedel et al., 2010; Moore et al., 2011; Winde et al., 2014; Santos et al.,
2012; 2015; Brenner et al., 2016; Burt et al., 2014; 2016; Seibert et al., 2019). Finally, shifts
within the carbonate system are driven by impacts from watershed processes and modulated
by changes in ecosystem structure and metabolism (Duarte et al., 2013).
Berner et al. (1970) and Ben-Yakoov (1973) were among the first who investigated elevated
TA and pH variations caused by microbial dissimilatory sulphate reduction in the anoxic pore
water of sediments. At the Californian coast, the observed enhanced TA export from
sediments was related to the burial of reduced sulphur compounds (pyrite) (Dollar et al., 1991;
Smith & Hollibaugh, 1993; Chambers et al., 1994). Other studies conducted in the Satilla and
Altamaha estuaries and the adjacent continental shelf found non-conservative mixing lines of
TA versus salinity, which was attributed to anaerobic TA production in nearshore sediments
(Wang & Cai, 2004; Cai et al., 2010). Iron dynamics and pyrite formation in the Baltic Sea were
found to impact benthic TA generation from the sediments (Gustafsson et al., 2019; Łukawska-
Matuszewska and Graca, 2017).
The focus of the present study is the southern part of the North Sea, located on the Northwest-
European Shelf. This shallow part of the North Sea is connected with the tidal basins of the
Wadden Sea via channels between barrier islands enabling an exchange of water, and
dissolved and suspended material (Rullkötter, 2009; Lettmann et al., 2009; Kohlmeier and
Ebenhöh, 2009). The Wadden Sea extends from Den Helder (The Netherlands) in the west to
Esbjerg (Denmark) in the north and covers an area of about 9500 km² (Ehlers, 1994). The
entire system is characterised by semidiurnal tides with a tidal range between 1.5 m in the
westernmost part and 4 m in the estuaries of the rivers Weser and Elbe (Streif, 1990). During
low tide about 50 % of the area are falling dry (van Beusekom et al., 2019). Large rivers
discharge nutrients into the Wadden Sea, which in turn shows a high degree of eutrophication,
aggravated by mineralisation of organic material imported into the Wadden Sea from the
open North Sea (van Beusekom et al., 2012).
In comparison to the central and northern part of the North Sea, TA ~~concentrations~~levels in
the southern part are significantly elevated during summer (Salt et al., 2013; Thomas et al.,
2009; Brenner et al., 2016; Burt et al., 2016). The observed high TA ~~concentrations~~levels have
been attributed to an impact from the adjacent tidal areas (Hoppema, 1990; Kempe & Pegler,
1991; Brasse et al., 1999; Reimer et al., 1999; Thomas et al., 2009; Winde et al., 2014), but this
impact has not been rigorously quantified. Using several assumptions, Thomas et al. (2009)
calculated an annual TA export from the Wadden Sea / Southern Bight of 73 Gmol TA yr$^{-1}$ to
close the TA budget for the southern North Sea.
The aim of this study is to reproduce the elevated summer ~~concentrations~~levels of TA in the
southern North Sea with a 3D biogeochemical model that has TA as prognostic variable. With
this tool at hand, we balance the budget TA in the relevant area on an annual basis.
Quantifying the different budget terms, like river input, Wadden Sea export, internal pelagic
and benthic production, degradation and respiration allows us to determine the most
important contributors to TA variations. In this way we refine the budget terms by Thomas et
al. (2009) and replace the original closing term by data. The new results are discussed on the
background of the budget approach proposed by Thomas et al. (2009).

## 2.  Methods
### 2.1. Model specifications

#### *2.1.1. Model domain and validation area*

The ECOHAM model domain for this study (Fig. 1) was first applied by Pätsch et al. (2010). For
model validations (magenta: validation area, Fig. 1), an area was chosen that includes the
German Bight as well as parts along the Danish and the Dutch coast. The western boundary of
the validation area is situated at 4.5° E. The southern and northern boundaries are at 53.5°
and 55.5° N, respectively. The validation area is divided by the magenta dashed line at 7° E
into the western and eastern part. For the calculation of box averages of DIC and TA a bias
towards the deeper areas with more volume and more data should be avoided. Therefore,
each water column covered with data within the validation area delivered one mean value,
which is calculated by vertical averaging. These mean water column averages were

horizontally interpolated onto the model grid. After this procedure average box values were calculated. In case of box-averaging model output, the same procedure was applied, but without horizontal interpolation.

### 2.1.2. The hydrodynamic module

The physical parameters temperature, salinity, horizontal and vertical advection as well as turbulent mixing were calculated by the submodule HAMSOM (Backhaus, 1985), which was integrated in the ECOHAM model. It is a baroclinic primitive equation model using the hydrostatic and Boussinesq approximation. It is applied to several regional sea areas worldwide (Mayer et al., 2018; Su & Pohlmann, 2009). Details are described by Backhaus & Hainbucher (1987) and Pohlmann (1996). The hydrodynamic model ran prior to the biogeochemical part. Daily result fields were stored for driving the biogeochemical model in offline mode. Surface elevation, temperature and salinity resulting from the Northwest-European Shelf model application (Lorkowski et al., 2012) were used as boundary conditions at the southern and northern boundaries. The temperature of the shelf run by Lorkowski et al. (2012) showed a constant offset compared with observations (their Fig. 3), because incoming solar radiation was calculated too high. For the present simulations the shelf run has been repeated with adequate solar radiation forcing.

River-induced horizontal transport due to the hydraulic gradient is incorporated (Große et al., 2017; Kerimoglu et al., 2018). This component of the hydrodynamic horizontal transport corresponds to the amount of freshwater discharge.

Within this study we use the term flushing time. It is the average time when a basin is filled with laterally advected water. The flushing time depends on the specific basin: large basins have usually higher flushing times than smaller basins. High flushing times correspond with low water renewal times.

### 2.1.3. The biogeochemical module

The relevant biogeochemical processes and their parameterisations have been detailed in Lorkowski et al. (2012). In former model setups TA was restored to prescribed values derived from observations (Thomas et al., 2009) with a relaxation time of two weeks (Kühn et al., 2010; Lorkowski et al., 2012). The changes in TA treatment for the study at hand is described below. Results from the Northwest-European Shelf model application (Lorkowski et al., 2012) were

used as boundary conditions for the recent biogeochemical simulations at the southern and
northern boundaries (Fig. 1).
The main model extension was the introduction of a prognostic treatment of TA in order to
study the impact of biogeochemical and physical driven changes of TA onto the carbonate
system and especially on acidification (Pätsch et al., 2018). The physical part contains
advective and mixing processes as well as dilution by riverine freshwater input. The pelagic
biogeochemical part is driven by planktonic production and respiration, formation and
dissolution of calcite, pelagic and benthic degradation and remineralisation, and also by
atmospheric deposition of reduced and oxidised nitrogen. All these processes impact TA. In
this model version benthic denitrification has no impact on pelagic TA ~~concentrations.~~. Other
benthic anaerobic processes are not considered. Only the carbonate ions from benthic calcite
dilution increase pelagic TA ~~concentrations.~~. Aerobic remineralisation releases ammonium
and phosphate, which enter the pelagic system across the benthic-pelagic interface and alter
the pelagic TA ~~concentration.~~. The theoretical background to this has been outlined by Wolf-
Gladrow et al. (2007).
The years 2001 to 2009 were simulated with 3 spin up years in 2000. Two different scenarios
(A and B) were conducted. Scenario A is the reference scenario without implementation of
any Wadden Sea processes. For scenario B we used the same model configuration as for
scenario A and additionally implemented Wadden Sea export rates of TA and DIC as described
in section 2.3.1. The respective Wadden Sea export rates (Fig. 2) are calculated by the
temporal integration of the product of wad_sta and wad_exc over one month (see section
2.3.1, equation 2).
**2.2. External sources and boundary conditions**
*2.2.1. Freshwater discharge*
Daily data of freshwater fluxes from 16 rivers were used (Fig. 1). For the German Bight and the
other continental rivers daily observations of runoff provided by Pätsch & Lenhart (2008) were
incorporated. The discharges of the rivers Elbe, Weser and Ems were increased by 21 %, 19 %
and 30 % in order to take additional drainage into account that originated from the area
downstream of the respective points of observation (Radach and Pätsch, 2007). The respective
tracer loads were increased accordingly. The data of Neal (2002) were implemented for the
British rivers for all years with daily values for freshwater. The annual amounts of freshwater
of the different rivers are shown in the appendix (Table A1). Riverine freshwater discharge
was also considered for the calculation of the concentrations of all biogeochemical tracers in
the model.
### 2.2.2. River input
**Data sources**
River load data for the main continental rivers were taken from the report by Pätsch & Lenhart
(2008) that was kept up to date continuously so that data for the years 2007 – 2009 were also
available (https://wiki.cen.uni-hamburg.de/ifm/ECOHAM/DATA_RIVER). They calculated
daily loads of nutrients and organic matter based on data provided by the different river
authorities. Additionally, loads of the River Eider were calculated according to Johannsen et
al. (2008).
Up to now, all ECOHAM applications used constant riverine DIC concentrations. TA was not
used. For the study at hand we introduced time varying riverine TA and DIC
~~concentrations.~~values. New data of freshwater discharge were introduced, as well as TA and
DIC loads for the British rivers (Neal, 2002). Monthly mean concentrations of nitrate, TA and
DIC were added for the Dutch rivers (www.waterbase.nl) and for the German river Elbe
(Amann et al., 2015). The Dutch river data were observed in the years 2007 – 2009. The river
Elbe data were taken in the years 2009 – 2011. These concentration data were prescribed for
all simulation years as mean annual cycle.
The data sources and positions of the river mouths of all 16 rivers are shown in Table A2 and
in Fig. 1. The respective riverine concentrations of TA and DIC are given in Table A3.
Schwichtenberg (2013) describes the river data in detail.
A few small flood gates ("Siel") and rivers transport fresh water from the recharge areas into
the intertidal areas (Streif, 1990). The recharge areas for these inlets differ considerably from
each other, leading to different relative contributions for the fresh water input. Whereas the
catchments of Schweiburger Siel (22.2 km$^2$) and the Hooksieler Binnentief are only of minor
importance, the Vareler Siel, the Eckenwarder Siel, and the Maade Siel are of medium
importance, and the highest contribution may originate from the Wangersiel, the Dangaster
Siel, and the Jade-Wapeler Siel (Lipinski, 1999).

**Effective river input**

In order to analyse the net effect on concentrations in the sea due to river input, the effective

river input ($Riv_{eff}$ [Gmol yr$^{-1}$]) is introduced:


$$Riv_{eff} = \frac{\Delta C|_{riv}}{\rho \cdot yr} \cdot V \cdot C \qquad (1)$$


with $\Delta C|_{riv}$ [µmol kg$^{-1}$]: the concentration change in the river mouth cell due to river load $riv$

and the freshwater flux from the river. $V$ [l] is the volume of the river mouth cell, $\rho$ [kg l$^{-1}$]

density of water, yr is one year, C [10$^{-15}$ l$^{-1}$] is a constant.

Bulk alkalinity discharged by rivers is quite large but most of the rivers entering the North Sea

(here the German Bight) have lower TA concentrations than the sea water. In case of identical

concentrations, the effective river load $Riv_{eff}$ is zero. The TA related molecules enter the sea,

and in most cases, they are leaving it via transport. In case of tracing or budgeting both the

real TA river discharge and the transport must be recognized. In order to understand TA

concentration changes in the sea $Riv_{eff}$ is appropriate.

227

### 2.2.3. Meteorological forcing

The meteorological forcing was provided by NCEP Reanalysis (Kalnay et al., 1996) and

interpolated on the model grid field. It consisted of six-hourly fields of air temperature,

relative humidity, cloud coverage, wind speed, atmospheric pressure, and wind stress for

every year. 2-hourly and daily mean short wave radiation were calculated from astronomic

insolation and cloudiness with an improved formula (Lorkowski et al., 2012).

**2.3. The Wadden Sea**

### 2.3.1. Implementation of Wadden Sea dynamics

For the present study the exchange of TA and DIC between North Sea and Wadden Sea was

implemented into the model by defining sinks and sources of TA and DIC for some of the

south-eastern cells of the North Sea grid (Fig. 1). The cells with adjacent Wadden Sea were

separated into three exchange areas: The East Frisian, the North Frisian Wadden Sea and the
Jade Bay, marked by "E", "N" and "J" (Fig. 1, right side).
Two parameters were determined in order to quantify the TA and DIC exchange between the
Wadden Sea and the North Sea.
1. Concentration changes of pelagic TA and DIC in the Wadden Sea during one tide, and
2. Water mass exchange between the back-barrier islands and the open sea during one

245        tide

Measured concentrations of TA and DIC (Winde, 2013; Winde et al., 2014) as well as modelled
water mass exchange rates of the export areas by Grashorn (2015) served as bases for the
calculated exchange. Details on flux calculations and measurements are described below. The
daily Wadden Sea exchange of TA and DIC was calculated as:

$$wad\_flu = \frac{wad\_sta * wad\_exc}{vol} \qquad (2)$$

Differences in measured concentrations in the Wadden Sea during rising and falling water
levels, as described in section 2.3.2, were temporally interpolated and summarized as *wad_sta*
[mmol m$^{-3}$]. Modelled daily Wadden Sea exchange rates of water masses (tidal prisms during
falling water level) were defined as *wad_exc* [m³ d$^{-1}$], and the volume of the corresponding
North Sea grid cell was *vol* [m³]. *wad_flu* [mmol m$^{-3}$ d$^{-1}$] were the daily concentration changes
of TA and DIC in the respective North Sea grid cells.
In fact, some amounts of the tidal prisms return without mixing with North Sea water, and
calculations of Wadden Sea – North Sea exchange should therefore consider flushing times in
the respective back-barrier areas. Since differences in measured concentrations between
rising and falling water levels were used, this effect is already assumed to be represented in
the data. This approach enabled the use of tidal prisms without consideration of any flushing
times.
**2.3.2. Wadden Sea - measurements**
The flux calculations for the Wadden Sea – North Sea exchange were carried out in tidal basins
of the East and North Frisian Wadden Sea (Spiekeroog Island, Sylt-Rømø) as well as in the Jade
Bay. For the present study seawater samples representing tidal cycles during different seasons
(Winde, 2013). The mean concentrations of TA and DIC during rising and falling water levels
and the respective differences (ΔTA and ΔDIC) are given in Table 1. Measurements in August
2002 were taken from Moore et al. (2011). The Δ-values were used as *wad_sta* and were
linearly interpolated between the times of observations for the simulations. In this procedure,
the linear progress of the Δ-values does not represent the natural behaviour perfectly,
especially if only few data are available. As a consequence, possible short events of high TA
and DIC export rates that occurred in periods outside the observation periods may have been
missed.
Due to the low number of concentration measurements a statistical analysis of uncertainties
of ΔTA and ΔDIC was not possible. They were measured with a lag of 2 hours after low tide
and high tide. This was done in order to obtain representative concentrations of rising and
falling water levels. As a consequence, only 2 - 3 measurements for each location and season
were considered for calculations of ΔTA and ΔDIC.

279         ### 2.3.3. Wadden Sea – modelling the exchange rates

Grashorn (2015) performed the hydrodynamic computations of exchanged water masses
(*wad_exc*) with the model FVCOM (Chen et al., 2003) by adding up the cumulative seaward
transport during falling water level (tidal prisms) between the back-barrier islands that were
located near the respective ECOHAM cells with adjacent Wadden Sea area. These values are
given in Table 2 for each ECOHAM cell in the respective export areas. The definition of the first
cell N1 and the last cell E4 is in accordance to the clockwise order in Fig. 1 (right side). The
mean daily runoff of all N-, J- and E-positions was 8.1 km³ d$^{-1}$, 0.8 km³ d$^{-1}$ and 2.3 km³ d$^{-1}$
respectively.

288         ### *2.3.4. Additional Sampling of DIC and TA*

DIC and TA ~~concentrations~~ for selected freshwater inlets sampled in October 2010 and May
2011 are presented in Table 3. Sampling and analyses took place as described by Winde et al.
(2014) and are here reported for completeness and input for discussion only. The autumn data
are deposited under doi:10.1594/PANGEA.841976. The samples for TA measurements were
filled without headspace into pre-cleaned 12 ccm Exetainer®, filled with 0.1 ml saturated HgCl$_2$
solution. The samples for DIC analysis were completely filled into 250 ccm ground-glass-
stoppered bottles, and then poisoned with 100 μl of a saturated HgCl$_2$ solution. The DIC
concentrations were determined at IOW by coulometric titration according to Johnson et al.
(1993), using reference material provided by A. Dickson (University of California, San Diego;
Dickson et al., 2003) for the calibration (batch 102). TA was measured by potentiometric
titration using HCl using a Schott titri plus equipped with an IOline electrode A157. Standard
deviations for DIC and TA measurements were better than +/-2 and +/-10 µmol kg$^{-1}$,
respectively.

### 2.4. Statistical analysis
A statistical overview of the simulation results in comparison to the observations (Salt et al.,
2013) is given in Table 4 and 5. In the validation area (magenta box in Fig. 1) observations of
10 different stations were available, each with four to six measurements at different depths
(51 measured points). Measured TA and DIC ~~concentrations~~ of each point were compared with
modelled TA and DIC ~~concentrations~~ in the respective grid cells, respectively. The standard
deviations (Stdv), the root means square errors (RMSE), and correlation coefficients (r) were
calculated for each simulation. In addition to the year 2008, which we focus on in this study,
observations were performed at the same positions in summer 2005 and 2001. These data are
also statistically compared with the model results.

## 3. Results

### 3.1. Model validation - TA ~~concentrations~~ in summer 2008
The results of scenarios A and B were compared with observations of TA in August 2008 (Salt
et al., 2013) for surface water. The observations revealed high TA ~~concentrations~~levels in the
German Bight (east of 7° E and south of 55° N) and around the Danish coast (around 56° N) as
shown in Fig. 3a. The observed concentrations in these areas ranged between 2350 and
2387 µmol TA kg$^{-1}$. These findings were in accordance with observed TA ~~concentrations~~ in
August / September 2001 (Thomas et al., 2009). TA ~~concentrations~~ in other parts of the
observation domain ranged between 2270 µmol TA kg$^{-1}$ near the British coast (53° N – 56° N)
and 2330 µmol TA kg$^{-1}$ near the Dutch coast and the Channel. In the validation box the overall
average and the standard deviation of all observed TA concentrations (Stdv) was 2334 and
33 µmol TA kg$^{-1}$, respectively.
In scenario A the simulated surface TA ~~concentrations~~ showed a more homogeneous pattern
than observations with maximum values of 2396 µmol TA kg$^{-1}$ at the western part of the Dutch
coast and even higher (2450 µmol TA kg$^{-1}$) in the river mouth of the Wash estuary at the British
coast. Minimum values of 2235 and 2274 µmol TA kg$^{-1}$ were simulated at the mouths of the
rivers Elbe and Firth of Forth. The modelled TA ~~concentration~~ ranged from 2332 to
2351 µmol TA kg$^{-1}$ in the German Bight and in the Jade Bay. Strongest underestimations in
relation to observations are located in a band close to the coast stretching from the East
Frisian Islands to 57° N at the Danish coast (Fig. 4a). The deviation of simulation results of
scenario A from observations in the validation box was represented by a RMSE of
28 µmol TA kg$^{-1}$. The standard deviation was 7 µmol TA kg$^{-1}$ and the correlation amounted to
r = 0.77 (Table 4). In the years 2005 and 2001 similar statistical values are found, but the
correlation coefficient was smaller.
The scenario B was based on a Wadden Sea export of TA and DIC as described above. The
major difference in TA ~~concentrations~~ of this scenario compared to A occurred east of 6.5° E.
Surface TA ~~concentrations~~ there peaked in the Jade Bay (2769 µmol TA kg$^{-1}$) and were
elevated off the North Frisian and Danish coasts from 54.2° to 56° N (> 2400 µmol TA kg$^{-1}$).
Strongest underestimations in relation to observations are noted off the Danish coast
between 56° and 57° N (Fig. 4b). In the German Bight the model overestimated the
observations slightly, while at the East Frisian Islands the model underestimates TA. When
approaching the Dutch Frisian Islands the simulation overestimates TA compared to
observations and strongest overestimations can be seen near the river mouth of River Rhine.
Compared to scenario A the simulation of scenario B was closer to the observations in terms
of RMSE (18 µmol TA kg$^{-1}$) and the standard deviation (Stdv = 22 µmol TA kg$^{-1}$). Also, the
correlation (r = 0.86) improved (Table 4). In the years 2001 and 2005 the observed mean
values are slightly overestimated by the model. The statistical values for 2001 are better than
for 2005, where scenario A better compares with the observations.

### 3.2. Model validation - DIC concentrations in summer 2008

Analogously to TA the simulation results were compared with surface observations of DIC
~~concentrations~~ in summer 2008 (Salt et al., 2013). They also revealed high values in the
German Bight (east of 7° E and south of 55° N) and around the Danish coast (near 56° N) which
is shown in Fig. 5. The observed DIC concentrations in these areas ranged between 2110 and
2173 µmol DIC kg$^{-1}$. Observed DIC concentrations in other parts of the model domain ranged
between 2030 and 2070 µmol DIC kg$^{-1}$ in the north western part and 2080 - 2117 µmol DIC kg$^{-}$
$^1$ at the Dutch coast. In the validation box the overall average and the standard deviation of
all observed DIC concentrations were 2108 and 25.09 µmol DIC kg$^{-1}$, respectively.
The DIC concentrations in scenario A ranged between 1935 and 1977 µmol DIC kg$^{-1}$ at the
North Frisian and Danish coast (54.5° N - 55.5° N) and 1965 µmol DIC kg$^{-1}$ in the Jade Bay.
Maxima of up to 2164 µmol DIC kg$^{-1}$ were modelled at the western part of the Dutch coast
north of the mouth of River Rhine (Fig. 5). The DIC concentrations in the German Bight showed
a heterogeneous pattern in the model, and sometimes values decreased from west to east,
which contrasts the observations (Fig. 5a). This may be the reason for the negative correlation
coefficient r = -0.64 between model and observations (Table 5). The significant deviation from
observation of results from scenario A is also indicated by the RMSE of 43 µmol DIC kg$^{-1}$, and
a standard deviation of 14 µmol DIC kg$^{-1}$. In 2001 and 2005 the simulation results of this
scenario A are better, which is expressed in positive correlation coefficients and small RMSE
values.
In scenario B the surface DIC concentrations at the Wadden Sea coasts increased: The North
Frisian coast shows concentrations of up to 2200 µmol DIC kg$^{-1}$ while the German Bight has
values of 2100 − 2160 µmol DIC kg$^{-1}$, and Jade Bay concentrations were higher than
2250 µmol DIC kg$^{-1}$. The other areas are comparable to scenario A. In scenario B the RMSE in
the validation box decreased to 26 µmol DIC kg$^{-1}$ in comparison to scenario A. The standard
deviation decreased to 9.1 µmol DIC kg$^{-1}$, and the correlation improved to r = 0.55 (Table 5).
The average values are close to the observed ones for all years, even though in 2005 a large
RMSE was found.
The comparison between observations and simulation results of scenario A (Fig. 4c) clearly
show model underestimations in the south-eastern area and are strongest in the inner
German Bight towards the North Frisian coast (> 120 µmol DIC kg$^{-1}$). Scenario B also models
values lower than observations in the south-eastern area (Fig. 4d), but the agreement
between observation and model results is reasonable. Only off the Danish coast near 6.5° E,
56° N the model underestimates DIC by 93 µmol DIC kg$^{-1}$.

### 3.3. Hydrodynamic conditions and flushing times

The calculations of Wadden Sea TA export in Thomas et al. (2009) were based on several assumptions concerning riverine input of bulk TA and nitrate, atmospheric deposition of NOx, water column inventories of nitrate and the exchange between the Southern Bight and the adjacent North Sea (Lenhart et al., 1995). The latter was computed by considering that the water in the Southern Bight is flushed with water of the adjacent open North Sea at time scales of six weeks. For the study at hand, flushing times in the validation area in summer and winter are presented for the years 2001 to 2009 in Fig. 6. Additionally, monthly mean flow patterns of the model area are presented for June, July and August for the years 2003 and 2008, respectively (Fig. 7). They were chosen to highlight the pattern in summer 2003 with one of the highest flushing times (lowest water renewal times), and that in 2008 corresponding to one of the lowest flushing times (highest water renewal times).

The flushing times were determined for the three areas 1 – validation area, 2 – western part of the validation area, 3 – eastern part of the validation area. They were calculated by dividing the total volume of the respective areas 1 – 3 by the total inflow into the areas $m^3 (m^3 s^{-1})^{-1}$. Flushing times (rounded to integer values) were consistently higher in summer than in winter, meaning that highest inflow occurred in winter. Summer flushing times in the whole validation area ranged from 54 days in 2008 to 81 days in 2003 and 2006, whereas the winter values in the same area ranged from 32 days in 2008 to 51 days in 2003 and 2009. The flushing times in the western and eastern part of the validation area were smaller due to the smaller box sizes. Due to the position, flushing times in the western part were consistently shorter than in the eastern part. These differences ranged from 5 days in winter 2002 to 14 days in summer 2006 and 2008. The interannual variabilities of all areas were higher in summer than in winter.

The North Sea is mainly characterised by an anti-clockwise circulation pattern (Otto et al., 1990; Pätsch et al., 2017). This can be observed for the summer months in 2008 (Fig. 7). More disturbed circulation patterns in the south-eastern part of the model domain occurred in June 2003: In the German Bight and in the adjacent western area two gyres with reversed rotating direction are dominant. In August 2003 the complete eastern part shows a clockwise rotation which is due to the effect of easterly winds as opposed to prevalent westerlies. In this context such a situation is called meteorological blocking situation.

**3.4. Seasonal and interannual variability of TA and DIC ~~concentrations~~**

The period from 2001 to 2009 was simulated for the scenarios A and B. For both scenarios monthly mean surface ~~concentrations of~~ TA ~~were~~was calculated in the validation area and are shown in Fig. 8a and 8b. The highest TA ~~concentration~~ in scenario A was 2329 µmol TA kg$^{-1}$ and occurred in July 2003. The lowest TA ~~concentrations~~ in each year ~~were~~was about 2313 to 2318 µmol TA kg$^{-1}$ and occurred in February and March. Scenario B showed generally higher values: Summer concentrations were in the range of 2348 to 2362 µmol TA kg$^{-1}$ and the values peaked in 2003. The lowest values occurred in the years 2004 – 2008. Also, winter values were higher in scenario B than in scenario A: They range from 2322 to 2335 µmol TA kg$^{-1}$.

Corresponding to TA, monthly mean surface DIC ~~concentrations~~ in the validation area ~~are~~is shown in Fig. 8c and 8d. In scenario A the concentrations increased from October to February and decreased from March to August (Fig. 8c). In scenario B the time interval with increasing concentrations was extended into March. Maximum values of 2152 to 2172 µmol DIC kg$^{-1}$ in scenario A occur in February and March of each model year, and minimum values of 2060 to 2080 µmol DIC kg$^{-1}$ in August. Scenario B shows generally higher values: Highest values in February and March are 2161 to 2191 µmol DIC kg$^{-1}$. Lowest values in August range from 2095 to 2112 µmol DIC kg$^{-1}$. The amplitude of the annual cycle is smaller in scenario B, because the Wadden Sea export shows highest values in summer (Fig. 2).

The pattern of ~~the~~ monthly TA and DIC ~~concentrations~~ of the reference scenario A differ drastically in that TA does not show a strong seasonal variability, whereas DIC does vary significantly. In case of DIC this is due to the biological drawdown during summer. ~~On the other hand~~Contrariwise, the additional input (scenario B) from the Wadden Sea in summer creates a strong seasonality for TA and instead flattens the variations in DIC.

**4. Discussion**

Thomas et al. (2009) estimated the contribution of shallow intertidal and subtidal areas to the alkalinity budget of the SE North Sea. That estimate (by closure of mass fluxes) was about 73 Gmol TA yr$^{-1}$ originating from the Wadden Sea fringing the southern and eastern coast. These calculations were based on observations from the CANOBA dataset in 2001 and 2002.

The observed high TA ~~concentrations~~levels in the south-eastern North Sea were also
encountered in August 2008 (Salt et al., 2013) and these measurements were used for the
main model validation in this study. Our simulations result in 39 Gmol TA yr$^{-1}$ as export from
the Wadden Sea into the North Sea. Former modelling studies of the carbonate system of the
North Sea (Artioli et al., 2012; Lorkowski et al., 2012) did not consider the Wadden Sea as a
source of TA and DIC, and good to reasonable agreement to observations from the CANOBA
dataset was only achieved in the open North Sea in 2001 / 2002 (Thomas et al., 2009).
Subsequent simulations that included TA export from aerobic and anaerobic processes in the
sediment improved the agreement between data and models (Pätsch et al., 2018). When
focusing on the German Bight, however, the observed high TA ~~concentrations~~levels in summer
measurements east of 7° E could not be simulated satisfactorily.
The present study confirms the Wadden Sea as an important TA source for the German Bight
and quantifies the annual Wadden Sea TA export rate to 39 Gmol TA yr$^{-1}$. Additionally, the
contributions by most important rivers have been more precisely quantified and narrow down
uncertainties in the budgets of TA and DIC in the German Bight. All steps that were required
to calculate the budget including uncertainties are discussed in the following.

### 4.1. Uncertainties of Wadden Sea – German Bight exchange rates of TA and DIC

The Wadden Sea is an area of effective benthic decomposition of organic material (Böttcher
et al., 2004; Billerbeck et al., 2006; Al-Rai et al., 2009; van Beusekom et al., 2012) originating
both from land and from the North Sea (Thomas et al., 2009). In general, anaerobic
decomposition of the organic matter generates TA and increases the $CO_2$ buffer capacity of
seawater. On longer time scales TA can only be generated by processes that involve
permanent loss of anaerobic remineralisation products (Hu and Cai, 2011). A second
precondition is the nutrient availability to produce organic matter, which in turn serves as
necessary component of anaerobic decomposition (Gustafsson et al., 2019). The Wadden Sea
export rates of TA and DIC modelled in the present study are based on concentration
measurements during tidal cycles in the years 2002 and 2009 to 2011 (Table 1), and on
calculated tidal prisms of two day-periods that are considered to be representative of annual
mean values. This approach introduces uncertainties with respect to the true amplitudes of
concentrations differences in the tidal cycle and in seasonality due to the fact that differences
in concentrations during falling and rising water levels were linearly interpolated. These
interpolated values are based on four to five measurements in the three export areas and
were conducted in different years. Consequently, the approach does not reproduce the exact
TA and DIC ~~concentrations~~levels in the years 2001 to 2009, because only meteorological
forcing, river loads and nitrogen deposition were specified for these particular years. The
simulation of scenario B thus only approximates Wadden Sea export rates. More
measurements distributed with higher resolution over the annual cycle would clearly improve
our estimates. Nevertheless, the implementation of Wadden Sea export rates here results in
improved reproduction of observed high TA ~~concentrations~~levels in the German Bight in
summer in comparison to the reference run A (Fig. 3).
We calculated the sensitivity of our modelled annual TA export rates on uncertainties of the
Δ-values of Table 1. As the different areas North- and East Frisian Wadden Sea and Jade Bay
has different exchange rates of water, for each region the uncertainty of 1 µmol kg$^{-1}$ in ΔTA at
all times has been calculated. The East Frisian Wadden Sea export would differ by
0.84 Gmol TA yr$^{-1}$, the Jade Bay export by 0.09 Gmol TA yr$^{-1}$ and the North Frisian export by
3 Gmol TA yr$^{-1}$.
Primary processes that contribute to the TA generation in the Wadden Sea are denitrification,
sulphate reduction, or processes that are coupled to sulphate reduction and other processes
(Thomas et al., 2009). In our model, the implemented benthic denitrification does not
generate TA (Seitzinger & Giblin, 1996), because modelled benthic denitrification does not
consume nitrate (Pätsch & Kühn, 2008). Benthic denitrification is coupled to nitrification in the
upper layer of the sediment (Raaphorst et al., 1990), giving reason for neglecting TA
generation by this process in the model. The modelled production of N$_2$ by benthic
denitrification falls in the range of 20 – 25 Gmol N yr$^{-1}$ in the validation area, which would
result in a TA production of about 19 – 23 Gmol TA yr$^{-1}$ (Brenner et al., 2016). In the model
nitrate uptake by phytoplankton produces about 40 Gmol TA yr$^{-1}$~~, which partly~~. Assuming
large parts of organic matter exported out of the validation area this production compensates
the missing TA generation by benthic denitrification. This amount of nitrate would not fully be
available for primary production if parts of it would be consumed by denitrification. Different
from this, the TA budget of Thomas et al. (2009) included estimates for the entire benthic
denitrification as a TA generating process.
Sulphate reduction (not modelled here) also contributes to alkalinity generation. On longer
time scales the net effect is vanishing as the major part of the reduced components are
immediately re-oxidised in contact with oxygen. Iron- and sulphate- reduction generates TA
but only their reaction product iron sulphide (essentially pyrite) conserves the reduced
components from re-oxidation. As the formation of pyrite consumes TA, the TA contribution
of iron reduction in the North Sea is assumed to be small and to balance that of pyrite
formation (Brenner et al., 2016).
Atmospheric nitrogen deposition is taken into account in the simulations. Oxidised N-species
($NO_x$) dominate reduced species ($NH_y$) slightly in the validation area during 6 out of 9
simulation years. This implies that the deposition of dissolved inorganic nitrogen decreases TA
in 6 of 9 years. The average decrease within 6 years is about 0.4 Gmol TA $yr^{-1}$, whereas the
average increase within 3 years is only 0.1 Gmol TA $yr^{-1}$. Thomas et al. (2009) also assumed a
dominance of oxidised species and consequently defined a negative contribution to the TA
budget.
Dissolution of biogenic carbonates may be an efficient additional enhancement of the $CO_2$
buffer capacity (that is: source of TA), since most of the tidal flat surface sediments contain
carbonate shell debris (Hild, 1997). ~~On the other hand, shallow~~Shallow oxidation of biogenic
methane formed in deep ~~and~~or shallow tidal flat sediments (not modelled) (Höpner &
Michaelis, 1994; Neira & Rackemann, 1996; Böttcher et al., 2007) has the potential both to
lower or enhance the buffer capacity, thus counteracting or ~~balancing~~promoting the
respective effect of carbonate dissolution. The impact of methane oxidation on the developing
TA / DIC ratio in surface sediments, however, is complex and controlled by a number of
superimposing biogeochemical processes (e.g., Akam et al., 2020).
The net effect of evaporation and precipitation in the Wadden Sea also has to be considered
in budgeting TA. Although these processes are balanced in the North Sea (Schott, 1966),
enhanced evaporation can occur in the Wadden Sea due to increased heating during low tide
around noon. Onken & Riethmüller (2010) estimated an annual negative freshwater budget
in the Hörnum Basin based on long-term hydrographic time series from observations in a tidal

channel. From this data a mean salinity difference between flood and ebb currents of approximately -0.02 is calculated. This would result in an ~~increased~~increase of TA ~~concentration of~~by 1 µmol TA kg$^{-1}$, which is within the range of the uncertainty of measurements. Furthermore, the enhanced evaporation estimated from subtle salinity changes interferes with potential input of submarine groundwater into the tidal basins, that been identified by Moore et al. (2011), Winde et al. (2014), and Santos et al. (2015). The magnitude of this input is difficult to estimate at present, for example from salinity differences between flood and ebb tides, because the composition of SGD passing the sediment-water interfacial mixing zone has to be known. Although first characteristics have been reported (Moore et al., 2011; Winde et al., 2014; Santos et al., 2015), the quantitative effect of additional DIC, TA, and nutrient input via both fresh and recirculated SGD into the Wadden Sea remains unclear.

An input of potential significance are small inlets that provide fresh water as well as DIC and TA (Table 3). The current data base for seasonal dynamics of this source, however, is limited and, therefore, this source cannot yet be considered quantitatively in budgeting approaches.

### 4.2 TA / DIC ratios over the course of the year

Ratios of TA and DIC generated in the tidal basins (Table 1) give some indication of the dominant biogeochemical mineralisation and re-oxidation processes occurring in the sediments of individual Wadden Sea sectors, although these processes have not been explicitly modelled here (Chen & Wang, 1999; Zeebe & Wolf-Gladrow, 2001; Thomas et al. 2009; Sippo et al., 2016; Wurgaft et al., 2019; Akam et al., 2020). Candidate processes are numerous and the export ratios certainly express various combinations, but the most quantitatively relevant likely are aerobic degradation of organic material (resulting in a reduction of TA due to nitrification of ammonia to nitrate with a TA / DIC ratio of -0.16), denitrification (TA / DIC ratio of 0.8, see Rassmann et al., 2020), and anaerobic processes related to sulphate reduction of organoclastic material (TA / DIC ratio of 1, see Sippo et al., 2016). Other processes are aerobic (adding only DIC) and anaerobic (TA / DIC ratio of 2) oxidation of upward diffusing methane, oxidation of sedimentary sulphides upon

resuspension into an aerated water column (no effect on TA / DIC) followed by oxidation of
iron (consuming TA), and nitrification of ammonium (consuming TA, TA / DIC ratio is -2, see
Pätsch et al., 2018 and Zhai et al. 2017).
The TA / DIC export ratios of DIC and TA for the individual tidal basins in three Wadden Sea
sectors (East Frisian, Jade Bay and North Frisian) as calculated from observed ΔTA and ΔDIC
over tidal cycles in different seasons are depicted in Fig. 9. They may give an indication of
regionally and seasonally varying processes occurring in the sediments of the three study
regions. The ratios vary between 0.2 and 0.5 in the North Frisian Wadden Sea with slightly
more TA than DIC generated in spring, summer and autumn, and winter having a negative
ratio of -0.5. The winter ratio coincides with very small measured differences of DIC in
imported and exported waters (ΔDIC = -2 µmol kg$^{-1}$) and the negative TA / DIC ratio may thus
be spurious. The range of ratios in the other seasons is consistent with sulphate reduction and
denitrification as the dominant processes in the North Frisian tidal basins.
The TA / DIC ratios in the Jade Bay samples were consistently higher than those in the North
Frisian tidal basin and vary between 1 and 2 in spring and summer, suggesting a significant
contribution by organoclastic sulphate reduction and anaerobic oxidation of methane (Al-Raei
et al., 2009). The negative ratio of -0.4 in autumn is difficult to explain with remineralisation
or re-oxidation processes, but as with the fall ratio in Frisian tidal basin, it coincides with a
small change in ΔDIC (-3 µmol kg$^{-1}$) at positive ΔTA (8 µmol kg$^{-1}$). Taken at face value, the
resulting negative ratio of -0.4 implicates a re-oxidation of pyrite, normally at timescales of
early diagenesis thermodynamically stable (Hu and Cai, 2011), possibly promoted by
increasing wind forces and associated aeration and sulphide oxidation of anoxic sediment
layers (Kowalski et al., 2013). The DIC export rate from Jade Bay had its minimum in autumn,
consistent with a limited supply and mineralisation of organic matter, possibly modified by
seasonally changing impacts from small tidal inlets (Table 3).
The TA / DIC ratio of the East Frisian Wadden Sea is in the approximate range of those in Jade
Bay, but has one unusually high ratio in November caused by a significant increase in TA of
14 µmol kg$^{-1}$ at a low increase of 5 µmol kg$^{-1}$ in DIC. Barring an analytical artefact, the
maximum ratio of 3 may reflect a short-term effect of iron reduction.
Based on these results, processes in the North Frisian Wadden Sea export area differ from the
East Frisian Wadden Sea and the Jade Bay areas. The DIC export rates suggest that significant

amounts of organic matter were degraded in North Frisian tidal basins, possibly controlled by higher daily exchanged water masses in the North Frisian (8.1 km³ d⁻¹) than in the East Frisian Wadden Sea (2.3 km³ d⁻¹) and in the Jade Bay (0.8 km³ d⁻¹) (compare Table 2). ~~On the other hand~~However, TA export rates of the North Frisian and the East Frisian Wadden Sea were in the same range.

Regional differences in organic matter mineralisation in the Wadden Sea have been discussed by van Beusekom et al. (2012) and Kowalski et al. (2013) in the context of connectivity with the open North Sea and influences of eutrophication and sedimentology. They suggested that the organic matter turnover in the entire Wadden Sea is governed by organic matter import from the North Sea, but that regionally different eutrophication effects as well as sediment compositions modulate this general pattern. The reason for regional differences may be related to the shape and size of the individual tidal basins. van Beusekom et al. (2012) found that wider tidal basins with a large distance between barrier islands and mainland, as is the case in the North Frisian Wadden Sea, generally have a lower eutrophication status than narrower basins predominating in the East Frisian Wadden Sea. Together with the high-water exchange rate the accumulation of organic matter is reduced in the North Frisian Wadden Sea and the oxygen demand per volume is lower than in the more narrow eutrophicated basins. Therefore, aerobic degradation of organic matter dominated in the North Frisian Wadden Sea, where the distance between barrier islands and mainland is large. This leads to less TA production (in relation to DIC production) than in the East Frisian Wadden Sea, where anaerobic degradation of organic matter dominated in more restricted tidal basins.

### 4.3. TA budgets and variability of TA inventory in the German Bight

Modelled TA and DIC ~~concentrations~~ in the German Bight have a high interannual and seasonal variability (Fig. 8). The interannual variability of the model results are mainly driven by the physical prescribed environment. Overall, the TA variability is more sensitive to Wadden Sea export rates than DIC variability, because the latter is dominated by biological processes. However, the inclusion of Wadden Sea DIC export rates improved correspondence with observed DIC concentrations in the near-coastal North Sea.

It is a logical step to attribute the TA variability to variabilities of the different sources. In order
to calculate a realistic budget, scenario B was considered. Annual and seasonal budgets of TA
sources and sinks in this scenario are shown in Table 6. Note that $Riv_{eff}$ is not taken into
account for the budget calculations. This is explained in the Method Section 2.2.2 "River
Input".
Comparing the absolute values of all sources and sinks of the mean year results in a relative
ranking of the processes. 41 % of all TA inventory changes in the validation area were due to
river loads, 37 % were due to net transport, 16 % were due to Wadden Sea export rates, 6 %
were due to internal processes. River input ranged from 78 to 152 Gmol TA yr$^{-1}$ and had the
highest absolute variability of all TA sources in the validation area. This is mostly due to the
high variability of annual freshwater discharge, which is indicated by low (negative) values of
$Riv_{eff}$. The latter values show that the riverine TA loads together with the freshwater flux
induce a small dilution of TA in the validation area for each year. Certainly, this ranking
depends mainly on the characteristics of the Elbe estuary. Due to ~~the~~ high ~~concentration of~~
TA in rivers Rhine and Meuse (Netherlands) they had an effective river input of
+24 Gmol TA yr$^{-1}$ in 2008, which constitutes a much greater impact on TA ~~concentration~~
changes than the Elbe river. In a sensitivity test, we switched off the TA loads of rivers Rhine
and Meuse for the year 2008 and found that the net flow of -71 Gmol TA yr$^{-1}$ decreased
to -80 Gmol TA yr$^{-1}$, which indicates that water entering the validation box from the western
boundary is less TA-rich in the test case than in the reference run.
At seasonal time scales (Table 6 lower part) the net transport dominated the variations from
October to March, while internal processes play a more important role from April to June
(28 %). The impact of effective river input was less than 5 % in every quarter. The Wadden Sea
TA export rates had an impact of 36 % on TA mass changes in the validation area from July to
September. Note that these percentages are related to the sum of the absolute values of the
budgeting terms.
Summing up the sources and sinks, Wadden Sea exchange rates, internal processes and
effective river loads resulted in highest sums in 2002 and 2003 (51 and 52 Gmol TA yr$^{-1}$) and
lowest in 2009 (44 Gmol TA yr$^{-1}$). For the consideration of TA variation we excluded net
transport and actual river loads, because these fluxes are diluted and do not necessarily
change the TA concentrations. In agreement with this, the highest TA ~~concentrations were~~was
simulated in summer 2003 (Fig. 8). The high interannual variability of summer concentrations
was driven essentially by hydrodynamic differences between the years. Flushing times and
their interannual variability were higher in summer than in winter (Fig. 6) of every year. High
flushing times or less strong circulation do have an accumulating effect on exported TA in the
validation area. To understand the reasons of the different flushing times monthly stream
patterns were analysed (Fig. 7). Distinct anticlockwise stream patterns defined the
hydrodynamic conditions in every winter. Summer stream patterns were in most years
weaker, especially in the German Bight (compare Fig. 7, June 2003). In August 2003 the
eastern part of the German Bight shows a clockwise rotation, which transports TA-enriched
water from July back to the Wadden-Sea area for further enrichment. This could explain the
highest concentrations in summer 2003.
Thomas et al. (2009) estimated that 73 Gmol TA yr$^{-1}$ were produced in the Wadden Sea. Their
calculations were based on measurements in 2001 and 2002. The presented model was
validated with data measured in August 2008 (Salt et al., 2013) at the same positions. High TA
~~concentrations~~ in the German Bight ~~were~~was observed in summer 2001 and in summer 2008.
Due to the scarcity of data, the West Frisian Wadden Sea was not considered in the
simulations, but, as the western area is much larger than the eastern area, the amount of
exported TA from that area can be assumed to be in the same range as from the East Frisian
Wadden Sea (10 to 14 Gmol TA yr$^{-1}$). With additional export from the West Frisian Wadden
Sea, the maximum overall Wadden Sea export may be as high as 53 Gmol TA yr$^{-1}$. Thus, the TA
export from the Wadden Sea calculated in this study is 20 to 34 Gmol TA yr$^{-1}$ lower than that
assumed in the study of Thomas et al. (2009). This is mainly due to the flushing time that was
assumed by Thomas et al. (2009). They considered the water masses to be flushed within six
weeks (Lenhart et al., 1995). Flushing times calculated in the present study were significantly
longer and more variable in summer. Since the Wadden Sea export calculated by Thomas et
al. (2009) was defined as a closing term for the TA budget, underestimated summerly flushing
times led to an overestimation of the exchange with the adjacent North Sea.
Table 4 shows that our scenario B underestimates ~~the~~observed TA ~~concentration~~by about
5.1 µmol kg$^{-1}$ in 2008. Scenario A has lower TA~~concentration~~ than scenario B in the validation
area. The difference is about 11 µmol kg$^{-1}$. This means that the Wadden Sea export of
39 Gmol TA yr$^{-1}$ results in a concentration difference of 11 µmol kg$^{-1}$. Assuming linearity, the
deviation between scenario B and the observations (5.1 µmol kg$^{-1}$) would be compensated by
an additional Wadden Sea export of about 18 Gmol TA yr$^{-1}$. If we assume that the deviation
between observation and scenario B is entirely due to uncertainties or errors in the Wadden
Sea export estimate, then the uncertainty of this export is 18 Gmol TA yr$^{-1}$.
Another problematic aspect in the TA export estimate by Thomas et al. (2009) is the fact that
their TA budget merges the sources of anaerobic TA generation from sediment and from the
Wadden Sea into a single source "anaerobic processes in the Wadden Sea". Burt et al. (2014)
found a sediment TA generation of 12 mmol TA m$^{-2}$ d$^{-1}$ at one station in the German Bight
based on Ra-measurements. This fits into the range of microbial gross sulphate reduction rates
reported by Al-Raei et al. (2009) in the back-barrier tidal areas of Spiekeroog island, and by
Brenner et al. (2016) at the Dutch coast. Within the latter paper, the different sources of TA
from the sediment were quantified. The largest term was benthic calcite dissolution, which
would be cancelled out in terms of TA generation assuming a steady-state compensation by
biogenic calcite production. Extrapolating the southern North Sea TA generation (without
calcite dissolution) from the data for one station of Brenner et al. (2016) results in an annual
TA production of 12.2 Gmol in the German Bight (Area = 28.415 km$^2$). This is likely an upper
limit of sediment TA generation, as the measurements were done in summer when seasonal
fluxes are maximal. This calculation reduces the annual Wadden Sea TA generation estimated
by Thomas et al. (2009) from 73 to 61 Gmol, which is still higher than our present estimate. In
spite of the unidentified additional TA-fluxes, both the estimate by Thomas et al. (2009) and
our present model-based quantification confirm the importance of the Wadden-Sea export
fluxes of TA on the North Sea carbonate system at present and in the future.
***4.4 The impact of exported TA and DIC on the North Sea and influences on export***
***magnitude***
Observed high TA and DIC ~~concentrations~~ in the SE North Sea are mainly caused by TA and DIC
export from the Wadden Sea (Fig. 3-5). TA ~~concentrations~~ could be better reproduced than
DIC ~~concentrations~~ in the model experiments, which was mainly due to the higher sensitivity
of DIC to modelled biology. Nevertheless, from a present point of view the Wadden Sea is the
main driver of TA ~~concentrations~~ in the German Bight. Future forecast studies of the evolution
of the carbonate system in the German Bight will have to specifically focus on the Wadden
Sea and on processes occurring there. In this context the Wadden Sea evolution during future
sea level rise is the most important factor. The balance between sediment supply from the
North Sea and sea level rise is a general precondition for the persistence of the Wadden Sea
(Flemming and Davis, 1994; van Koningsveld et al., 2008). An accelerating sea level rise could
lead to a deficient sediment supply from the North Sea and shift the balance at first in the
largest tidal basins and at last in the smallest basins. (CPSL, 2001; van Goor et al., 2003). The
share of intertidal flats as potential sedimentation areas is larger in smaller tidal basins (van
Beusekom et al., 2012), whereas larger basins have a larger share of subtidal areas. Thus,
assuming an accelerating sea level rise, large tidal basins will turn into lagoons, while tidal flats
may still exist in smaller tidal basins. This effect could decrease the overall Wadden Sea export
rates of TA, because sediments would no longer be exposed to the atmosphere and the
products of sulphate reduction would re-oxidise in the water column. Moreover, benthic-
pelagic exchange in the former intertidal flats would be more diffusive and less advective then
today due to a lowering of the hydraulic gradients during ebb tides, when parts of the
sediment become unsaturated with water. This would decrease TA export into the North Sea.
Caused by changes in hydrography and sea level the sedimentological composition may also
change. If sediments become more sandy, aerobic degradation of organic matter is likely to
become more important (de Beer et al., 2005). In fine grained silt diffusive transport plays a
key role, while in the upper layer of coarse (sandy) sediments advection is the dominant
process. Regionally, the North Frisian Wadden Sea will be more affected by rising sea level
because there the tidal basins are larger than the tidal basins in the East Frisian Wadden Sea
and even larger than the inner Jade Bay.
The Wadden Sea export of TA and DIC is driven by the turnover of organic material. Decreasing
anthropogenic eutrophication can lead to decreasing phytoplankton biomass and production
(Cadée & Hegeman, 2002; van Beusekom et al., 2009). Thus, the natural variability of the
North Sea primary production becomes more important in determining the organic matter
turnover in the Wadden Sea (McQuatters-Gollop et al., 2007; McQuatters-Gollop & Vermaat,
2011). pH values in Dutch coastal waters decreased from 1990 to 2006 drastically. Changes in
nutrient variability were identified as possible drivers (Provoost et al., 2010), which is
consistent with model simulations by Borges and Gypens (2010). Moreover, despite the
assumption of decreasing overall TA export rates from the Wadden Sea the impact of the
North Frisian Wadden Sea on the carbonate system of the German Bight could potentially
adjust to a change of tidal prisms and thus a modulation in imported organic matter. If less
organic matter is remineralised in the North Frisian Wadden Sea, less TA and DIC will be
exported into the North Sea.
In the context of climate change, processes that have impact on the freshwater budget of tidal
mud flats will gain in importance. Future climate change will have an impact in coastal
hydrology due to changes in ground water formation rates (Faneca Sànchez et al., 2012;
Sulzbacher et al., 2012), that may change both surface and subterranean run-off into the
North Sea. An increasing discharge of small rivers and groundwater into the Wadden Sea is
likely to increase DIC, TA, and possibly nutrient loads and may enhance the production of
organic matter. Evaporation could also increase due to increased warming and become a more
important process than today (Onken & Riethmüller, 2010), as will methane cycling change
due to nutrient changes, sea level and temperature rise (e.g., Höpner and Michaelis, 1994;
Akam et al., 2020).
Concluding, in the course of climate change the North Frisian Wadden Sea will be affected first
by sea level rise, which will result in decreased TA and DIC export rates due to less turnover of
organic matter there. This could lead to a decreased buffering capacity in the German Bight
for atmospheric $CO_2$. Overall, less organic matter will be remineralised in the Wadden Sea.

*5 Conclusion and Outlook*

We present a budget calculation of TA sources in the German Bight and relate 16 % of the
annual TA inventory changes to TA exports from the Wadden Sea. The impact of riverine bulk
TA seems to be less important due to the comparatively low TA ~~concentrations~~levels in the
Elbe estuary, a finding that has to be proven by future research.
The evolution of the carbonate system in the German Bight under future changes depends on
the development of the Wadden Sea. The amount of TA and DIC that is exported from the
Wadden Sea depends on the amount of organic matter and / or nutrient that are imported
from the North Sea and finally remineralised in the Wadden Sea. Decreasing riverine nutrient
loads led to decreasing phytoplankton biomass and production (Cadée & Hegeman, 2002; van
Beusekom et al., 2009), a trend that is expected to continue in the future (European Water
Framework Directive). However, altered natural dynamics of nutrient cycling and productivity
can override the decreasing riverine nutrient loads (van Beusekom et al., 2012), but these will
not generate TA in the magnitude of denitrification of river-borne nitrate.
Sea level rise in the North Frisian Wadden Sea will potentially be more affected by a loss of
intertidal areas than the East Frisian Wadden Sea (van Beusekom et al., 2012). This effect will
likely reduce the turnover of organic material in this region of the Wadden Sea, which may
decrease TA production and transfer into the southern North Sea.
Thomas et al. (2009) estimated that the Wadden Sea facilitates approximately 7 – 10% of the
annual $CO_2$ uptake of the North Sea. This is motivation for model studies on the future role of
the Wadden Sea in the $CO_2$ balance of the North Sea under regional climate change.
Future research will also have to address the composition and amount of submarine ground
water discharge, as well as the magnitude and seasonal dynamics in discharge and
composition of small water inlets at the coast, which are in this study only implicitly included
and in other studies mostly ignored due to a lacking data base.
**Data availability**
The river data are available at https://wiki.cen.uni-hamburg.de/ifm/ECOHAM/DATA_RIVER
and www.waterbase.nl. Meteorological data are stored at https://psl.noaa.gov/. The North
Sea TA and DIC data are stored at https://doi.org/10.1594/PANGAEA.438791 (2001),
https://doi.org/10.1594/PANGAEA.441686 (2005). The data of the North Sea cruise 2008 have
not been published, yet, but can be requested via the CODIS data portal
(http://www.nioz.nl/portals-en; registration required). Additional Wadden Sea TA and DIC
data are deposited under doi:10.1594/PANGEA.841976.

**Author contributions**
The scientific concept for this study was originally developed by JP and MEB. FS wrote the
basic manuscript as part of his PhD thesis. VW provided field analytical data, as part of her
PhD thesis. JP developed the original text further with contributions from all co-authors.
**Competing interests**
The authors declare that they have no conflict of interest.

**Acknowledgements**
The authors appreciate the two constructive reviews, which greatly helped to improve the
manuscript, and the editorial handling by Jack Middelburg. I. Lorkowski, W. Kühn, and F. Große
are acknowledged for stimulating discussions, S. Grashorn for providing tidal prisms and P.
Escher for laboratory support. This work was financially supported by BMBF during the Joint
Research Project BIOACID (TP 5.1, 03F0608L and TP 3.4.1, 03F0608F), with further support
from Leibniz Institute for Baltic Sea Research. We also acknowledge the support by the Cluster
of Excellence 'CliSAP' (EXC177), University of Hamburg, funded by the German Science
Foundation (DFG) and the support by the German Academic Exchange service (DAAD,
MOPGA-GRI, #57429828) with funds of the German Federal Ministry of Education and
Research (BMBF). We used NCEP Reanalysis data provided by the NOAA/OAR/ESRL PSL,
Boulder, Colorado, USA, from their Web site at https://psl.noaa.gov/.

**Tables**

**Table 1: Mean TA and DIC concentrations [µmol l$^{-1}$] during rising and falling water levels and the respective differences (Δ-values) that were used as wad_sta in (1). Areas are the North Frisian (N), the East Frisian (E) Wadden Sea and the Jade Bay (J).**

| Area | Date | TA (rising) | TA (falling) | ΔTA | DIC (rising) | DIC (falling) | ΔDIC |
|------|------|-------------|--------------|-----|--------------|---------------|------|
| N | 29.04.2009 | 2343 | 2355 | 12 | 2082[*] | 2106 | 24 |
| | 17.06.2009 | 2328 | 2332 | 4 | 2170 | 2190 | 20 |
| | 26.08.2009 | 2238 | 2252 | 14 | 2077 | 2105 | 28 |
| | 05.11.2009 | 2335 | 2333 | -2 | 2205 | 2209 | 4 |
| J | 20.01.2010 | 2429 | 2443 | 14 | 2380 | 2392 | 12 |
| | 21.04.2010 | 2415 | 2448 | 33 | 2099 | 2132 | 33 |
| | 26.07.2010 | 2424 | 2485 | 61 | 2159 | 2187 | 28 |
| | 09.11.2010 | 2402 | 2399 | -3 | 2302 | 2310 | 8 |
| E | 03.03.2010 | 2379 | 2393 | 14 | 2313 | 2328 | 15 |
| | 07.04.2010 | 2346 | 2342 | -4 | 2068 | 2082 | 14 |
| | 17./18.05.2011 | 2445 | 2451 | 6 | 2209 | 2221 | 12 |
| | 20.08.2002 | 2377 | 2414 | 37 | 2010 | 2030 | 20 |
| | 01.11.2010 | 2423 | 2439 | 16 | 2293 | 2298 | 5 |

[*]: This value was estimated.

**Table 2: Daily Wadden Sea runoff to the North Sea at different export areas.**

| Position | wad_exc [10$^6$ m³ d$^{-1}$] |
|----------|------------------------------|
| N1 | 273 |
| N2 | 1225 |
| N3 | 1416 |
| N4 | 1128 |
| N5 | 4038 |
| N6 | 18 |
| J1 - J3 | 251 |
| E1 | 380 |
| E2 | 634 |
| E3 | 437 |
| E4 | 857 |

 **Table 3: Examples for the carbonate system composition of small fresh water inlets**

 **draining into the Jade Bay and the backbarrier tidal area of Spiekeroog Island, given in**

 **($\mu$mol kg$^{-1}$). Autumn results (A) (October 31$^{st}$, 2010) are taken from Winde et al. (2014);**

 **spring sampling (S) took place on May 20$^{th}$, 2011.**

| Site | Position | DIC(A) | TA(A) | DIC(S) | TA(S) |
|------|----------|--------|-------|--------|-------|
| Neuharlingersiel | 53°41.944 N 7°42.170 E | 2319 | 1773 | 1915 | 1878 |
| Harlesiel | 53°42.376 N 7°48.538 E | 3651 | 3183 | 1939 | 1983 |
| Wanger- /Horumersiel | 53°41.015 N 8°1.170 E | 5405 | 4880 | 6270 | 6602 |
| Hooksiel | 53°38.421 N 8°4.805 E | 2875 | 3105 | 3035 | 3302 |
| Maade | 53°33.534 N 8°7.082 E | 5047 | 4448 | 5960 | 6228 |
| Mariensiel | 53°30.895 N 8°2.873 E | 6455 | 5904 | 3665 | 3536 |
| Dangaster Siel | 53°26.737N 8°6.577 E | 1868 | 1246 | 1647 | 1498 |
| Wappelersiel | 53°23.414 N 8°12.437 E | 1373 | 630 | 1358 | 1152 |
| Schweiburger Siel | 53°24.725 N 8°16.968 E | 4397 | 3579 | 4656 | 4493 |
| Eckenwarder Siel | 53°31.249 N 8°16.527 E | 6542 | 6050 | 2119 | 4005 |


**Table 4: Averages (μmol kg$^{-1}$), standard deviations (μmol kg$^{-1}$), RMSE (μmol kg$^{-1}$), and correlation coefficients r for the observed TA concentrations and the corresponding scenarios A and B within the validation area.**

| TA | Average | Stdv | RMSE | r |
|---|---|---|---|---|
| Obs 2008 | 2333.52 | 32.51 | | |
| Obs 2005 | 2332.09 | 21.69 | | |
| Obs 2001 | 2333.83 | 33.19 | | |
| Sim A 2008 | 2327.64 | 6.84 | 27.97 | 0.77 |
| Sim A 2005 | 2322.16 | 5.21 | 22.05 | 0.45 |
| Sim A 2001 | 2329.79 | 5.32 | 31.89 | 0.24 |
| Sim B 2008 | 2338.60 | 22.09 | 18.34 | 0.86 |
| Sim B 2005 | 2339.48 | 26.81 | 31.81 | 0.18 |
| Sim B 2001 | 2342.96 | 17.28 | 30.07 | 0.47 |













**Table 5: Averages (μmol kg$^{-1}$), standard deviations (μmol kg$^{-1}$), RMSE (μmol kg$^{-1}$), and correlation coefficients r for the observed DIC concentrations and the corresponding scenarios A and B within the validation area.**


| DIC | Average | Stdv | RMSE | r |
|---|---|---|---|---|
| Obs 2008 | 2107.05 | 24.23 | | |
| Obs 2005 | 2098.20 | 33.42 | | |
| Obs 2001 | 2105.49 | 25.21 | | |
| Sim A 2008 | 2080.93 | 14.24 | 43.48 | -0.64 |
| Sim A 2005 | 2083.53 | 21.94 | 26.97 | 0.73 |
| Sim A 2001 | 2077.53 | 17.61 | 38.89 | 0.22 |
| Sim B 2008 | 2091.15 | 9.25 | 25.87 | 0.55 |
| Sim B 2005 | 2101.26 | 10.97 | 33.96 | 0.10 |
| Sim B 2001 | 2092.69 | 11.71 | 25.33 | 0.48 |











**Table 6: Annual TA budgets in the validation area of the years 2001 to 2009, annual averages and seasonal budgets of January to March, April to June, July to September and October to December [Gmol]. Net Flow is the annual net TA transport across the boundaries of the validation area. Negative values indicate a net export from the validation area to the adjacent North Sea. Δcontent indicates the difference of the TA contents between the last and the first time steps of the simulated year or quarter.**

| | Wadden Sea export | internal processes | river loads | Riv$_{eff}$ | net flow | Δcontent |
|---|---|---|---|---|---|---|
| | Gmol/yr | Gmol/yr | Gmol/yr | Gmol/yr | Gmol/yr | Gmol |
| 2001 | 39 | 13 | 87 | -5 | 38 | 177 |
| 2002 | 39 | 19 | 152 | -7 | -223 | -13 |
| 2003 | 39 | 16 | 91 | -3 | -98 | 48 |
| 2004 | 39 | 13 | 78 | -5 | -8 | 122 |
| 2005 | 39 | 12 | 89 | -5 | -98 | 42 |
| 2006 | 39 | 12 | 88 | -4 | -56 | 83 |
| 2007 | 39 | 12 | 110 | -5 | -132 | 29 |
| 2008 | 39 | 14 | 93 | -5 | -71 | 75 |
| 2009 | 39 | 10 | 83 | -5 | -151 | -19 |
| Average | Gmol/yr | Gmol/yr | Gmol/yr | Gmol/yr | Gmol/yr | Gmol |
| | 39 | 14 | 101 | -5 | -89 | 65 |
| t = 3 mon | Gmol/t | Gmol/t | Gmol/t | Gmol/t | Gmol/t | Gmol |
| Jan - Mar | 7 | -1 | 38 | -1 | -49 | -5 |
| Apr - Jun | 10 | 15 | 23 | -2 | 6 | 54 |
| Jul - Sep | 17 | -2 | 15 | -2 | 13 | 43 |
| Oct - Dec | 4 | 1 | 25 | 0 | -56 | -26 |

   **6.  Figure Captions**


Figure 1: Upper panel: Map of the south-eastern North Sea and the bordering land. Lower
panel: Model domains of ECOHAM (red) and FVCOM (blue), positions of rivers 1 – 16 (left,
see Table 2) and the Wadden Sea export areas grid cells (right). The magenta edges identify
the validation area, western and eastern part separated by the magenta dashed line.
Figure 2: Monthly Wadden Sea export of DIC and TA [Gmol mon$^{-1}$] at the North Frisian
coast (N), East Frisian coast (E) and the Jade Bay in scenario B. The export rates were
calculated for DIC and TA based on measured concentrations and simulated water fluxes.
Figure 3: Surface TA ~~concentrations~~ [µmol TA kg$^{-1}$] in August 2008 observed (a) and simulated
with scenario A (b) and B (c). The black lines indicate the validation box.
Figure 4: Differences between TA surface summer observations and results from
scenario A (a) and B (b) and the differences between DIC surface observations and results
from scenario A (c) and B (d), all in µmol kg$^{-1}$. The black lines indicate the validation box.
Figure 5: Surface DIC concentrations [µmol DIC kg$^{-1}$] in August 2008 observed (a) and
simulated with scenario A (b) and B (c). The black lines indicate the validation box.
Figure 6: Flushing times in the validation area in summer (June to August) and winter (January
to March). The whole validation area is represented in blue, green is the western part of the
validation area (4.5° E to 7° E) and red is the eastern part (east of 7° E).
Figure 7: Monthly mean simulated streamlines for summer months 2003 and 2008.
Figure 8: Simulated monthly mean ~~concentrations of~~ TA (scenario A (a), scenario B (b))
[µmol TA kg$^{-1}$] and DIC (scenario A (c), scenario B (d)) [µmol DIC kg$^{-1}$] in the validation area for
the years 2001-2009.
Figure 9: Temporally interpolated TA/DIC ratio of the export rates in the North Frisian, East
Frisian, and Jade Bay. These ratios are calculated using the Δ-values of Table 1.

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

**8. Appendix**



**Table A1: Annual riverine freshwater discharge [km³ yr⁻¹]. The numbering refers to Fig. 1.**

|  | 2001 | 2002 | 2003 | 2004 | 2005 | 2006 | 2007 | 2008 | 2009 |
|---|---|---|---|---|---|---|---|---|---|
| 1) Elbe | 23.05 | 43.38 | 23.95 | 19.56 | 25.56 | 26.98 | 26.61 | 24.62 | 24.28 |
| 2) Ems | 3.47 | 4.48 | 3.15 | 3.52 | 2.99 | 2.54 | 4.32 | 3.32 | 2.58 |
| 3) Noordzeekanaal | 3.21 | 2.98 | 2.49 | 3.05 | 3.03 | 2.96 | 1.55 | 3.05 | 2.46 |
| 4) Ijsselmeer (east) | 9.55 | 9.94 | 6.27 | 7.97 | 7.35 | 7.30 | 9.10 | 8.23 | 6.59 |
| 5) Ijsselmeer (west) | 9.55 | 9.94 | 6.27 | 7.97 | 7.35 | 7.30 | 9.10 | 8.23 | 6.59 |
| 6) Nieuwe Waterweg | 50.37 | 51.33 | 34.72 | 42.91 | 41.61 | 44.21 | 49.59 | 49.76 | 44.69 |
| 7) Haringvliet | 33.10 | 35.18 | 17.92 | 10.77 | 12.36 | 16.02 | 24.00 | 15.70 | 11.06 |
| 8) Scheldt | 7.28 | 2.74 | 4.31 | 3.64 | 3.59 | 3.74 | 4.63 | 4.57 | 3.63 |
| 9) Weser | 11.43 | 18.97 | 11.80 | 10.52 | 10.37 | 9.72 | 16.21 | 12.59 | 9.58 |
| 10) Firth of Forth | 2.72 | 3.76 | 2.06 | 3.01 | 3.00 | 2.84 | 2.85 | 3.59 | 3.66 |
| 11) Tyne | 1.81 | 2.25 | 1.18 | 2.04 | 1.92 | 1.78 | 2.09 | 2.70 | 2.05 |
| 12) Tees | 1.33 | 1.78 | 0.94 | 1.59 | 1.27 | 1.45 | 1.49 | 1.99 | 1.55 |
| 13) Humber | 10.76 | 12.10 | 7.16 | 10.51 | 7.68 | 11.11 | 12.03 | 13.87 | 9.60 |
| 14) Wash | 5.46 | 4.39 | 3.08 | 3.91 | 1.96 | 2.72 | 5.24 | 4.77 | 3.21 |
| 15) Thames | 4.47 | 3.23 | 2.41 | 2.13 | 0.96 | 1.57 | 3.52 | 3.20 | 2.38 |
| 16) Eider | 0.67 | 0.97 | 0.47 | 0.70 | 0.68 | 0.67 | 0.63 | 0.58 | 0.57 |
| Sum | 178.2 | 207.4 | 128.1 | 133.7 | 131.6 | 142.9 | 172.9 | 160.7 | 134.4 |










**Table A2: River numbers in Fig. 1, their positions and source of data**

| Number in Fig. 1 | Name | River mouth position | | Data source |
|---|---|---|---|---|
| 1 | Elbe | 53°53'20"N | 08°55'00"E | Pätsch & Lenhart (2008); TA-, DIC- and nitrate-concentrations by Amann (2015) |
| 2 | Ems | 53°29'20"N | 06°55'00"E | Pätsch & Lenhart (2008) |
| 3 | Noordzeekanaal | 52°17'20"N | 04°15'00"E | Pätsch & Lenhart (2008); TA-, DIC- and nitrate- |

| | | | | | |
|---|---|---|---|---|---|
| | | | | concentrations from waterbase.nl | |
| 4 | Ijsselmeer (east) | 53°17'20"N | 05°15'00"E | As above | |
| 5 | Ijsselmeer (west) | 53°05'20"N | 04°55'00"E | As above | |
| 6 | Nieuwe Waterweg | 52°05'20"N | 03°55'00"E | As above | |
| 7 | Haringvliet | 51°53'20"N | 03°55'00"E | As above | |
| 8 | Scheldt | 51°29'20"N | 03°15'00"E | As above | |
| 9 | Weser | 53°53'20"N | 08°15'00"E | Pätsch & Lenhart (2008) | |
| 10 | Firth of Forth | 56°05'20"N 02°45'00"W | | HASEC (2012) | |
| 11 | Tyne | 55°05'20"N 01°25'00"W | | HASEC (2012) | |
| 12 | Tees | 54°41'20"N 01°05'00"W | | HASEC (2012) | |
| 13 | Humber | 53°41'20"N 00°25'00"W | | HASEC (2012) | |
| 14 | Wash | 52°53'20"N | 00°15'00"E | HASEC (2012): sum of 4 rivers: Nene, Ouse, Welland and Witham | |
| 15 | Thames | 51°29'20"N | 00°55'00"E | HASEC (2012) | |
| 16 | Eider | 54°05'20"N | 08°55'00"E | Johannsen et al, 2008 | |


**Table A3: Monthly values of TA, DIC and NO$_3$ concentrations [µmol kg$^{-1}$] of rivers, the annual**
**mean and the standard deviation**

| River parameter | Jan | Feb | Mar | Apr | May | Jun | Jul | Aug | Sep | Oct | Nov | Dec | Mean | SD |
|---|---|---|---|---|---|---|---|---|---|---|---|---|---|---|
| Elbe TA | 2380 | 2272 | 2293 | 2083 | 2017 | 1967 | 1916 | 1768 | 1988 | 2156 | 2342 | 2488 | 2139 | 218 |
| Noordzeekanaal TA | 3762 | 3550 | 3524 | 3441 | 4748 | 3278 | 3419 | 3183 | 3027 | 3299 | 3210 | 3413 | 3488 | 441 |
| Nieuwe Waterweg TA | 2778 | 2708 | 2765 | 3006 | 2883 | 2658 | 2876 | 2695 | 2834 | 2761 | 2834 | 2927 | 2810 | 102 |
| Haringvliet TA | 2588 | 2635 | 2532 | 3666 | 2826 | 2829 | 2659 | 2660 | 2496 | 2816 | 2758 | 2585 | 2754 | 309 |
| Scheldt TA | 3781 | 3863 | 3708 | 3725 | 3758 | 3626 | 3722 | 3514 | 3367 | 3666 | 3825 | 3801 | 3696 | 140 |
| Ijsselmeer TA | 2829 | 3005 | 2472 | 2259 | 2611 | 1864 | 1672 | 1419 | 1445 | 2172 | 2286 | 2551 | 2215 | 521 |
| Elbe DIC | 2415 | 2319 | 2362 | 2179 | 2093 | 2025 | 1956 | 1853 | 2018 | 2200 | 2428 | 2512 | 2197 | 211 |
| Noordzeekanaal DIC | 3748 | 3579 | 3470 | 3334 | 3901 | 3252 | 3331 | 3136 | 2977 | 3214 | 3183 | 3405 | 3378 | 264 |
| Nieuwe Waterweg DIC | 2861 | 2794 | 2823 | 2991 | 2879 | 2657 | 2886 | 2706 | 2828 | 2773 | 2907 | 3036 | 2845 | 108 |
| Haringvliet DIC | 2673 | 2735 | 2600 | 3661 | 2850 | 2846 | 2687 | 2681 | 2512 | 2859 | 2803 | 2670 | 2798 | 292 |
| Scheldt DIC | 3798 | 3909 | 3829 | 3737 | 3704 | 3592 | 3705 | 3490 | 3316 | 3648 | 3733 | 3868 | 3694 | 167 |
| Ijsselmeer DIC | 2824 | 3008 | 2458 | 2234 | 2576 | 1826 | 1636 | 1369 | 1399 | 2134 | 2285 | 2565 | 2193 | 538 |
| Elbe NO$_3$ | 247 | 330 | 277 | 225 | 193 | 161 | 129 | 103 | 112 | 157 | 267 | 164 | 197 | 72 |
| Noordzeekanaal NO$_3$ | 150 | 168 | 190 | 118 | 79 | 71 | 64 | 73 | 78 | 92 | 107 | 137 | 111 | 42 |
| Nieuwe Waterweg NO$_3$ | 232 | 243 | 231 | 195 | 150 | 140 | 132 | 135 | 113 | 145 | 201 | 220 | 178 | 47 |
| Haringvliet NO$_3$ | 233 | 252 | 218 | 200 | 143 | 144 | 133 | 117 | 128 | 127 | 143 | 228 | 172 | 50 |
| Scheldt NO$_3$ | 320 | 341 | 347 | 345 | 243 | 221 | 219 | 215 | 189 | 202 | 190 | 274 | 259 | 63 |
| Ijsselmeer NO$_3$ | 136 | 159 | 190 | 192 | 135 | 46 | 20 | 14 | 7 | 18 | 20 | 79 | 85 | 73 |
