# Peer review of "thanks for these hints. In most cases it was possible to incorporate them."

_Biogeosciences, 2020_

## Short Comment (SC1) · 31 Jan 2020

On a purely technical note, I would suggest that the authors adapt the color palettes they use for their figures. For many of the plots the colors are indistinguishable in grey-scale and, thus, for colorblind people. This applies to Figs. 3-5, 8 and 9. Figure 6 lacks a legend.

I am not familiar with ODV color palettes, so I can't give advice for Figs. 3-5. The other figures seem to be produced with MatLab, for which the authors could either use the 'new' default color palette parula or any of the cmocean color palettes (https://uk.mathworks.com/matlabcentral/fileexchange/57773-cmocean-perceptually-uniform-colormaps). For Fig. 9, it would suffice to add different markers

to the different lines.

---

## Author Comment (AC1) · 5 Feb 2020

For ODV we can offer palette TERMAL.

———————————————

---

## Referee Comment (RC1) · Anonymous Referee #1 · 16 Feb 2020

General comments: This manuscript discusses the role of alkalinity export from the Wadden Sea on the carbonate system of the southern North Sea. Specifically, it aims at quantifying this export, its importance for the alkalinity budget, and the relative role of aerobic versus anaerobic processes in generating the alkalinity in the Wadden Sea.

The manuscript presents interesting work, which is worth publishing, but in my opinion the manuscript itself needs some work. The aim and take home messages of the work are not made very clear in either the abstract or the last paragraph of the introduction. The order and relative length of sections does not always appear logical to me. The construction of the TA budget raises some questions. Also, the manuscript is at times difficult to follow without knowing the details of previous studies, especially in the methods section. For example, the biogeochemistry in the model is merely explained. In

short, the writing can be much sharper. In the specific comments I'll provide examples and suggestions.

Also the presentation of data can be improved. As already indicated in another comment, the figures can be improved to support black & white reading, e.g. by using dashed and dotted lines or bars. I also found the order of the tables highly confusing. If I counted correctly they are presented in the text in the order Table 6 – Table 1 – Table 3 – Table 4 – Table 5 – Table 2. Please change this in the next version.

Content-wise, a major point I don't fully understand is the lack of quantification of the uncertainty in the calculated export flux of 39 Gmol TA y-1, which is such a central result of this study. I understand it is based on sparse measurements of DIC and TA concentrations in the different Wadden Sea areas but some estimate of the uncertainty with the use of equation (2) and upscaling of the results should be possible to make. Also, using a fixed value for the Wadden Sea export for the years 2001-2009 but taking into account interannual variability in the other terms of the TA budget makes it difficult to actually quantify the relative contributions for each of the years, as you also expect quite some interannual variability in the export flux. I would therefore suggest to also calculate an average TA budget for the period 2001-2009, as you did for the seasonal pattern, and mostly use that in the discussion of the budget. In my opinion, some more odd choices were made in the TA budget, such as excluding Riveff and the leap years, which need revision and/or better explanations.

Distinguishing between anaerobic and aerobic processes generating TA in the Wadden Sea appears somewhat problematic, since many processes highly relevant for the TA dynamics are initially not taken into account, but are required to explain your data anyway. Take for example oxidation of methane, but also reoxidation of other reduced species (e.g. previously buried sulphur). Given that there are many different processes, the system is so dynamic and exposure of sediments plays such an important role, how can you be sure that the TA/DIC ratio is a reliable metric for the message you want to convey with respect to aerobic and anaerobic degradation? I also miss a discussion

on the relevant time scale of processes here.

The role of the data presented in Table 6 is not entirely clear to me; section 2.7 also does not really seem to match with the rest of the manuscript, especially since in the end they are not used for the TA budget. And if these data have indeed been published elsewhere, it seems unnecessary to publish them here as well. Finally, I do not understand why the model was only validated with 2008 data, whereas there are also data from 2001/2002 (as discussed by the authors) and 2005 available in the CANOBA (or related) datasets. Validating with these data as well, especially with the 2001/2002 data that include multiple seasons, would strengthen the manuscript.

Specific comments: Abstract: I miss a clear aim and a concluding sentence in the abstract. The work is described in lines 23-25, but what is the underlying aim? Confirming the high TA export from the Wadden Sea? Finding out the underlying mechanisms? And similarly, what can we conclude from this work? L. 18-19: This sentence focuses entirely on the physics of coastal oceans, whereas a major reason for coastal acidification being different from open ocean acidification is the fact that inputs and process rates are much higher. See e.g. Duarte et al (2013) (reference added at the end) L. 25: "sources" do you refer to concentrations or fluxes or both here? As the sources are calculated based on measured concentrations and modelled exchange rates, so they are not truly observed. L. 34: can you briefly elaborate what you mean with "weak meteorological blocking conditions"? I'd suggest to paraphrase this in the abstract and explain the term later in the manuscript (e.g. at L. 365 where it appears again). L. 37-38: does the 'net transport' have a particular direction? L. 38: "internal production" is this the gross or net TA production? Of the water column only, or of the combined water and sediment system? L. 42-43: "anaerobic degradation dominated" with which pathway? L. 54-58: Add the suggested mechanisms for the observation in the Provoost article (i.e. change in production-respiration balance). Also refer to Duarte et al (2013) here, as they summarise many important processes impacting pH balance. Now it seems as if only biogeochemical processes in the sediment are important, which is

obviously not the case in many coastal areas. L. 64-67: Also the Baltic Sea is a key example of this; see e.g. Łukawska-Matuszewska (2017) and Gustafsson et al (2019) L. 87-90: If this is the aim of the work it is not written very clearly. What is the general aim? What is the key research question that will be addressed? Is there a certain time period associated with this or is your aim more general? What do you hypothesise? This section really needs some work. Methods: the division of subsections seems oddly chosen. 11 subsections is way too many and yet details are lacking. Subsection 2.7 seems unnecessary and subsection 2.8 way too short. I would suggest to merge some of the subsections, or use a third level instead. L. 158: NO3 data are also presented in this table, but not mentioned here or much discussed in the manuscript. L. 178-197: The purpose of this subsection (and of this data in general) is not entirely clear to me and someone it feels like they were added last-minute. Maybe because they are the first presented but referred to as Table 6. Can the authors please elaborate on why this data were added? Especially since in the end they are not used for the calculation of the TA budget. Also, if the data are presented elsewhere (as L. 180-181 seems to suggest, "reported for completeness only"), there is no need to discuss the methodology here. L. 190-197: if this is the novel part of the manuscript, as seems to be suggested by L. 191 ("the main extension in the present study"), then it really needs a more detailed explanation. Also readers not familiar with Pätsch et al. (2018) need to understand this. Explain which biogeochemical processes are involved and where they take place (water column or sediment or both). How are sediments included in general? A brief mention to Wolf-Gladrow et al. (2007) is not sufficient, as I don't think that the exact same components are included in this study. L. 195: "nutrient dynamics" i.e. productivity and decomposition? L. 196: what about atmospheric sulphur deposition? L. 241-242: I don't understand why the time lag is the reason for the lack of statistical analysis. The low number of observations is the reason. If this is what you mean, then please paraphrase this section. L. 257-258: so scenario A has no Wadden Sea export but the same internal biogeochemistry as scenario B? Or is it the same as previously published implementations of the model? Please elaborate. L. 259-260: So

the data in Fig. 2 are calculated according to Eq. 2? And then multiplied by the area of what? Summed area of the grid cells? Please explain. L. 268-269: This belongs in the introduction. Mention all aims clearly there in the last paragraph. Results: Sections 3.1. and 3.2: why do you discuss the validation of DIC an TA separately? It seems more logical to me that, because they are so connected, you can also discuss them at the same time. That would also shorten this relatively long section. L. 303: what do you mean by "the standard deviation improved"? In scenario A it was lower, i.e. 7 umol/kg. Or do you mean to say that the standard deviation comes closer to that of the observations? L. 348: rather than mentioning high flushing times, I would paraphrase to focus on the low water renewal or long mean residence time. Same in L. 351-352: I would paraphrase to say that highest inflows occurred in winter. L. 354-356: Can't you use a metric to correct for this feature, allowing fairer comparisons? Sections 3.4 and 3.5: Again, I'd suggest to merge these two. L. 385-386: How is this for TA? Discussion: I'd suggest to change the order. Subsection 4.2, which is to a large extent an outlook to the future and partly relies on te TA budget, is much more logical as final subsection. Subsection 4.5 is connected to the Wadden Sea data and it seems logical for it to immediately follow subsection 4.1. L. 394-396: I don't understand why the model was only validated with 2008 data, whereas there are also data from 2001/2002 (as discussed by the authors) and 2005 available in the CANOBA (or related) datasets. Validating with these data as well, especially with the 2001/2002 data that include multiple seasons, would strengthen the manuscript L. 397-399: Move to last paragraph of introduction, this aim is not in there yet. L. 416-417: This is quite a simplified statement; the temporal and spatial scale you consider are highly relevant for whether this is the case and for which processes associated with anaerobic decomposition this is relevant. See e.g. Hu and Cai (2011) and Gustafsson et al (2019) for discussions on this. L. 424-428: Could you add some suggestions for improvement? L. 431-453: Why is S burial not discussed here? This seems highly relevant, especially on the longer term. L. 435-437: So external NO3 inputs are not relevant for benthic denitrification? L. 440-441: Why does this compensate the TA generation? Please explain. L. 444-446:

But how high and relevant is the deposition of these inputs for the TA budget? This is a very qualitative paragraph. L. 470-472: So what is the aim of adding these data to the manuscript if they are published elsewhere and not taken into account for the budget? L. 478-479: How has the sensitivity of DIC to modelled biology been confirmed? L. 479-480: The reader doesn't know this yet as the TA budget has not yet been presented. I'd suggest to change the order. L. 494-496: Thus slower exchange, what would be the effect on the TA export? L. 497-499: Why? L. 504-506: On which time scale? Maybe this already occurred in the time period 2001-2009? Could you elaborate on that with the Provoost et al (2010) and Borges and Gypens (2010) papers as references? L. 527-529: Can you really say that TA variability is more sensitive to Wadden Sea export given that the export is kept constant over the years?. To me it seems you can only make this statement for seasonal variability, not for interannual variability. L. 534-535: Why is Riveff not taken into account for the budget? Also since you seem to refer to it later in the text (i.e. L. 543-544 "3% were due to river input Riveff of TA", and L 558, "effective river loads") If there is a good reason, you need to explain this. L. 541: Why only use non-leap years? You miss two of the nine years in your data set by doing so, and it may create unintentional bias. Also, you can easily correct for it (data/91*90 for the first three months). Again, this is a really odd choice that I don't understand. L. 544-547: why discussing this if Riveff is not in the budget? Also, referring back to the relevant terms in equation 1 can aid the reader in understanding this statement. L. 552-556: Where do these percentages come from? If I understand correctly, 47% refers to 14/51 Gmol/t, but this is less than 47%. Similarly for the 59% term, which I assumed was calculated as 17/38 Gmol/t. L. 557-559: So why are the effective river loads used in this sum and not the actual river loads, which are – apparently – used in the rest of the budget? The construction of the budget and the choices made really need a better explanation. L. 571-587: What is miss in this paragraph is that there is no discussion of the uncertainty related to differences between the modelled and measured TA concentrations in the North Sea. For example, if you assume that the deviation between measured and modelled TA is entirely due to

uncertainties / errors in the Wadden Sea export estimate, what is then the uncertainty in this export? L. 577: "safely" why? Are their characteristics similar enough? Explain. L. 588-606: I assumed that the ECOHAM model also calculates TA generation from the sediments in the German Bight. If not, that should then be better explained in the method section. If yes, what is the magnitude of TA generation in the sediments in the model? How does it compare to the 12.2 Gmol/y estimate of Brenner et al (2016). And can you make a similar upscaling from the result of Burt et al (2016) which was acquired using a different method? L. 611-651: My main issue with this section is that many processes highly relevant for the TA dynamics are initially not taken into account, but are required to explain your data anyway. Not only oxidation of methane, but also reoxidation of other reduced species (e.g. previously buried sulphur). You actually run into that problem when discussing your results, noticing you cannot ignore them. So, given that there are many different processes, the system is so dynamic and exposure of sediments plays such an important role, how can you be sure that the TA/DIC ratio is a reliable metric for the message you want to convey with respect to aerobic and anaerobic degradation? I also miss a discussion on the relevant time scale of processes in relation to your results. L. 615-616: I don't think the change in DIC concentration is relevant for the change in TA. TA is reduced during the oxidation of ammonium to nitrate, which consumes acid but doesn't affect the DIC concentration. The impact of aerobic organic matter degradation on TA is minor and only comes from the production of ammonium and phosphate. The changes in DIC obviously impact the TA/DIC ratio, but not the generation or consumption of TA. L. 616: The TA/DIC ratio of denitrification is not 1 but 0.8. See e.g. R5 in Table 1 of Rassmann et al (2020). L. 619-620: And what about the sulphur dynamics? E.g. when previously buried reduced sulphur becomes exposed and reoxidised. You need to mention that here already, not only later at L. 624 L. 624: An example of what I wrote above: the ratio becomes negative, but in fact lower than -0.16, so this means that something else besides aerobic decomposition must explain this. You use reoxidation of pyrite, which consumes 2 mol of TA per mol of S oxidised. Besides this process, also other processes can affect the TA/DIC ratio at the

same time. So how can you tell the relative importance of all of them? L. 633: Another example: here you need the processes you initially neglected to explain your results. L. 640-641: How can you know this negative ratio does not result from reoxidation of reduced species? L. 647-648: Finally a mention of time scales, but please cite Hu and Cai (2011) and/or Gustafsson et al (2019) here. L. 674: The role of allochthonous nitrate is merely discussed in the rest of the manuscript and needs to be elaborated on in the discussion. L. 676-689: This "outlook" section is such a large part of the conclusions, but it isn't even a result of your study. I'd suggest to either present it as an "outlook" subsection within your conclusions, or shorten it such that your conclusions actually reflect your manuscript. L. 681-684: Why? Explain. L. 713: how was this value estimated? What is the uncertainty and how does this uncertainty impact your budget? L. 756-757: These dates actually fall in autumn and spring, not in winter and summer. Table A3: What is the time span of these data? What is the variation? (s.d. for the mean as well as for the separate months)

Technical comments: L. 51: 'regional' is stated twice, please remove the second mentioning L. 63-64: Ben-Yaakov (1973) also is a seminal paper to mention in this context L. 75: change "Netherland" to "the Netherlands" L. 82: it seems that Brenner et al. (2016) and Burt et al. (2016) can also be mentioned here L. 93: The domain of which model? ECOHAM? Should also be clear to readers unfamiliar with Pätsch et al. (2010) L. 97-100: I only understood this when reading the second time. Perhaps rephrase. Also, make clear you use measurements to calculate these box averages. L. 98: "water column" point? grid cell? L. 133: a 1996 reference is used for data from 2001-2009? L. 144: "below" where exactly? L. 159: "monthly mean concentrations" also for the years 2001-2009? L. 173-176: please provide units for each of the terms introduced here. L. 182: add direct link to Pangaea reference. L. 183: volume of Exetainer? L. 184: volume of bottle? L. 185: how much HgCl2 added? L. 188: which batch of CRM was used? L. 189: what were the accuracy and precision? L. 248: add "the model" to "FVCOM" L. 251: why not add E1, N1, etc to Fig 1 for clarity, rather than this description? L. 253: "overall" i.e. cumulative? L. 263: "table 4". Also in Table 5, although

I would suggest to merge both tables. L. 280: "TA" add "surface-water" L. 289: add validation box to Fig 5, possibly also to Figs. 3 and 4. L. 291: "standard variation" don't you mean "standard deviation"? L. 298: rephrase to "the model underestimated TA", passive tense seems odd here L. 299-300: change to "the Dutch Frisian Islands" L. 369: "TA-concentration" remove hyphen. L. 395: change to "were also" L. 460: "this would result in an increased TA concentration of 1 umol/kg" L. 487: "shift the balance" in which direction / with which result? L. 536: "highest variability" in an absolute or a relative sense? L. 554: replace "smaller" by "less" L. 611-612: add "based on measured concentrations and modelled water fluxes" L. 614-615: add Brenner et al (2016 as reference) L. 627: add the TA/DIC ratio of this process (-2). L. 645: don't you mean "organoclastic"? L. 648: add "leading" between "re-oxidised" and "to" Tables: They are presented in the text in the order Table 6 – Table 1 – Table 3 – Table 4 – Table 5 – Table 2. Please change to a logical order. The aim of Table 6 is not clear. L. 725: should be "non-leap years" L. 728: change "of" to "between" Table 2: change first header to "Wadden Sea export" for clarity L. 780: "temporally interpolated" L. 1172: "values of TA, DIC and NO3" Table A3: add horizonal lines in between the parameters for clarity. Figures: As said above, please make them as black & white friendly as possible Fig. 1 (and L. 94): green area is not visible in black & white. Maybe use dashed or dotted lines instead. Fig. 6: Add a legend in the figure itself, not only in the caption. Use striped and dotted bars to make black & white friendly Fig. 9: What is the purpose of the dots? Also this plot can easily be made black & white friendly.

References - Ben-Yaakov, S., (1973), pH BUFFERING OF PORE WATER OF RECENT ANOXIC MARINE SEDIMENTS, Limnology and Oceanography, 18, doi: 10.4319/lo.1973.18.1.0086. - Borges, Alberto V., Gypens, Nathalie, (2010), Carbonate chemistry in the coastal zone responds more strongly to eutrophication than ocean acidification, Limnology and Oceanography, 55, doi: 10.4319/lo.2010.55.1.0346. - Duarte, C.M., Hendriks, I.E., Moore, T.S. et al. Is Ocean Acidification an Open-Ocean Syndrome? Understanding Anthropogenic Impacts on Seawater pH. Estuaries and Coasts 36, 221–236 (2013). https://doi.org/10.1007/s12237-013-9594-3

- Hu, X., and Cai, W.-J. ( 2011), An assessment of ocean margin anaerobic processes on oceanic alkalinity budget, Global Biogeochem. Cycles, 25, GB3003, doi:10.1029/2010GB003859. - Gustafsson, Erik; Hagens, Mathilde; Sun, Xiaole; Reed, Daniel C.; Humborg, Christoph; Slomp, Caroline P.; Gustafsson, Bo G. (2019) Sedimentary alkalinity generation and long-term alkalinity development in the Baltic Sea. Biogeosciences, 16, 437-456, doi:10.5194/bg-16-437-2019. - Łukawska-Matuszewska, K. and Graca, B.: Pore water alkalinity below the permanent halocline in the Gdánsk Deep (Baltic Sea) – Concentration variability and benthic fluxes, Marine Chemistry, 204,49–61, https://doi.org/10.1016/j.marchem.2018.05.011, 2018 - Rassmann, J., Eitel, E. M., Lansard, B., Cathalot, C., Brandily, C., Taillefert, M., Rabouille, C., (2020) Benthic alkalinity and dissolved inorganic carbon fluxes in the Rhône River prodelta generated by decoupled aerobic and anaerobic processes. Biogeosciences, 17, 13-33, doi:10.5194/bg-17-13-2020.

---

## Referee Comment (RC2) · Anonymous Referee #2 · 30 Apr 2020

The authors examined the impacts of alkalinity export from the Wadden Sea tidal flats on the carbonate system in the southern North Sea, mainly using a digital modeling method. The topic is interesting, and the result explanation looks fair. However, I find one of their references (Pätsch et al., 2018) had demonstrated the same issue using similar or even the same digital model. So the novelty should be further refined. Also it is difficult for me to follow the manuscript, due to the poor organization of the text and the insufficient annotation of charts.

Major concerns

1. The study area is unclearly defined. As an Asian reader, Figure 1 is quite unfriendly for me. For example, where are "the German Bight as well as parts of the Danish and the Dutch coast" (lines 94-95)? Also the Wadden Sea is strange for me. After a

internet searching, I know that the Wadden Sea is the largest tidal flats system in the world (https://www.waddensea-worldheritage.org/). Since the Wadden Sea is a key area in this study, its geography should be clearly introduced to readers. I suggest that a striking section or subsection of "Study area" should be set up, after the Introduction. In this section or subsection, more geographical details and biogeochemical knowledge should be presented. Some contents of the currect subsections 2.6.1 and 2.9.1 could be integrated in the subsection of Study area.

2. The model structure and settings are unclear. A structure diagram is needed. As for the the submodule HAMSOM and the original ECOHAM model, some details are needed here, although their "details were described by Backhaus & Hainbucher (1987) and Pohlmann (1996)" (lines 107-109). At least their background and assumptions and fundamental structure and application strengths and limitations should be introduced. I wonder whether it is specially designed for the area under study. This information is also critical for general readers. In the current subsection 2.8, the authors said that "The main extension in the present study was the introduction of a prognostic treatment of TA (Pätsch et al., 2018)" (Lines 191-192). I wonder whether they give any modification on Pätsch et al. (2018) treatment.

3. How did the authors plot Figure 2? There is no relevant information (such as data source) in both the figure caption and main text. Since Figure 2 is the key to distinguish the two scenarios defined in this study, this information is a must to be clarified.

Some minor comments and suggestions

1. To avoid confusion, please unify abbreviations for North Frisian coast (N or NF), East Frisian coast (E or EF) and Jade bay (J or JB). "The respective areas 1-3" in line 350 also refers to the three regions?

2. What is "yr" in Equation (1)? Please clarify.

3. Lines 440-441: Why does this partly compensate the missing TA generation by

benthic denitrification? Please explain.

4. Line 453: Carbonate dissolution cannot be counteracted by DIC additions. DIC additions (mostly refer to free CO2) are usually in favor of carbonate dissolution.

5. Lines 534-535: Why was Riv_eff not taken into account for the budget calculations? I notice that the authors mentioned it to the later discussion, i.e. lines 557-559 "Summing up the source and sinks, Wadeen Sea exchange rates, internal processes and effective river loads resulted in highest sums in 2002 and 2003 and lowest in 2009".

6. Line 552-556: How to get those percentages (47%, 10% and 59%) based on results in Table 2. Please clarify.

7. Lines 615-616: The reduction of TA here is associated with the oxidation of ammonium to nitrate, instead of the change in DIC. Refers to Zhai et al. (2017, https://doi.org/10.1016/j.ecss.2017.08.027).

8. Lines 617-619: The TA/DIC ratio of denitrification is not 1 but 0.8. The TA/DIC ratio of sulphate reduction is not 2 but 1. Refers to Sippo et al. (2016), Are mangroves drivers or buffers of coastal acidification? Insights from alkalinity and dissolved inorganic carbon export estimates across a latitudinal transect, Global Biogeochemical Cycles, 30, 753–766, doi:10.1002/2015GB005324.

9. Lines 619-620: The authors mentioned that aerobic and anaerobic oxidation of upward diffusing methane were not considered in present study. How to relate this statement to line 633 "When sulphate reduction associated with organic matter and/or methane oxidation and pyrite burial became the dominant processes..."?

10. Lines 624-638 and Figure 9: TA/DIC ratios of $<-0.16$ in regions under study may indicate other processes than aerobic decomposition ($-0.16$) and anaerobic reaction ($>0$). What are them? I would like to suggest the authors mention more possible processes at the beginning of Section 4.4.

11. Table 2 is discussed after all other tables. Please change the order of the tables.

Additionally, Table 6 can be shifted to the Appendix.

12. Are those TA/DIC ratios presented in Figure 9 mean values in the given regions? Please clarify. Also I would like to suggest the authors compare all data in grid cells with the typical stoichiometric ratios of biogeochemical processes. Refers to Figure 4 in Sippo et al. (2016), Global Biogeochemical Cycles.

13. The "Wadden Sea tidal flats" should appear in the title.

---

## Author Comment (AC2) · 19 May 2020

Answers to the reviewers of "The impact of intertidal areas on the carbonate system of the southern North Sea" by Fabian Schwichtenberg, Johannes Pätsch, Michael Ernst Böttcher, Helmuth Thomas, Vera Winde, Kay-Christian Emeis.

The authors thank the reviewers for their interest and their detail-rich comments, which helped to improve the manuscript. We followed the suggestions of Referee #1 and re-ordered the methods part. Additionally, we re-organized the discussion section. The former section 4.4. (TA/DIC ratios) was shifted, and has now the numbering 4.2. Following the suggestions of both anonymous referees especially this section has been improved also by including valuable references.

In the following we list all reviewer comments (blue, italic) and give answers (black).

*Fabian Große*

*On a purely technical note, I would suggest that the authors adapt the color palettes they use for their figures. For many of the plots the colors are indistinguishable in greyscale and, thus, for colorblind people. This applies to Figs. 3-5, 8 and 9. Figure 6 lacks a legend.*

*I am not familiar with ODV color palettes, so I can't give advice for Figs. 3-5. The other figures seem to be produced with MatLab, for which the authors could either use the 'new' default color palette parula or any of the cmocean color palettes (https://uk.mathworks.com/matlabcentral/fileexchange/57773-cmoceanperceptually-uniform-colormaps). For Fig. 9, it would suffice to add different markers to the different lines.*

We changed the corresponding figures according to the advices. Figures 8 is now black and white, Figure 9 without colors but with different line styles.

*Anonymous Referee #1*

*General comments: This manuscript discusses the role of alkalinity export from the Wadden Sea on the carbonate system of the southern North Sea. Specifically, it aims at quantifying this export, its importance for the alkalinity budget, and the relative role of aerobic versus anaerobic processes in generating the alkalinity in the Wadden Sea.*

*The manuscript presents interesting work, which is worth publishing, but in my opinion the manuscript itself needs some work. The aim and take home messages of the work are not made very clear in either the abstract or the last paragraph of the introduction. The order and relative length of sections does not always appear logical to me. The construction of the TA budget raises some questions. Also, the manuscript is at times difficult to follow without knowing the details of previous studies, especially in the methods section. For example, the biogeochemistry in the model is merely explained. In short, the writing can be much sharper. In the specific comments I'll provide examples and suggestions.*

Thanks for the detailed review. Within the abstract we added the aims and reorganized the Methods section according to the reviewer's suggestions. The TA budget may have been misleading as the role of "effective river input" was not explained in detail. We added explanations. The biogeochemical model description has been augmented.

*Also the presentation of data can be improved. As already indicated in another comment, the figures can be improved to support black & white reading, e.g. by using dashed and dotted lines or bars. I also found the order of the tables highly confusing. If I counted correctly they are presented in the text in the order Table 6 – Table 1 – Table 3 – Table 4 – Table 5 – Table 2. Please change this in the next version.*

We improved figures 1, 3, 4, 5, 8, and 9 (old numbering) and changed the order of the tables.

*Content-wise, a major point I don't fully understand is the lack of quantification of the uncertainty in the calculated export flux of 39 Gmol TA y-1, which is such a central result of this study. I understand it is based on sparse measurements of DIC and TA concentrations in the different Wadden Sea areas but some estimate of the uncertainty with the use of equation (2) and upscaling of the results should be possible to make.*

Within the budgeting section we introduced an estimate of the uncertainty.

*Also, using a fixed value for the Wadden Sea export for the years 2001-2009 but taking into account interannual variability in the other terms of the TA budget makes it difficult to actually quantify the relative contributions for each of the years, as you also expect quite some interannual variability in the export flux. I would therefore suggest to also calculate an average TA budget for the period 2001-2009, as you did for the seasonal pattern, and mostly use that in the discussion of the budget. In my opinion, some more odd choices were made in the TA budget, such as excluding Riveff and the leap years, which need revision and/or better explanations.*

We added an averaged budget, explained the role of Riveff. The seasonal and annual averages use now all years. We left out all days with date 29 February.

*Distinguishing between anaerobic and aerobic processes generating TA in the Wadden Sea appears somewhat problematic, since many processes highly relevant for the TA dynamics are initially not taken into account, but are required to explain your data anyway. Take for example oxidation of methane, but also reoxidation of other reduced species (e.g. previously buried sulphur). Given that there are many different processes, the system is so dynamic and exposure of sediments plays such an important role, how can you be sure that the TA/DIC ratio is a reliable metric for the message you want to convey with respect to aerobic and anaerobic degradation? I also miss a discussion on the relevant time scale of processes here.*

This is certainly a valid statement, and processes are in all likelihood comingled. What struck us is the fact that the export ratios are statistically different in the three sectors and that these differences appear to be related to morphological features (areas and strait/channel geometries of the tidal basins) that have previously been invoked to explain differences in eutrophication status. The predominance of organic carbon mineralization by aerobic

processes in the North Frisian and of anaerobic processes in West Frisian and Jade Bay is well documented in the observations, even though the data base is limited. The model may be ignorant of these differences, but in our opinion the regional variation in processes is an interesting addition to this manuscript.

*The role of the data presented in Table 6 is not entirely clear to me; section 2.7 also does not really seem to match with the rest of the manuscript, especially since in the end they are not used for the TA budget. And if these data have indeed been published elsewhere, it seems unnecessary to publish them here as well.*

We use the additional TA and DIC data (now Table 3) in the new ordered section 4.1. and 4.2. as discussion distribution. In total the data were not published elsewhere.

*Finally, I do not understand why the model was only validated with 2008 data, whereas there are also data from 2001/2002 (as discussed by the authors) and 2005 available in the CANOBA (or related) datasets. Validating with these data as well, especially with the 2001/2002 data that include multiple seasons, would strengthen the manuscript.*

As this study focuses on summer situations in the southern North Sea we added statistics for summer 2001 and 2005 in Table 4 and 5.

*Specific comments:*

*Abstract: I miss a clear aim and a concluding sentence in the abstract. The work is described in lines 23-25, but what is the underlying aim? Confirming the high TA export from the Wadden Sea? Finding out the underlying mechanisms? And similarly, what can we conclude from this work?*

We now included the sentence "Aim of this work is to reproduce the observed high summer TA concentrations in the southern North Sea and to differentiate the various sources contributing to these elevated values". Later in the abstract we show that we have reached these aim and give percentages of the different contributors to the elevated Alkalinity concentration in summer. We used the observed TA and DIC concentrations as boundary values for the model. In addition, we used these data for a coarse estimate of the different underlying degradation processes.

*L. 18-19: This sentence focuses entirely on the physics of coastal oceans, whereas a major reason for coastal acidification being different from open ocean acidification is the fact that inputs and process rates are much higher. See e.g. Duarte et al (2013) (reference added at the end)*

We changed the first sentence of the abstract: "The coastal ocean is strongly affected by ocean acidification because it is shallow, has a low volume, and is in close contact with terrestrial dynamics." In addition, we incorporated the reference Duarte et al (2013) and augmented the corresponding discussion in the introduction.

*L. 25: "sources" do you refer to concentrations or fluxes or both here? As the sources are calculated based on measured concentrations and modelled exchange rates, so they are not truly observed.*

The basic observation data were DIC and TA concentration. We changed this in the text.

*L. 34: can you briefly elaborate what you mean with "weak meteorological blocking conditions"? I'd suggest to paraphrase this in the abstract and explain the term later in the manuscript (e.g. at L. 365 where it appears again).*

We followed your suggestions and explained this concept at the end of 3.3.

*L. 37-38: does the 'net transport' have a particular direction?*

No, it does not have a defined direction. But, normally when westerly winds prevail we have an anti-clockwise circulation like figure 7 August 2008 shows.

*L. 38: "internal production" is this the gross or net TA production? Of the water column only, or of the combined water and sediment system?*

It is the net TA production including benthic and pelagic processes. "net" is added in the text.

*L. 42-43: "anaerobic degradation dominated" with which pathway?*

The pathways are denitrification and sulphate- and iron – reduction. We added this in the text.

*L. 54-58: Add the suggested mechanisms for the observation in the Provoost article (i.e. change in production-respiration balance). Also refer to Duarte et al (2013) here, as they summarise many important processes impacting pH balance. Now it seems as if only biogeochemical processes in the sediment are important, which is obviously not the case in many coastal areas.*

Thanks. Both articles and their arguments are used now.

*L. 64-67: Also the Baltic Sea is a key example of this; see e.g. Łukawska-Matuszewska (2017) and Gustafsson et al (2019)*

We incorporated these articles and their issues.

*L. 87-90: If this is the aim of the work it is not written very clearly. What is the general aim? What is the key research question that will be addressed? Is there a certain time period associated with this or is your aim more general? What do you hypothesise? This section really needs some work.*

We changed the text accordingly.

*Methods: the division of subsections seems oddly chosen. 11 subsections is way too many and yet details are lacking. Subsection 2.7 seems unnecessary and subsection 2.8 way too short. I would suggest to merge some of the subsections, or use a third level instead.*

We followed the reviewers suggestion and introduced a third level. Old section 2.8 was augmented.

*L. 158: NO3 data are also presented in this table, but not mentioned here or much discussed in the manuscript.*

We use the nitrate data, which are indicated in Table A3 and mentioned this in the text: "Monthly mean concentrations of nitrate, TA and DIC were added for the Dutch rivers (www.waterbase.nl) and for the German river Elbe (Amann et al., 2015)."

*L. 178-197: The purpose of this subsection (and of this data in general) is not entirely clear to me and someone it feels like they were added last-minute. Maybe because they are the first presented but referred to as Table 6. Can the authors please elaborate on why this data were added? Especially since in the end they are not used for the calculation of the TA budget. Also, if the data are presented elsewhere (as L. 180-181 seems to suggest, "reported for completeness only"), there is no need to discuss the methodology here.*

As already mentioned above the additional TA and DIC data are used in the new ordered section 4.1. and 4.2. as discussion distribution. In total the data were not published elsewhere.

*L. 190-197: if this is the novel part of the manuscript, as seems to be suggested by L. 191 ("the main extension in the present study"), then it really needs a more detailed explanation. Also readers not familiar with Pätsch et al. (2018) need to understand this. Explain which biogeochemical processes are involved and where they take place (water column or sediment or both). How are sediments included in general? A brief mention to Wolf-Gladrow et al. (2007) is not sufficient, as I don't think that the exact same components are included in this study.*

The main model extension was indeed the prognostic treatment of TA, which was introduced by Pätsch et al. (2018). We changed this section accordingly and give additional information on pelagic and benthic processes which affect TA variations.

*L. 195: "nutrient dynamics" i.e. productivity and decomposition?*

We now say: "The pelagic biogeochemical part is driven by planktonic production and respiration, formation and dissolution of calcite, pelagic and benthic degradation and remineralisation, and also by atmospheric deposition of reduced and oxidised nitrogen."

*L. 196: what about atmospheric sulphur deposition?*

We use only nitrogen deposition. The model does not treat Sulphur dynamics explicitly.

*L. 241-242: I don't understand why the time lag is the reason for the lack of statistical analysis. The low number of observations is the reason. If this is what you mean, then please paraphrase this section.*

We rephrased accordingly: "Due to the low number of concentration measurements a statistical analysis of uncertainties of ΔTA and ΔDIC was not possible."

*L. 257-258: so scenario A has no Wadden Sea export but the same internal biogeochemistry as scenario B? Or is it the same as previously published implementations of the model? Please elaborate.*

*We rephrased this section and write now: "For scenario B we used the same model configuration as for scenario A and additionally implemented Wadden Sea export rates of TA and DIC as described above."*

*L. 259-260: So the data in Fig. 2 are calculated according to Eq. 2? And then multiplied by the area of what? Summed area of the grid cells? Please explain.*

*We sharpened the description of the export rates: "The respective Wadden Sea export rates (Fig. 2) are calculated by the temporal integration of the product of wad_sta and wad_exc over one month."*

*L. 268-269: This belongs in the introduction. Mention all aims clearly there in the last paragraph.*

We mention our aims now in the abstract and in the introduction and omit this sentence at this position.

*Results: Sections 3.1. and 3.2: why do you discuss the validation of DIC an TA separately? It seems more logical to me that, because they are so connected, you can also discuss them at the same time. That would also shorten this relatively long section.*

We decided to stay with the separation of TA and DIC validation within two separate subchapters as we think that it will be confusing to discuss both parameters together. The structure of both sections is very clear and helps the reader to compare the corresponding features.

*L. 303: what do you mean by "the standard deviation improved"? In scenario A it was lower, i.e. 7 umol/kg. Or do you mean to say that the standard deviation comes closer to that of the observations?*

We now write: "Compared to scenario A the simulation of scenario B was closer to the observations in terms of RMSE (18 µmol TA kg$^{-1}$) and the standard deviation (Stdv = 22 µmol TA kg$^{-1}$). Also the correlation (r = 0.86) improved (Table 4)."

*L. 348: rather than mentioning high flushing times, I would paraphrase to focus on the low water renewal or long mean residence time.*

We changed this sentence: "They were chosen to highlight the pattern in summer 2003 with one of the highest flushing times (lowest water renewal times), and that in 2008 corresponding to one of the lowest flushing times (highest water renewal times)." In addition, we added the definition of flushing times and the relation to renewal times in section 2.1.2.

*Same in L. 351-352: I would paraphrase to say that highest inflows occurred in winter.*

We changed this sentence: "Flushing times (rounded to integer values) were consistently higher in summer than in winter, meaning that highest inflow occurred in winter."

*L. 354-356: Can't you use a metric to correct for this feature, allowing fairer comparisons?*
One of the main statement within our manuscript is that one of the reasons that our annual

Wadden Sea TA export differs from that given by Thomas et al. (2009), is, that the latter authors assumed constant flushing times (6 weeks). We show that for different years and different seasons the flushing times differ strongly. In order to compare the water exchange rate in weeks (Thomas et al., 2009) with our flushing times we stick to this term. We admit that the comparison of flushing times of different basins is not fair, but here we focus on temporal comparisons.

*Sections 3.4 and 3.5: Again, I'd suggest to merge these two.*

We merged these two sections, stayed with the two-block structure and created an additional common analysis.

*L. 385-386: How is this for TA?*

As mentioned above, in an additional section we compare both TA and DIC structures.

*Discussion: I'd suggest to change the order. Subsection 4.2, which is to a large extent an outlook to the future and partly relies on te TA budget, is much more logical as final subsection.*

*Subsection 4.5 is connected to the Wadden Sea data and it seems logical for it to immediately follow subsection 4.1.*

We followed your suggestions. The order is now:
4.1 Uncertainties
4.2 TA/DIC ratios
4.3 TA budget
4.4 The impact of export

*L. 394-396: I don't understand why the model was only validated with 2008 data, whereas there are also data from 2001/2002 (as discussed by the authors) and 2005 available in the CANOBA (or related) datasets. Validating with these data as well, especially with the 2001/2002 data that include multiple seasons, would strengthen the manuscript*

We added the statistical validation for summer 2001 and 2005 (Tables 4 and 5)

*L. 397-399: Move to last paragraph of introduction, this aim is not in there yet.*

We defined our aims in the abstract and in the introduction. We cancelled the listing of these more technical aims in this section.

*L. 416-417: This is quite a simplified statement; the temporal and spatial scale you consider are highly relevant for whether this is the case and for which processes associated with anaerobic decomposition this is relevant. See e.g. Hu and Cai (2011) and Gustafsson et al (2019) for discussions on this.*

Thanks, we incorporated these sentences: "On longer time scales TA can only be generated by processes that involve permanent loss of anaerobic remineralisation products (Hu and Cai, 2011). A second precondition is the nutrient availability to produce organic matter, which in turn serves as necessary component of anaerobic decomposition (Gustafsson et al., 2019)."

*L. 424-428: Could you add some suggestions for improvement?*

We added: "More measurements distributed with higher resolution over the annual cycle would clearly improve our estimates."

*L. 431-453: Why is S burial not discussed here? This seems highly relevant, especially on the longer term.*

We discuss this now: "Sulphate reduction (not modelled here) also contributes to alkalinity generation. On longer time scales the net effect is vanishing as the major part of the reduced components are immediately re-oxidized in contact with oxygen. Iron- and sulphate - reduction generates TA but only their reaction product iron sulphide (essentially pyrite) conserves the reduced components from re-oxidation. As the formation of pyrite consumes TA, the TA contribution of iron reduction in the North Sea is assumed to be small and to balance that of pyrite formation (Brenner et al., 2016)."

*L. 435-437: So external NO3 inputs are not relevant for benthic denitrification?*

In our model $N_2$ production due to denitrification is recharged by OM in relation to the benthic oxygen consumption (Seitzinger and Giblin, 1996). So external nitrate inputs implicitly stimulate this denitrification as the amount of benthic oxygen consumption is stimulated by primary production and the subsequent provision of OM to the benthic realm.

*L. 440-441: Why does this compensate the TA generation? Please explain.*

We added: "This amount of nitrate would not fully be available for primary production if parts of it would be consumed by denitrification."

*L. 444-446: But how high and relevant is the deposition of these inputs for the TA budget? This is a very qualitative paragraph.*

We added: "The average decrease within 6 years is about 0.4 Gmol TA $yr^{-1}$, whereas the average increase within 3 years is only 0.1 Gmol TA $yr^{-1}$."

*L. 470-472: So what is the aim of adding these data to the manuscript if they are published elsewhere and not taken into account for the budget?*

In total this table is not published elsewhere. In the context of this study it serves supporting the discussion of uncertainties. The data show large variabilities.

*L. 478-479: How has the sensitivity of DIC to modelled biology been confirmed?*

Figure 8c shows the summer drawdown of DIC which is due to primary production. Also Lorkowski et al., (2012) showed this relation.

*L. 479-480: The reader doesn't know this yet as the TA budget has not yet been presented. I'd suggest to change the order.*

Yes, order has been changed.

*L. 494-496: Thus slower exchange, what would be the effect on the TA export?*

We added: "This would decrease TA export into the North Sea."

*L. 497-499: Why?*

We added: "In fine grained silt diffusive transport plays a key role, while in the upper layer of coarse (sandy) sediments advection is the dominant process."

*L. 504-506: On which time scale? Maybe this already occurred in the time period 2001-2009? Could you elaborate on that with the Provoost et al (2010) and Borges and Gypens (2010) papers as references?*

We added: "pH values in Dutch coastal waters decreased from 1990 to 2006 drastically. Changes in nutrient variability were identified as possible drivers (Provoost et al., 2010), which is consistent with model simulations by Borges and Gypens (2010)."

*L. 527-529: Can you really say that TA variability is more sensitive to Wadden Sea export given that the export is kept constant over the years?. To me it seems you can only make this statement for seasonal variability, not for interannual variability.*

*You are right. We induce interannual variability of TA and DIC concentrations mainly by interannual variability of the physical environment. So, we suppose that we underestimate the interannual variability of TA and DIC concentrations. We added "The interannual variability of the model results are mainly driven by the physical prescribed environment."*

*L. 534-535: Why is Riveff not taken into account for the budget? Also since you seem to refer to it later in the text (i.e. L 543-544 "3% were due to river input Riveff of TA", and L 558, "effective river loads") If there is a good reason, you need to explain this.*

We added: "This is explained in the Method Section "River Input"." The reason why we introduced "effective river input" is to understand the role of river input on concentration changes.

*L. 541: Why only use non-leap years? You miss two of the nine years in your data set by doing so, and it may create unintentional bias. Also, you can easily correct for it (data/91*90 for the first three months). Again, this is a really odd choice that I don't understand.*

OK. We recalculated the averages using all years. The data from all 29 of February were cancelled.

*L. 544-547: why discussing this if Riveff is not in the budget? Also, referring back to the relevant terms in equation 1 can aid the reader in understanding this statement.*

We hope that the concept of $Riv_{eff}$ is clear now.

*L. 552-556: Where do these percentages come from? If I understand correctly, 47% refers to 14/51 Gmol/t, but this is less than 47%. Similarly for the 59% term, which I assumed was calculated as 17/38 Gmol/t.*

You are right, we corrected these percentages. We also added: "Note that these percentages are related to the sum of the absolute values of the budgeting terms."

*L. 557-559: So why are the effective river loads used in this sum and not the actual river loads, which are – apparently – used in the rest of the budget? The construction of the budget and the choices made really need a better explanation.*

Here we discuss the sinks and sources, which change the TA concentration necessarily. Net transport and actual river load may change the concentration depending on the concentration of added or leaving water. Please keep in mind that the volume may also change over time due to the free surface elevation. We added: "For the consideration of TA concentration variations we excluded net transport and actual river loads, because these fluxes are diluted and do not necessarily change the TA concentrations."

*L. 571-587: What is miss in this paragraph is that there is no discussion of the uncertainty related to differences between the modelled and measured TA concentrations in the North Sea. For example, if you assume that the deviation between measured and modelled TA is entirely due to uncertainties / errors in the Wadden Sea export estimate, what is then the uncertainty in this export?*

We added an estimate of the uncertainty of the additional Wadden Sea TA export flux, and added: "Table 4 shows that our scenario B underestimates the observed TA concentration by about 5.1 µmol kg$^{-1}$ in 2008. Scenario A has lower TA concentration than scenario B in the validation area. The difference is about 11 µmol kg$^{-1}$. This means that the Wadden Sea export of 39 Gmol TA yr$^{-1}$ results in a concentration difference of 11 µmol kg$^{-1}$. Assuming linearity, the deviation between scenario B and the observations (5.1 µmol kg$^{-1}$) would be compensated by an additional Wadden Sea export of about 18 Gmol TA yr$^{-1}$. If we assume that the deviation between observation and scenario B is entirely due to uncertainties or errors in the Wadden Sea export estimate, then the uncertainty of this export is 18 Gmol TA yr$^{-1}$."

*L. 577: "safely" why? Are their characteristics similar enough? Explain.*

The Area of the West Frisian Wadden Sea is much larger than the area of the East Frisian Wadden Sea. We changed this sentence: "Due to the scarcity of data, the West Frisian Wadden Sea was not considered in the simulations, but, as the western area is much larger than the eastern area, the amount of exported TA from that area can be assumed to be in the same range as from the East Frisian Wadden Sea (10 to 14 Gmol TA yr$^{-1}$)."

*L. 588-606: I assumed that the ECOHAM model also calculates TA generation from the sediments in the German Bight. If not, that should then be better explained in the method section. If yes, what is the magnitude of TA generation in the sediments in the model? How does it compare to the 12.2 Gmol/y estimate of Brenner et al (2016). And can you make a similar upscaling from the result of Burt et al (2016) which was acquired using a different method?*

In section 2.1.3 we mention "Benthic denitrification and other anaerobic processes have no impact on pelagic TA concentrations in this model version. Only the carbonate ions from benthic calcite dilution and the remineralisation products ammonium and phosphate which

enter the pelagic system across the benthic-pelagic interface alter the pelagic TA concentration."

*L. 611-651: My main issue with this section is that many processes highly relevant for the TA dynamics are initially not taken into account, but are required to explain your data anyway. Not only oxidation of methane, but also reoxidation of other reduced species (e.g. previously buried sulphur). You actually run into that problem when discussing your results, noticing you cannot ignore them. So, given that there are many different processes, the system is so dynamic and exposure of sediments plays such an important role, how can you be sure that the TA/DIC ratio is a reliable metric for the message you want to convey with respect to aerobic and anaerobic degradation? I also miss a discussion on the relevant time scale of processes in relation to your results.*

Following the suggestions and queries of the two reviewers, we re-organised the entire discussion of the processes relevant to the regional differences in TA/DIC ratios, which is now part of section 4.2. In the course of this re-organisation , we corrected the stoichiometric TA/DIC ratios generated by the putative processes, included relevant references suggested by reviewers, and generally tried to be less assertive with respect to putative sources of TA and DIC.

*L. 615-616: I don't think the change in DIC concentration is relevant for the change in TA. TA is reduced during the oxidation of ammonium to nitrate, which consumes acid but doesn't affect the DIC concentration. The impact of aerobic organic matter degradation on TA is minor and only comes from the production of ammonium and phosphate. The changes in DIC obviously impact the TA/DIC ratio, but not the generation or consumption of TA.*

We changed the text accordingly, and write now: "Candidate processes are numerous and the export ratios certainly express various combinations, but the most quantitatively relevant likely are aerobic degradation of organic material (resulting in a reduction of TA due to nitrification of ammonia to nitrate with a TA / DIC ratio of -0.16)".

*L. 616: The TA/DIC ratio of denitrification is not 1 but 0.8. See e.g. R5 in Table 1 of Rassmann et al (2020).*

We changed accordingly and added this reference.

*L. 619-620: And what about the sulphur dynamics? E.g. when previously buried reduced sulphur becomes exposed and reoxidised. You need to mention that here already, not only later at L. 624*

We added: "Other processes are aerobic (adding only DIC) and anaerobic (TA/DIC ratio of 2) oxidation of upward diffusing methane, oxidation of sedimentary sulphides upon resuspension into an aerated water column (no effect on TA/DIC) followed by oxidation of iron (adding TA), and nitrification of ammonium (consuming TA)."

*L. 624: An example of what I wrote above: the ratio becomes negative, but in fact lower than -0.16, so this means that something else besides aerobic decomposition must explain this. You use reoxidation of pyrite, which consumes 2 mol of TA per mol of S oxidised. Besides this*

*process, also other processes can affect the TA/DIC ratio at the same time. So how can you tell the relative importance of all of them?*

See above

*L. 633: Another example: here you need the processes you initially neglected to explain your results.*

See above

*L. 640-641: How can you know this negative ratio does not result from reoxidation of reduced species?*

See above

*L. 647-648: Finally a mention of time scales, but please cite Hu and Cai (2011) and/or Gustafsson et al (2019) here.*

We added the citation of Hu and Cai (2011): "Taken at face value, the resulting negative ratio of -0.4 implicates re-oxidation of pyrite, normally on timescales of early diagenesis thermodynamically stable (Hu and Cai, 2011), possibly promoted by increasing wind forces and associated aeration and sulphide oxidation of anoxic sediment layers (Kowalski et al., 2013)."

*L. 674: The role of allochthonous nitrate is merely discussed in the rest of the manuscript and needs to be elaborated on in the discussion.*

We omitted this sentence.

*L. 676-689: This "outlook" section is such a large part of the conclusions, but it isn't even a result of your study. I'd suggest to either present it as an "outlook" subsection within your conclusions, or shorten it such that your conclusions actually reflect your manuscript.*

We renamed this chapter: "Conclusions and Outlook"

*L. 681-684: Why? Explain.*

The trend is expected to continue due to the European Water Framework Directive, which requires less nitrogen input. We added: "(European Water Framework Directive)"

*L. 713: how was this value estimated? What is the uncertainty and how does this uncertainty impact your budget?*

In section 4.1 Uncertainties of .. exchange we added: "We calculated the sensitivity of our annual TA export rates on uncertainties of the Δ-values of Table 1. As the different areas North- and East Frisian Wadden Sea and Jade Bay has different exchange rates of water, for each region the uncertainty of 1 μmol kg$^{-1}$ in ΔTA at all times has been calculated. The East Frisian Wadden Sea export would differ by 0.84 Gmol TA yr$^{-1}$, the Jade Bay export by 0.09 Gmol TA yr$^{-1}$ and the North Frisian export by 3 Gmol TA yr$^{-1}$."

*L. 756-757: These dates actually fall in autumn and spring, not in winter and summer. Table A3: What is the time span of these data? What is the variation? (s.d. for the mean as well as for the separate months)*

Due to the scarcity of the original data, statistic for individual months was not possible. In Table A3 we added SD (standard deviation) of the monthly means. We added in chapter 2.2.2: "The Dutch data were observed in the years 2007 – 2009. The river Elbe data stem from the years 2009 – 2011."

*Technical comments:*

*L. 51: 'regional' is stated twice, please remove the second mentioning*

Done

*L. 63-64: Ben-Yaakov (1973) also is a seminal paper to mention in this context*

We incorporated the reference

*L. 75: change "Netherland" to "the Netherlands"*

Done

*L. 82: it seems that Brenner et al. (2016) and Burt et al. (2016) can also be mentioned here*

We incorporated the references

*L. 93: The domain of which model? ECOHAM? Should also be clear to readers unfamiliar with Pätsch et al. (2010)*

OK, we changed the text accordingly.

*L. 97-100: I only understood this when reading the second time. Perhaps rephrase. Also, make clear you use measurements to calculate these box averages.*

We changed the text: "For the calculation of box averages of DIC and TA a bias towards the deeper areas with more volume and more data should be avoided. Therefore, each water column covered with data within the validation area delivered one mean value, which is calculated by vertical averaging. These mean water column averages were horizontally interpolated onto the model grid. After this procedure average box values were calculated."

*L. 98: "water column" point? grid cell?*

It is the water column, because at this stage we have observational data at arbitrary locations.

*L. 133: a 1996 reference is used for data from 2001-2009?*

Yes, this is the official reference provided by NCEP/NCAR. In addition, we added in the acknowledgements "We used NCEP Reanalysis data provided by the NOAA/OAR/ESRL PSL, Boulder, Colorado, USA, from their Web site at https://psl.noaa.gov/"

*L. 144: "below" where exactly?*

We introduced this description in the following section.

*L. 159: "monthly mean concentrations" also for the years 2001-2009?*

We added: "The Dutch river data were observed in the years 2007 – 2009. The river Elbe data were taken in the years 2009 – 2011. These concentration data were prescribed for all simulation years as mean annual cycle.*"*

*L. 173-176: please provide units for each of the terms introduced here.*

Done

*L. 182: add direct link to Pangaea reference.*

We now give the DOI

*L. 183: volume of Exetainer?*

12 ccm

*L. 184: volume of bottle?*

250 ccm

*L. 185: how much HgCl2 added?*

*100* µl

*L. 188: which batch of CRM was used?*

It was batch 102. We clarified this in the text.

*L. 189: what were the accuracy and precision?*

We added: "Standard deviations for DIC and TA measurements were better than +/-2 and +/-10 µmol/kg, respectively."

*L. 248: add "the model" to "FVCOM"*

Done

*L. 251: why not add E1, N1, etc to Fig 1 for clarity, rather than this description?*

Done

*L. 253: "overall" i.e. cumulative?*

The mean daily runoff.

*L. 263: "table 4". Also in Table 5, although I would suggest to merge both tables.*

The tables were augmented by the years 2001 and 2005. To keep clarity we did not merge the tables.

*L. 280: "TA" add "surface-water"*

We changed the first sentence of this section: "The results of scenarios A and B were compared with observations of TA in August 2008 (Salt et al., 2013) for surface water."

*L. 289: add validation box to Fig 5, possibly also to Figs. 3 and 4.*

Done

*L. 291: "standard variation" don't you mean "standard deviation"?*

Yes. We changed it in the text.

*L. 298: rephrase to "the model underestimated TA", passive tense seems odd here*

Done

*L. 299-300: change to "the Dutch Frisian Islands"*

Done

*L. 369: "TA-concentration" remove hyphen.*

Done

*L. 395: change to "were also"*

Done

*L. 460: "this would result in an increased TA concentration of 1 umol/kg"*

Done

*L. 487: "shift the balance" in which direction / with which result?*

This is discussed in the following sentences.

*L. 536: "highest variability" in an absolute or a relative sense?*

In an absolute sense. We added "absolute".

*L. 554: replace "smaller" by "less"*

Done

*L. 611-612: add "based on measured concentrations and modelled water fluxes"*

We changed in the figure caption for figure 2: "The export rates were calculated for DIC and TA based on measured concentrations and simulated water fluxes."

*L. 614-615: add Brenner et al (2016 as reference)*

Done

*L. 627: add the TA/DIC ratio of this process (-2).*

Following the suggestions and queries of the two reviewers, we re-organised the entire discussion of the processes relevant to the regional differences in TA/DIC ratios, which is now part of section 4.2. In the course of this re-organisation , we corrected the stoichiometric TA/DIC ratios generated by the putative processes, included relevant references suggested by reviewers and generally tried to be less assertive with respect to putative sources of TA and DIC.

*L. 645: don't you mean "organoclastic"?*

Thanks, we incorporated this term: ".. and anaerobic processes related to sulphate reduction of organoclastic material (TA / DIC ratio of 1)."

*L. 648: add "leading" between "re-oxidised" and "to"*

Done

*Tables: They are presented in the text in the order Table 6 – Table 1 – Table 3 – Table 4 – Table 5 – Table 2. Please change to a logical order. The aim of Table 6 is not clear.*

Done

*L. 725: should be "non-leap years"*

The exclusion of leap years is cancelled.

*L. 728: change "of" to "between"*

Done

*Table 2: change first header to "Wadden Sea export" for clarity*

Done

*L. 780: "temporally interpolated"*

Done

*L. 1172: "values of TA, DIC and NO3"*

Done

*Table A3: add horizonal lines in between the parameters for clarity.*

Done

*Figures: As said above, please make them as black & white friendly as possible Fig. 1 (and L. 94): green area is not visible in black & white. Maybe use dashed or dotted lines instead.*

We changed the green box into magenta. The dashed line is indicated within the text. This makes the magenta validation box identifiable.

*Fig. 6: Add a legend in the figure itself, not only in the caption. Use striped and dotted bars to make black & white friendly*

Done. The order of the bars are indicated now.

*Fig. 9: What is the purpose of the dots? Also this plot can easily be made black & white friendly.*

Done

*References - Ben-Yaakov, S., (1973), pH BUFFERING OF PORE WATER OF RECENT ANOXIC MARINE SEDIMENTS, Limnology and Oceanography, 18, doi: 10.4319/lo.1973.18.1.0086. –*

*Borges, Alberto V., Gypens, Nathalie, (2010), Carbonate chemistry in the coastal zone responds more strongly to eutrophication than ocean acidification, Limnology and Oceanography, 55, doi: 10.4319/lo.2010.55.1.0346. –*

*Duarte, C.M., Hendriks, I.E., Moore, T.S. et al. Is Ocean Acidification an Open- Ocean Syndrome? Understanding Anthropogenic Impacts on Seawater pH. Estuaries and Coasts 36, 221–236 (2013). https://doi.org/10.1007/s12237-013-9594-3 -*

*Hu, X., and Cai, W.-J. ( 2011), An assessment of ocean margin anaerobic processes on oceanic alkalinity budget, Global Biogeochem. Cycles, 25, GB3003, doi:10.1029/2010GB003859. –*

*Gustafsson, Erik; Hagens, Mathilde; Sun, Xiaole; Reed, Daniel C.; Humborg, Christoph; Slomp, Caroline P.; Gustafsson, Bo G. (2019) Sedimentary alkalinity generation and long-term alkalinity development in the Baltic Sea. Biogeosciences, 16, 437-456, doi:10.5194/bg-16-437-2019. –*

*Łukawska- Matuszewska, K. and Graca, B.: Pore water alkalinity below the permanent halocline in the Gdánsk Deep (Baltic Sea) – Concentration variability and benthic fluxes, Marine Chemistry, 204,49–61, https://doi.org/10.1016/j.marchem.2018.05.011, 2018 –*

*Rassmann, J., Eitel, E. M., Lansard, B., Cathalot, C., Brandily, C., Taillefert, M., Rabouille, C., (2020) Benthic alkalinity and dissolved inorganic carbon fluxes in the Rhône River prodelta generated by decoupled aerobic and anaerobic processes. Biogeosciences, 17, 13-33, doi:10.5194/bg-17-13-2020.*

*Anonymous Referee #2*

*The authors examined the impacts of alkalinity export from the Wadden Sea tidal flats on the carbonate system in the southern North Sea, mainly using a digital modeling method. The topic is interesting, and the result explanation looks fair. However, I find one of their references (Pätsch et al., 2018) had demonstrated the same issue using similar or even the same digital model. So the novelty should be further refined. Also it is difficult for me to follow the manuscript, due to the poor organization of the text and the insufficient annotation of charts.*

We thank the reviewer for the interest and the helpful review. Pätsch et al. (2018) used indeed a similar model. Only the benthic module has been exchanged. The novelty of the study in hand is the scientific question: Which TA contributors cause the elevated TA concentration elevations in the southern North Sea and German Bight during summer? How large are the different fluxes? These aims are now clearly defined in the abstract and in a broader manner within the introduction. Reviewer#1 also found need of improvement of the general structure. We reorganized the Methods and Discussion chapter. Also, the annotation of charts has been improved.

*Major concerns*

*1. The study area is unclearly defined. As an Asian reader, Figure 1 is quite unfriendly for me. For example, where are "the German Bight as well as parts of the Danish and the Dutch coast" (lines 94-95)? Also the Wadden Sea is strange for me. After a internet searching, I know that the Wadden Sea is the largest tidal flats system in the world (https://www.waddensea-worldheritage.org/). Since the Wadden Sea is a key area in this study, its geography should be clearly introduced to readers. I suggest that a striking section or subsection of "Study area" should be set up, after the Introduction. In this section or subsection, more geographical details and biogeochemical knowledge should be presented. Some contents of the currect subsections 2.6.1 and 2.9.1 could be integrated in the subsection of Study area.*

We added a map into Figure 1, where all these areas are identified. We augmented the third section of the introduction where the area of the Wadden Sea was described. We added: "During low tide about 50 % of the area are falling dry (van Beusekom et al., 2019). Large rivers discharge nutrients into the Wadden Sea, which in turn shows a high degree of eutrophication, aggravated by mineralisation of organic material imported into the Wadden Sea from the open North Sea (van Beusekom et al., 2012)."

*2. The model structure and settings are unclear. A structure diagram is needed. As for the the submodule HAMSOM and the original ECOHAM model, some details are needed here, although their "details were described by Backhaus & Hainbucher (1987) and Pohlmann (1996)" (lines 107-109). At least their background and assumptions and fundamental structure and application strengths and limitations should be introduced. I wonder whether it is specially designed for the area under study. This information is also critical for general*

*readers. In the current subsection 2.8, the authors said that "The main extension in the present study was the introduction of a prognostic treatment of TA (Pätsch et al., 2018)" (Lines 191-192). I wonder whether they give any modification on Pätsch et al. (2018) treatment.*

We added a structure diagram as supplemental material. For HAMSOM we added: "It is a baroclinic primitive equation model using the hydrostatic and Boussinesq approximation. It is applied to several regional sea areas worldwide.". For this study we use the ECOHAM version of Pätsch et al. (2018). Only the benthic module is exchanged. We added: "The pelagic biogeochemical part is driven by planktonic production and respiration, formation and dissolution of calcite, pelagic and benthic degradation and remineralisation, and also by atmospheric deposition of reduced and oxidised nitrogen. All these processes impact TA. Benthic denitrification and other anaerobic processes have no impact on pelagic TA concentrations in this model version. Only the carbonate ions from benthic calcite dilution and the remineralisation products ammonium and phosphate which enter the pelagic system across the benthic-pelagic interface alter the pelagic TA concentration.".

*3. How did the authors plot Figure 2? There is no relevant information (such as data source) in both the figure caption and main text. Since Figure 2 is the key to distinguish the two scenarios defined in this study, this information is a must to be clarified*

In the new chapter "2.3 The Wadden Sea" the data behind Fig. 2 are described in detail. In chapter 2.1.3. we added: "The respective Wadden Sea export rates (Fig. 2) are calculated by the temporal integration of the product of wad_sta and wad_exc over one month (see equation 2)." In the figure caption we have now: "Figure 2: Monthly Wadden Sea export of DIC and TA [Gmol mon$^{-1}$] at the North Frisian coast (N), East Frisian coast (E) and the Jade Bay in scenario B. The export rates were calculated for DIC and TA based on measured concentrations and simulated water fluxes.".

*Some minor comments and suggestions*

*1. To avoid confusion, please unify abbreviations for North Frisian coast (N or NF),East Frisian coast (E or EF) and Jade bay (J or JB).*

Done

*"The respective areas 1-3" in line 350 also refers to the three regions?*

The three areas concerning the flushing times are the validation area and the western and eastern part of the validation area. We added in chapter "3.3 Hydrodynamic conditions and flushing times": "The flushing times were determined for the three areas 1 – validation area, 2 – western part of the validation area, 3 – eastern part of the validation area.". In addition we define in chapter "2.1.1 Model domain and validation area": ".The validation area is divided by the magenta dashed line at 7° E into the western and eastern part.".

*2. What is "yr" in Equation (1)? Please clarify.*

yr = year. We clarified this in the text.

*3. Lines 440-441: Why does this partly compensate the missing TA generation by benthic denitrification? Please explain.*

We added: ". This amount of nitrate would not fully be available for primary production if parts of it would be consumed by denitrification.".

*4. Line 453: Carbonate dissolution cannot be counteracted by DIC additions. DIC additions (mostly refer to free CO2) are usually in favor of carbonate dissolution.*

We rephrased this section: "Dissolution Dissolution of biogenic carbonates may be an efficient additional enhancement of the $CO_2$ buffer capacity (that is: source of TA), since most of the tidal flat surface sediments contain carbonate shell debris (Hild, 1997). On the other hand, shallow oxidation of biogenic methane formed in deep and shallow tidal flat sediments (not modelled) (Höpner & Michaelis, 1994; Neira & Rackemann, 1996; Böttcher et al., 2007) has the potential to lower the buffer capacity, thus counteracting or balancing the respective effect of carbonate dissolution."

*5. Lines 534-535: Why was Riv_eff not taken into account for the budget calculations? I notice that the authors mentioned it to the later discussion, i.e. lines 557-559 "Summing up the source and sinks, Wadeen Sea exchange rates, internal processes and effective river loads resulted in highest sums in 2002 and 2003 and lowest in 2009".*

This was also an issue of Reviewer#1. We clarified this point in chapter "2.2.2 River input": "Bulk alkalinity discharged by rivers is quite large but most of the rivers entering the North Sea (here the German Bight) have lower TA concentrations than the sea water. In case of identical concentrations the effective river load $Riv_{eff}$ is zero. The TA related molecules enter the sea, and in most cases they are leaving it via transport. In case of tracing or budgeting both the real TA river discharge and the transport must be recognized. In order to understand TA concentration changes in the sea $Riv_{eff}$ is appropriate.".

*6. Line 552-556: How to get those percentages (47%, 10% and 59%) based on results*

*in Table 2. Please clarify.*

This was also an issue of Reviewer#1. We added in chapter "4.3 TA budgets and variability of TA mass in the German Bight": "Comparing the absolute values of all sources and sinks of the mean year results in a relative ranking of the processes.". The values has been changed, as in the previous version we erroneously excluded the actual river input.

*7. Lines 615-616: The reduction of TA here is associated with the oxidation of ammonium to nitrate, instead of the change in DIC. Refers to Zhai et al. (2017,*

*https://doi.org/10.1016/j.ecss.2017.08.027).*

We changed the text: ".. and nitrification of ammonium (consuming TA, TA/DIC ratio is -2, see Pätsch et al., 2018 and Zhai et al., 2017).".

*8. Lines 617-619: The TA/DIC ratio of denitrification is not 1 but 0.8.*

We changed : " denitrification (TA / DIC ratio of 0.8, see Rassmann et al., 2020)"

*The TA/DIC ratio of sulphate reduction is not 2 but 1. Refers to Sippo et al. (2016), Are mangroves drivers or buffers of coastal acidification? Insights from alkalinity and dissolved inorganic carbon export estimates across a latitudinal transect, Global Biogeochemical Cycles, 30, 753–766, doi:10.1002/2015GB005324.*

We added: ".. and anaerobic processes related to sulphate reduction of organoclastic material (TA / DIC ratio of 1, see Sippo et al., 2016)."

*9. Lines 619-620: The authors mentioned that aerobic and anaerobic oxidation of upward diffusing methane were not considered in present study. How to relate this statement to line 633 "When sulphate reduction associated with organic matter and/or methane oxidation and pyrite burial became the dominant processes..."?*

We cancelled the content of lines 619-620

*10. Lines 624-638 and Figure 9: TA/DIC ratios of <–0.16 in regions under study may indicate other processes than aerobic decomposition (–0.16) and anaerobic reaction (>0). What are them? I would like to suggest the authors mention more possible processes at the beginning of Section 4.4.*

We added: "Candidate processes are numerous and the export ratios certainly express various combinations, but the most quantitatively relevant likely are aerobic degradation of organic material (resulting in a reduction of TA due to nitrification of ammonia to nitrate with a TA / DIC ratio of -0.16),  denitrification (TA / DIC ratio of 0.8, see Rassmann et al., 2020), and anaerobic processes related to sulphate reduction of organoclastic material (TA / DIC ratio of 1, see Sippo et al., 2016). Other processes are aerobic (adding only DIC) and anaerobic (TA/DIC ratio of 2) oxidation of upward diffusing methane, oxidation of sedimentary sulphides upon resuspension into an aerated water column (no effect on TA/DIC) followed by oxidation of iron (adding TA), and nitrification of ammonium (consuming TA)."

*11. Table 2 is discussed after all other tables. Please change the order of the tables. Additionally, Table 6 can be shifted to the Appendix.*

Done

*12. Are those TA/DIC ratios presented in Figure 9 mean values in the given regions? Please clarify.*

The basis of Fig. 9 are the data of the new Table 1. This means that the TA/DIC ratios are not average values. We added a corresponding statement in the caption of Fig. 9.

*Also I would like to suggest the authors compare all data in grid cells with the typical stoichiometric ratios of biogeochemical processes. Refers to Figure 4 in Sippo et al. (2016), Global Biogeochemical Cycles.*

Fig. 4 in Sippo et al. (2016) shows measured data. Our new Table 1 comprises too less data for such an exercise. To use simulated data is not necessary, as the different processes (if included in the model) are known.

*13. The "Wadden Sea tidal flats" should appear in the title.*

As we only analysed parts of the Wadden Sea we stay with the old title.

---

## Author Comment (AC3) · 19 May 2020

Dear Referee #1, please find attached our answers and comments. Thanks for reviewing.

Please also note the supplement to this comment:
https://www.biogeosciences-discuss.net/bg-2020-24/bg-2020-24-AC3-supplement.pdf

[Figure]

[Figure]

[Figure]

Fig. 1

**Fig. 1.**

[Figure]

[Figure]

Fig. 2

**Fig. 2.**

[Figure]

[Figure]

Fig. 3

**Fig. 3.**

[Figure]

Fig. 4

**Fig. 4.**

[Figure]

Fig. 5

**Fig. 5.**

[Figure]

[Figure]

Fig. 6

Fig. 6.

[Figure]

Fig. 7

**Fig. 7.**

[Figure]

Fig. 8

**Fig. 8.**

[Figure]

Fig. 9

**Fig. 9.**

---

## Author Response (AR2)

Answers to the reviewers of "The impact of intertidal areas on the carbonate system of the southern North Sea" by Fabian Schwichtenberg, Johannes Pätsch, Michael Ernst Böttcher, Helmuth Thomas, Vera Winde, Kay-Christian Emeis.

The authors thank the reviewers for their comments to improve the BGD-manuscript.

In the following we list all reviewer comments (blue, italic) and give answers (black).

*Anonymous Referee #1*

*I went through the responses to both reviewers' comments and the revised manuscript version. I believe they have well addressed most of my earlier comments. The structure has much improved, the aim of the manuscript has become clearer, and the TA budget and TA/DIC ratios are better explained.*

*A few minor comments I still have:*
*- The abstract could still benefit from a concluding statement*

We introduced the following sentence at the end of the abstract: "Despite of the scarcity of high-resolution field data it is shown that anaerobic degradation in the Wadden Sea is one of the main contributors of elevated summer TA values in the southern North Sea."

*- I don't fully understand your description on L. 158-162. On L.158 you mention "pelagic and benthic degradation and remineralisation" If I understand your description well, this is only aerobic (since on L.160-161 you write: "Benthic denitrification and other anaerobic processes have no impact on pelagic TA concentrations in this model version". However, "the remineralisation products ammonium and phosphate"(L.162) can also come from anaerobic decomposition of organic matter. Please clarify in your text.*

You are right, this may be confusing: We write now: "In this model version benthic denitrification has no impact on pelagic TA concentrations. Other benthic anaerobic processes are not considered. Only the carbonate ions from benthic calcite dilution increase pelagic TA concentrations. Aerobic remineralisation releases ammonium and phosphate, which enter the pelagic system across the benthic-pelagic interface and alter the pelagic TA concentration."

*- L.169: "above" where exactly? Add a reference to the corresponding section. I believe they are only discussed afterwards, equation (2) is presented much later in the text.*

Yes, "above" stems from the original text. We write now "For scenario B we used the same model configuration as for scenario A and additionally implemented Wadden Sea export rates of TA and DIC as described in section 2.3.1."

*- L.220-225 are a great addition and very helpful but an actual definition (in words) of effective river input is still lacking. I suggest to add that around the introduction of eq. 1.*

We augmented the first sentence of this section, and write now: "In order to analyse the net effect on concentrations in the sea due to river input, the effective river input ($Riv_{eff}$ [Gmol yr$^{-1}$]) is introduced:"

*- It might make sense to swap the order of 2.3.1 and 2.3.2, or at least refer to 2.3.2 on L.252*

After equation (2) we reference now to 2.3.2: "Differences in measured concentrations in the Wadden Sea during rising and falling water levels, as decribed in section 2.3.2, were temporally interpolated and summarized as wad_sta [mmol m$^{-3}$]."

*Anonymous Referee #2*

*Technical corrections:*
*Note that TA is a equivalent, rather than a concentration.*

We have chosen to apply SI units or direct derivatives in our study. As such the unit attributed to alkalinity is the one of a concentration. Accordingly, we have chosen to attribute concentration to alkalinity.

*Lines 128-129, "It is applied to several regional sea areas worldwide", references?*

We added corresponding references.

*Lines 569-571, Note that oxidation of iron will reduce TA, and nitrification of ammonium consumes TA without changes in DIC.*

You are right, oxidation of iron reduces TA. We changed the text accordingly: " .. by oxidation of iron (consuming TA) .."